# On the long-range offshore transport of organic carbon from the Canary Upwelling System to the open North Atlantic

Elisa Lovecchio[1], Nicolas Gruber[1], Matthias Münnich[1], and Zouhair Lachkar[2]

[1]ETH-Zürich, Universitätstrasse 16, 8092 Zürich, Switzerland
[2]Center for Prototype Climate Modeling, New York University Abu Dhabi, Abu Dhabi, UAE

*Correspondence to:* Elisa Lovecchio (elisa.lovecchio@usys.ethz.ch)

**Abstract.** A compilation of measurements of Net Community Production (NCP) in the upper waters of the eastern subtropical North Atlantic had suggested net heterotrophic conditions, purportedly supported by the lateral export of organic carbon from the adjacent highly productive Canary Upwelling System (CanUS). Here, we quantify and assess this lateral export using the Regional Ocean Modeling System (ROMS) coupled to a Nutrient, Phytoplankton, Zooplankton, and Detritus (NPZD) ecosystem model. We employ a new Atlantic telescopic grid with a strong refinement towards the north-western African shelf to combine an eddy-resolving resolution in the CanUS with a full Atlantic basin perspective. Our climatologically forced simulation reveals an intense offshore flux of organic carbon that transports over the whole CanUS about 19 Tg C yr$^{-1}$ away from the nearshore 100 km, amounting to more than a third of the NCP in this region. The offshore transport extends beyond 1500 km into the subtropical North Atlantic, along the way adding organic carbon to the upper 100 m at rates of between 8% and 34% of the alongshore average NCP as a function of offshore distance. Although the divergence of this lateral export of organic carbon enhances local respiration, the upper 100 m layer in our model remains net autotrophic in the entire eastern subtropical North Atlantic. However, the vertical export of this organic carbon and its subsequent remineralization at depth makes the vertically-integrated NCP strongly negative throughout this region, with the exception of a narrow band on the north-western African shelf. The magnitude and efficiency of the lateral export varies substantially between the different subregions. In particular, the central coast near Cape Blanc is particularly efficient in collecting organic carbon on the shelf and subsequently transporting it offshore. In this central subregion, the offshore transport adds to the upper 100m as much organic carbon as nearly 60% of the local NCP, giving rise to a sharp peak of offshore respiration that extends to the middle of the gyre. Our modeled offshore transport of organic carbon is likely a lower bound estimate due to our lack of full consideration of the contribution of dissolved organic carbon and that of particulate organic carbon stemming from the resuspension of sediments. But even in the absence of these contributions, our results emphasize the fundamental role of the lateral redistribution of the organic carbon for the maintenance of the heterotrophic activity in the open sea.

# 1 Introduction

Owing to the dominance of the sinking flux of particulate organic carbon (POC), the ocean's biological pump is often simplified to a one-dimensional vertical process, consisting of the production of POC in the euphotic zone, its vertical export by gravitational sinking, and the remineralization of this organic carbon in the aphotic zone (e.g., Sarmiento and Gruber, 2006).
Reflecting this simplified view, most biogeochemical models currently used in the context of global climate models consider only the vertical export pathway for POC, thus neglecting its potential lateral transport by horizontal advection and diffusion (e.g., Aumont et al., 2003; Moore et al., 2004; Galbraith et al., 2010; Shigemitsu et al., 2012).

Nevertheless, the horizontal transport of POC can be substantial, even in the presence of vertical sinking speeds of 10 m day$^{-1}$ or more. This is especially the case in places characterized by high lateral advective velocities (Helmke et al., 2005) and the presence of upward vertical transports that can help to maintain high organic matter concentrations in the upper ocean against its gravitational sinking (Plattner et al., 2005). In addition, resuspension of bottom sediments can create nepheloid layers that can transport POC over hundreds of kilometers (Ohde et al., 2015; Inthorn et al., 2006a; Falkowski et al., 1994). A further important contribution to the lateral transport of organic carbon is that of dissolved organic carbon (DOC), estimated to account for 20 % of the export to depth and for about 10 % of the respiration rates in the deep ocean (Hansell, 2002; Arístegui et al., 2002; Ducklow et al., 2001). As a consequence of these transport processes, the different organic carbon pools get redistributed laterally from regions of excess production to regions of intense remineralization and burial (Inthorn et al., 2006b; Hwang et al., 2008), giving rise to a complex three-dimensional pattern of organic carbon cycling.

Such a lateral connection between organic carbon sources and sinks in the marine environment is at the heart of a long-standing controversy regarding the net metabolic state of the upper ocean in the oligotrophic subtropical gyres (Williams et al., 2013; Duarte et al., 2013; Ducklow and Doney, 2013). Based on a compilation of data from bottle-incubations that measure the net changes of oxygen over time, Duarte and Agustí (1998) and Del Giorgio and Duarte (2002) had suggested that oligotrophic systems, and particularly the near-surface layer in the center of the subtropical gyres tend to be heterotrophic. They suggested, although without any quantification, that this net heterotrophy is sustained by organic carbon that is supplied laterally to the center of the gyres from the adjacent more productive regions. This claim has fueled an intense debate, ranging from a discussion of the suitability of oxygen incubation experiments to assess the metabolic state of the ocean, to the question of whether it is actually possible to supply such a large amount of organic carbon through lateral processes (Williams et al., 2013; Duarte et al., 2013; Ducklow and Doney, 2013).

A key role in this debate is taken by the Eastern Boundary Upwelling Systems (EBUS), as these very productive continental margins (Chavez and Messié, 2009; Carr, 2002) are straddling the oligotrophic subtropical gyres. They thus may provide the source of the organic carbon that fuels the purportedly heterotrophic conditions in the latter regions (Liu et al., 2010; Walsh, 1991). In fact, with most of the NCP measurements indicating net heterotrophic stemming from the eastern subtropical North Atlantic (Duarte et al., 2013), the Canary Upwelling System (CanUS) has been at the center of studies addressing the offshore transport of organic carbon (e.g. Pelegrí et al., 2005).

Located on the eastern side of the North Atlantic Ocean, the CanUS spans the region between the North African Coast and the adjacent portion of the North Atlantic Gyre, between 9°N and 33°N (Váldes and Déniz-González, 2015). A complex circulation pattern determines strong subregional differences in the CanUS in terms of circulation, mesoscale activity, seasonality of upwelling and biology (Arístegui et al., 2009). The high productivity in the CanUS is sustained both by local coastal upwelling and by meridional alongshore advection of nutrients (Carr and Kearns, 2003; Pelegrí et al., 2006; Pastor et al., 2013; Auger et al., 2016). Sufficiently long water residence times in the nearshore region, favorable light and temperature conditions also contribute to sustain high levels of production (Lachkar and Gruber, 2011). Dedicated local surveys have demonstrated that the export of coastally produced organic carbon from the CanUS shelf to the open sea can be intense (Pelegrí et al., 2005; Arístegui et al., 2003) and include living organisms (Brochier et al., 2014). On top of the mean Ekman transport, persistent filaments originating on the CanUS shelf have been reported to be able to export up to 50 % of the coastally produced organic matter as far as several hundreds of km offshore (Gabric et al., 1993; Ohde et al., 2015). Due to these fluxes, a substantial amount of coastally produced organic carbon in the CanUS escapes remineralization in the nearshore region and is advected offshore towards the center of the North Atlantic Gyre (Fischer et al., 2009; García-Muñoz et al., 2005; Álvarez-Salgado and Arístegui, 2015). Estimates from multiple local surveys indicate that on average about 16 % of the coastal production by phytoplankton is laterally exported to the open sea (Duarte and Cebrián, 1996). The CanUS constitutes therefore a good potential candidate source region for the organic carbon required to fuel the purportedly heterotrophic conditions in the subtropical North Atlantic. However, despite its potential impact on the offshore biological activity (Alonso-González et al., 2009; Álvarez-Salgado et al., 2007), the magnitude and range of the total long-range offshore transport of organic carbon in the CanUS is still poorly quantified.

The quantification of this export is notoriously difficult to achieve through in-situ studies owing to the intermittency of the transport and the importance of eddies, filaments and other turbulent structures (e.g., Peliz et al., 2004; Nagai et al., 2015), providing models an opportunity to fill the gap. These models need to have relatively high resolution, in order to resolve this mesoscale dynamics, forcing most studies to employ regional models instead of global ones. But so far, relatively few high resolution modeling studies have focused on the CanUS compared to other upwelling regions and even fewer have employed a fully coupled biogeochemical model (Auger et al., 2016; Fischer and Karakaş, 2009; Lachkar and Gruber, 2011; Pastor et al., 2013).

Most of the modeling work conducted to assess the lateral redistribution of organic carbon relied on regional configurations of the Regional Ocean Modeling System (ROMS) and tended to focus on sub-aspects of the offshore transport of organic matter, either by focusing on sub-regions, or by focusing on the offshore transport of a subset of constituents. In the most recent study, Auger et al. (2016) used a regional ROMS configuration coupled to the biogeochemical/ecological module PISCES (Pelagic Interactions Scheme for Carbon and Ecosystem Studies) to highlight the important role of the lateral redistribution of nutrients and phytoplankton on the CanUS shelf for determining the complex seasonal pattern of chlorophyll and for the fueling of the persistent Cape Blanc offshore bloom. Despite the specific focus on phytoplankton, their analysis extended the work of previous studies which concentrated on the lateral transport of organic carbon in limited portions of the CanUS. Among these studies, Sangrà et al. (2009) estimated the integrated lateral export and production of organic carbon in the eddy corridor

shed by the Canary Archipelago and its potentially big impact on the region combining a physical ROMS simulation with estimates of carbon concentration in the eddies based on a few eddy surveys. With a ROMS-driven Lagrangian experiment, Brochier et al. (2014) studied the observed biological coupling between the African coast and the Canary Islands in terms of the offshore transport of ichtyoplankton (eggs and larvae of fish) by filaments. With a wider focus area, Fischer and Karakaş

(2009) employed a relatively simple Nutrient Phytoplankton Zooplankton and Detritus (NPZD) model coupled to their regional configuration of ROMS to study the role of sinking speeds in the vertical export of organic carbon to the deep ocean in the CanUS. While these studies demonstrated clearly the importance of offshore fluxes in the CanUS, they did not address the long-range transport of organic carbon from the CanUS into the oligotrophic subtropical North Atlantic. This is largely due to the limited offshore dimension of the regional ROMS configurations, with typical offshore extents of a few hundred kilometers

only.

   Here we overcome this limitation and provide a first comprehensive quantification of the long-range lateral fluxes of organic carbon from the CanUS shelf to the open North Atlantic using a new regional configuration of ROMS. This configuration employs a basin-scale telescopic grid that allows us to model the whole Atlantic basin in a continuous manner, while maintaining a full eddy resolving resolution in the region of study. Thus, this configuration is ideally suited to assess the long-range

transport owing to its fully resolving all scales. Furthermore, this permits us to push the lateral boundaries far away from the region of interest, thus avoiding the many challenges and distortions associated with the lateral boundary conditions in regional studies. We couple this physical setup with a NPZD-type ecosystem model that fully resolves the three-dimensional dynamics of the redistribution of POC. In particular, the vertical sinking of the different pools of organic matter is explicitly solved for, permitting us to advect and diffuse POC in the vertical and horizontal directions. Even though we are not taking the transport of

DOC explicitly into consideration, our results show that the organic carbon offshore flux in the CanUS significantly enhances the carbon availability of the open waters. Substantial subregional differences in the pattern of lateral and vertical fluxes and key pathways for the carbon lateral redistribution are highlighted and discussed in the context of the previous research.

## 2   Methods

### 2.1   Model configuration

We employ the UCLA-ETH version of the Regional Ocean Modeling System (ROMS) (Shchepetkin and McWilliams, 2005), coupled with the Nutrient, Phytoplankton, Zooplankton and Detritus (NPZD) biogeochemical ecosystem module of Gruber et al. (2006). ROMS solves the 3D hydrostatic primitive equations of flow on a discretized curvilinear grid, using terrain following vertical coordinates (sigma levels). Surface elevation, barotropic and baroclinic horizontal velocity components, potential temperature and salinity are its prognostic variables. Vertical mixing is parameterized by the K profile parameterization

(KPP) scheme (Large et al., 1994).

   The NPZD ecosystem module is a nitrogen-based model with two limiting nutrients, i.e., nitrate and ammonium, one class of phytoplankton, one class of zooplankton and two detritus pools, i.e., a small one that sinks very slowly, and a large one that is subject to more rapid sinking (Gruber et al., 2006). These components plus a dynamic chlorophyll-to-carbon ratio form

the seven prognostic variables of the nitrogen component of the model. An additional four state variables have been added to reflect the cycling of carbon and oxygen, namely dissolved inorganic carbon (DIC), alkalinity, mineral $CaCO_3$ and dissolved oxygen ($O_2$) (Hauri et al., 2013; Turi et al., 2014; Lachkar and Gruber, 2013). The carbon and oxygen cycles are linked to the nitrogen cycle by fixed stochiometric ratios. Thus, the fluxes of organic carbon diagnosed in our model are actually the fluxes

of organic nitrogen multiplied by the C:N ratio assumed in the model, i.e., 117:16.

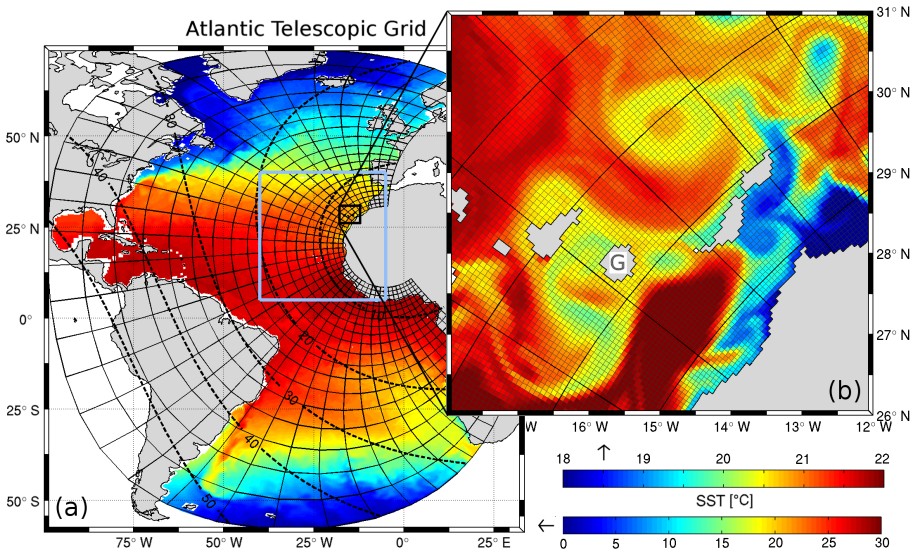

**Figure 1.** Map of the domain of the Atlantic telescopic grid together with a snapshot of the modeled sea-surface temperature. (a) Map of the full domain. Shown are every 20th grid line. Dashed isolines indicate the resolution of the grid in km. The light blue square highlights the region of interest used for the CanUS plots throughout the whole paper [5°N to 40°N, and -40°E to -5°E]; the black square indicates the region used for the zoomed-in subplot (b) Zoom on the Canary Islands region with actual grid resolution. Every 20th line is plotted thicker, corresponding to the grid lines shown in (a). As a point of reference, the island of Gran Canaria (G) has a diameter of about 45 km.

Particulate organic carbon (POC) in the model, i.e., the sum of phytoplankton, zooplankton, and the two detritus pools, is subject to advection and diffusion in vertical and horizontal directions, as well as explicit vertical sinking (the latter does not apply to zooplankton, however). Sinking velocities for phytoplankton, small detritus and large detritus are 0.5 m day$^{-1}$, 1 m day$^{-1}$, 10 m day$^{-1}$, respectively (see Gruber et al. (2011) for a complete set of parameters). Particulate organic carbon is

formed through phytoplankton growth and is lost through zooplankton respiration and the remineralization of the two detrital pools. Within the organic carbon pool, phytoplankton mortality feeds into the small detritus pool, while zooplankton mortality and phytoplankton excretion is routed to both the small and large detritus pools with constant proportions. Coagulation of phytoplankton with small detritus as well as coagulation of small detritus with small detritus also forms large detritus, while no disaggregation is considered in our NPZD model, i.e., large detritus cannot disaggregate back into small detritus. Bacterial

remineralization of organic matter is modeled as an implicit process through the definition of constant remineralization rates and takes place both in the water column and in the sediments. The sediments act as a temporal buffer in our model, receiving

the organic matter from the water column, and then slowly remineralizing it back to its inorganic constituents, which are released back immediately to the overlying water column. Neither burial nor sediment resuspension is considered. As the model does not include an explicit DOC pool, our modeled total organic carbon corresponds to POC only. However, the small detritus, given its very small sinking speed, behaves essentially as a suspended POC pool, i.e., shares many similarities to DOC.

To assess the possible implications of our neglecting DOC, we run a sensitivity experiment where we turned the small detritus pool into essentially DOC by setting its sinking velocity to zero and by reducing it coagulation rate.

In order to optimally model the long-range transport of organic matter from the CanUS to the open North Atlantic, we employ a newly developed Atlantic telescopic grid (Figure 1) that covers the full Atlantic basin (-60°N to 70°N) while having a strong focus in resolution toward the north-western African coast. This was achieved using a conformal mapping that moves

one pole of the ROMS grid over Northwest Africa (4°W, 20°N) and the other over Central Asia (75°E, 37°N). The grid has a dimension of 813 x 397 with a resolution that goes from a full eddy resolving 4.7 km near the African coast to a relatively coarse 50 km in the western South Atlantic and in the Caribbean (Figure 1). Within the CanUS region as defined in our analysis, the resolution ranges between 4.7 km at the coast and 19.5 km. In the vertical, we are considering 42 terrain following (sigma) levels with a surface refinement that allows to have a better vertical resolution in the euphotic layer. We use a new vertically

stretched grid that transitions more rapidly from a pure terrain following orientation to a more horizontal orientation in order to reduce spurious mixing in the stratified open ocean.

## 2.2  Boundary Conditions

The coupled ROMS+NPZD model was run with monthly mean climatological forcing at the surface including the fluxes of heat and freshwater, solar shortwave radiation, wind stress and atmospheric $pCO_2$. All forcings were derived from ERA-Interim

reanalysis (Dee et al., 2011), with the exception of atmospheric $pCO_2$, which was computed from the GLOBALVIEW marine boundary layer product (GLOBALVIEW-$CO_2$ (2011), see Landschützer et al. (2014) for details). A detailed description of the data sources used for the forcing is provided in Appendix A: Datasets, Table A1a and Table A1b. We next describe some corrections we had to apply to the forcing.

The ERA-Interim-based shortwave radiation and total heat flux fields have been shown to be biased high in regions of

persistent cover with stratocumulus clouds (Brodeau et al., 2010). This is particularly relevant in the southern subregion of the CanUS, where stratocumulus are very pervasive and not well represented in the ERA-Interim reanalysis. We thus apply corrections to the original ERA-Interim reanalysis fields, taking advantage of the work by the Drakkar community (Dussin et al., 2016). They have already developed corrected forcing fields, i.e., the Drakkar Forcing Sets (DFS) (Dussin et al., 2016) on the basis of the ERA-Interim reanalysis (Brodeau et al., 2010), providing the corrected fields for the downwelling surface

short wave radiation (DSWR) and downwelling surface long wave radiation (DLWR). We thus compute correction masks ourselves, using the difference between the DFS and the uncorrected ERA-Interim data sets as the basis. Concretely, we first computed monthly means of the DFS daily climatology DSWR and DLWR. The monthly climatological means of the same two variables from ERA-Interim for the period were then used as a reference to calculate correction masks (C) for each month by simply differencing, i.e., $C_{dswr} = ERA_{dswr} - DFS_{dswr}$ and $C_{dlwr} = ERA_{dlwr} - DFS_{dlwr}$. These correction masks were

then regridded to our grid and applied to our ERA-Interim-derived monthly climatological mean forcing solar radiation (S) and total heat flux (TH) so that $S' = S - C_{dswr}$ and $TH' = TH - (C_{dlwr} + C_{dswr})$.

Another correction to the forcing regards the regions with sea ice cover at the northern boundary of the domain. Here we account for freshwater and latent heat fluxes associated with sea ice formation and melting by correcting the surface fluxes of the model forcing. An offline correction for the forcing was calculated from the ERA-interim sea ice fraction ($c$) and NSIDC sea ice drift ($\boldsymbol{u}$) monthly climatologies. Using these two datasets, the corrections to the freshwater flux ($F_{fw}$) were calculated according to Haumann et al. (2016), but simplified by using monthly mean climatologies, a constant sea ice thickness $h_0 = 1.5$ m and a sea ice density $\rho_{ice} = 910$ kg m$^{-3}$. An analogous equation was used to calculate the correction to the heat flux ($F_h$), so that: $F_h = \rho_{ice} L \cdot h_0 (\partial (Ac)/\partial t + \boldsymbol{\nabla}(Ac\boldsymbol{u}))$ where $A$ is the grid box area, and $L$ the latent heat of fusion of water. The heat flux is constrained throughout the model simulation, so that the surface temperature cannot drop below the freezing temperature (Steele et al., 1989), which prevents strong heat loss in sea ice covered areas.

River runoff in the form of monthly climatological data (Dai et al., 2009) was also added to the freshwater fluxes. River sources were regridded to the closest ocean grid point and spread to a number of adjacent ocean grid points that depends the order of magnitude of the incoming flux to avoid numerical problems.

The model was run with open lateral boundaries at all grid boundaries confined by water, including at the Strait of Gibraltar. These climatological monthly lateral boundary conditions were prepared the same way as in previous studies (Lachkar and Gruber, 2013). A detailed description of the datasets used for the boundary conditions can be found in Appendix A: Datasets, Table A2.

## 2.3 Simulation and Analysis

The model was initialized to be at rest with temperature, salinity, nitrate, and the inorganic carbon parameters corresponding to the mean climatological value of December and January. The remaining biogeochemical variables were initialized to small non-zero values. The model was then run forward in time for a total of 35 years, using the monthly climatological forcing described above. We use the first 29 years as spin-up, and undertake our analysis using the last 6 years (years 30 to 35) of the simulation. This permits us to obtain a good representation of the climatological mean state of the CanUS, i.e., to average out the substantial intrinsic variability present in the setup.

To quantify the offshore transport of organic carbon including all its biogeochemical transformations, we undertake a full budget analysis, calculating all fluxes of organic carbon within each grid box and between all adjacent boxes. The organic carbon budget analysis fluxes include physical fluxes through the boundaries of the boxes and the integrated biological flux within each box. Physical fluxes include: vertical and horizontal advective fluxes in the three directions, vertical mixing fluxes associated with the eddy diffusivity and the vertical sinking flux. The net biological flux of organic carbon within each box is equivalent to the Net Community Production (NCP), i.e., the net amount of carbon added or removed by biological activity, computed by summing all organic carbon production processes (phytoplankton growth) minus the sum of all processes that convert organic carbon back to inorganic forms (respiration and remineralization). In our analysis we disregarded the contribution of the horizontal and vertical mixing associated with the background diffusivity. We also used a fixed depth for the sigma

layers that define the box boundaries, disregarding their vertical oscillations. Both approximations can result in small residuals in the budget analysis.

For the whole CanUS, defined as the region between 9.5°N and 32°N, we have defined two layers of 3-dimensional large-scale boxes. Each depth layer has a constant thickness of 100 m, very close to the mean depth of the euphotic layer in the

CanUS region (98.7 m), so that our analysis spans in total the first 200 m of depth. Each depth layer is subdivided into the same five offshore boxes up to 2000 km distance from the north-west African coast according to the following ranges of distances: (1) 0 to 100 km from the coast, narrow "coastal box" directly influenced by the coastal upwelling; (2) from 100 km to 500 km offshore; (3) from 500 km to 1000 km offshore; (4) from 1000 km to 1500 km offshore; (5) from 1500 km to 2000 km offshore.

To highlight the different roles of the three fundamental zonal bands in the CanUS, we divided the EBUS into three subre-

gions (Southern, Central, Northern), maintaining for each subregion the same five offshore domains as for the previous analysis and considering only the euphotic layer (0 m -100 m). Subregional boundaries were placed at 17°N and 24.5°N. This allows us to distinguish between three regimes of circulation and production of the CanUS: a northern subregion dominated by coastal upwelling and coastal filaments, a southern tropical subregion, and a central subregion where the Canary and Mauritanian Currents converge to form the Cape Verde front (Pelegrí and Peña-Izquierdo, 2015). The lateral extension of the full CanUS

boxes and of the subregional boxes is presented in Figure 3 together with the pattern of the modeled currents.

## 3   Evaluation

The model represents well the general circulation of the whole Atlantic Basin with a particularly good agreement between the modeled Sea Surface Height (SSH) and the observed one (Appendix B: Supplementary figures, Figure B1). Less well modeled is the SSH in the eastern side of the North Atlantic Gyre and in particular in the Gulf Stream region. Especially problematic

is the too northerly separation of the Gulf Stream from the North American coast, a problem shared with many ocean general circulation models. These deviations are likely connected to our relatively coarse resolution in that part of the domain (Figure 1). But they occur far away from the region of interest, and are thus considered tolerable for the purpose of this study. The SST pattern is also well represented; differences are concentrated in the near equatorial region and are probably connected to a weaker equatorial circulation and a possible residual overestimation of the net heat flux in this region despite the correction

we applied to the original forcing.

A zoom on the North-East Atlantic region allows us to evaluate the representation of the variables in our region of study, i.e., the CanUS. Here, modeled annual mean SST (Figure 2a) and SSH (Figure 2b) are well reproduced with a clearly visible signature of low SSH and cold water along the African coast as a result of the Ekman upwelling. Some differences in SSH are discernible at the northern boundary owing to the eastward flowing Azores Current being located slightly more south than

observed. A slight shift south is also visible at the southern boundary. Despite the stratus cloud correction, the modeled SSTs are still a bit too warm in the southern sector of the CanUS. However, differences between model and observations are limited to the interval [-0.75°C,1°C] over the large majority of the domain, with a large fraction of this bias having a range of only ±0.5°C. Larger differences are confined to a very narrow coastal band. The model also captures the observed Sea Surface

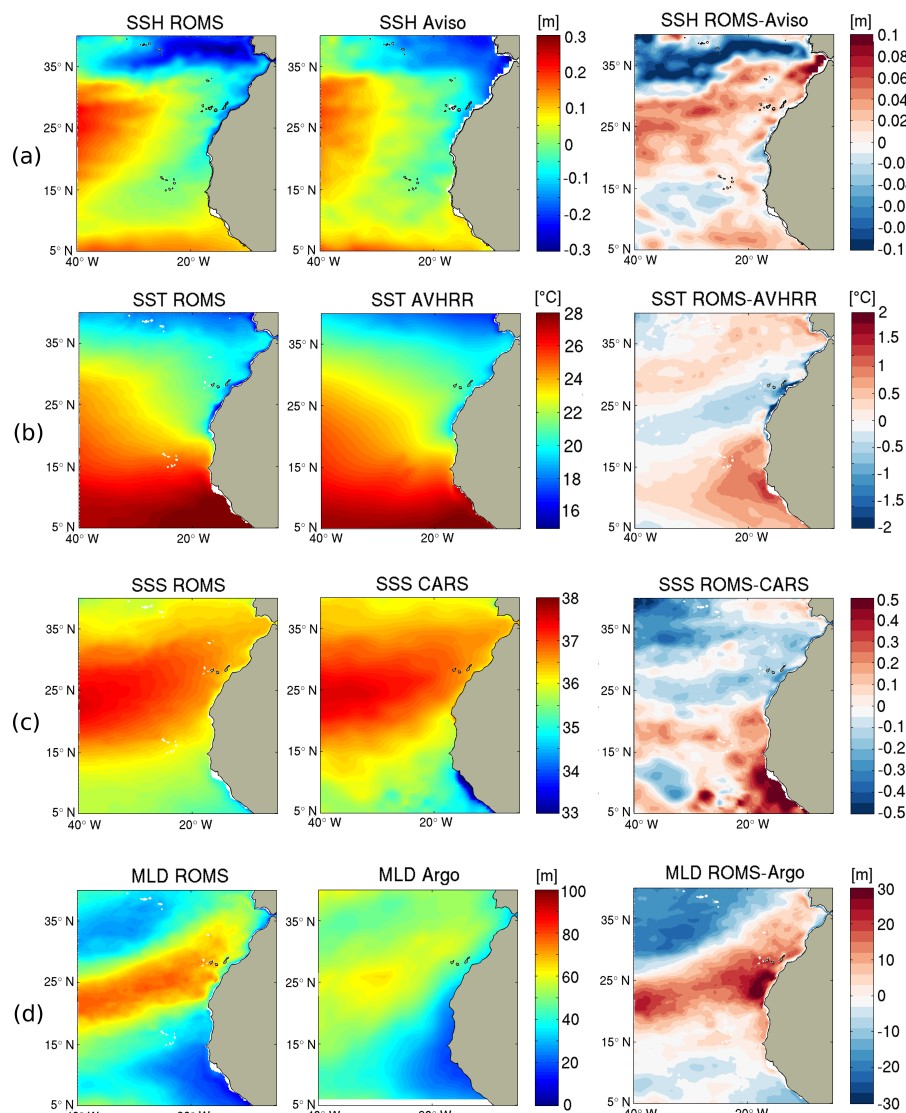

**Figure 2.** Evaluation of the modeled annual mean fields in the CanUS region. (a) Sea Surface Height (SSH); (b) Sea Surface Temperature (SST); (c) Sea surface Salinity (SSS); (d) Mixed Layer Depth (MLD). Left column: model; Middle column: observations; Left column: difference between model and data. A detailed description of the data used for the evaluation is provided in Appendix A: Datasets, Table A3.

Salinity (SSS, Figure 2c) well; relevant negative differences are only observed in the southern CanUS, in connection with the warm SST bias, resulting in a compensation of the density. Overall, we consider these biases to be small relative to the spatial and temporal variations, therefore we expect these SST and SSS biases to have a minor impact on the conclusions of our study.

The modeled annual mean Mixed Layer Depth (MLD, Figure 2d) is consistent with the general pattern of the Argo-based
5   MLD product, even tough the modeled pattern has sharper gradients. Deeper than observed values of the MLD are visible in the

northern sector of the CanUS and in the nearshore waters of the southern sector of the CanUS. It is worth noting that the Argo dataset was generated on a relatively low resolution grid, i.e., 2°x 2°, and thus is likely underestimating lateral gradients. In addition, the float coverage in Eastern Boundary Current system is relatively low, owing to the strong currents and the offshore transport, making the Argo-based MLD product vulnerable for biases in these regions. Nevertheless, some of the differences are likely real, as they appear also in other products. This is particularly the case for the overestimation of MLD in the nearshore region of the southern CanUS and in the long strip extending southwestward from the Canary Island, possibly due to biases in the position of the large-scale currents as evidenced by the differences in SSH (Figure 2a).

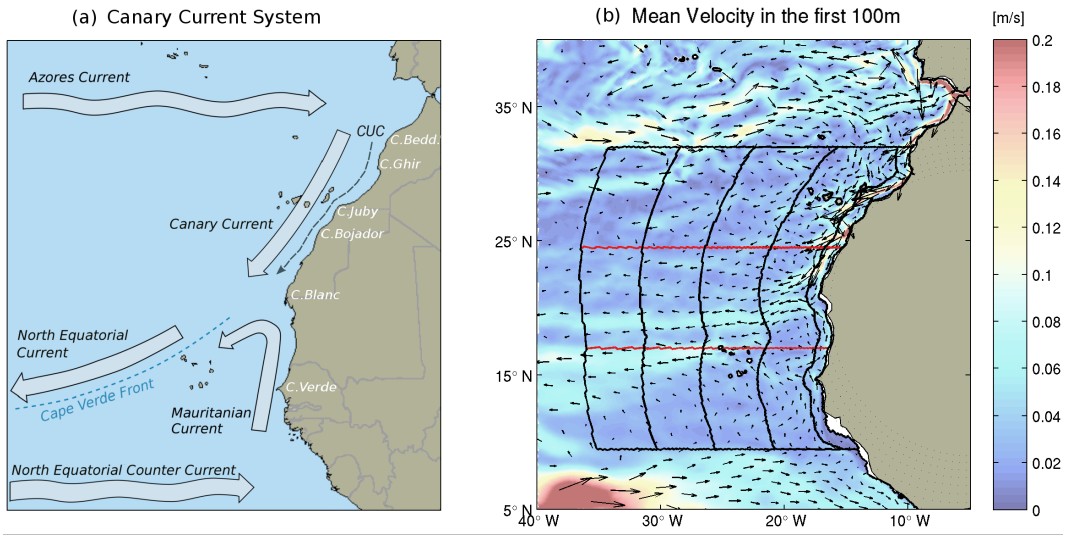

**Figure 3.** Maps of the circulation of the Canary Current System. (a) Schematic depiction with major currents adapted from Arístegui et al. (2009); (b) Modeled system of currents: vertically averaged flow in the first 100 m depth. Black lines represent the boundaries of the budget analysis regions for the Full CanUS analysis, while red lines indicate the 2 extra subregional boundaries. The Full CanUS covers the region between 32°N and 9.5°N. Subregional boundaries are at 17°N and 24.5°N. From East (African coast) to West boxes span the following ranges of offshore distances: 0 km-100 km, 100 km-500 km, 500 km-1000 km, 1000 km-1500 km, 1500 km-2000 km.

The modeled annual mean circulation averaged over the first 100 m of depth corresponds well to the system of currents described schematically in Mackas et al. (2006) and Arístegui et al. (2009) (Figure 3). The Canary Current System is delimited on the northern edge by the eastward flowing Azores Current and on the southern edge by the eastward flowing North Equatorial Counter Current. Within these boundaries two currents flow in opposite directions along the African coast: the Canary Current (CC) flows southward between Cape Beddouza (33°N) and Cape Blanc (21°N), while the weaker and seasonal Mauritanian Current (MC) flows northward between 10°N and Cape Blanc (21°N). Between the CC and the African coast, an intense and narrow Canary Upwelling Current (CUC) flows southward along the shelf (Mason et al., 2011). A poleward undercurrent (not shown) flows along the whole North African coast with its core typically centered at 200 m - 300m depth (Pelegrí and Benazzouz, 2015). Next to Cape Blanc, both the CC and the MC detach from the coast and flow offshore forming the Cape

Verde frontal zone, a natural boundary for the flow of water masses and tracers in the region. This front divides the region into a northern so-called Moroccan subregion and a southern Mauritanian-Senegalese subregion that differ in both physical circulation and biological activity.

The offshore gradients in annual mean sea surface chlorophyll (CHL) are well captured by the model (Figure 4a) in the northern CanUS, where the absolute values are very close to the observations. Less well captured is the surface CHL in the productive southern sector of the CanUS, where the model substantially underestimates CHL at the surface. This is also the region where the model is biased too warm and salty, and where the modeled MLD exceeds the expected near-zero value, suggesting that this low surface CHL is primarily consequence of our physical biases in circulation and vertical stratification.

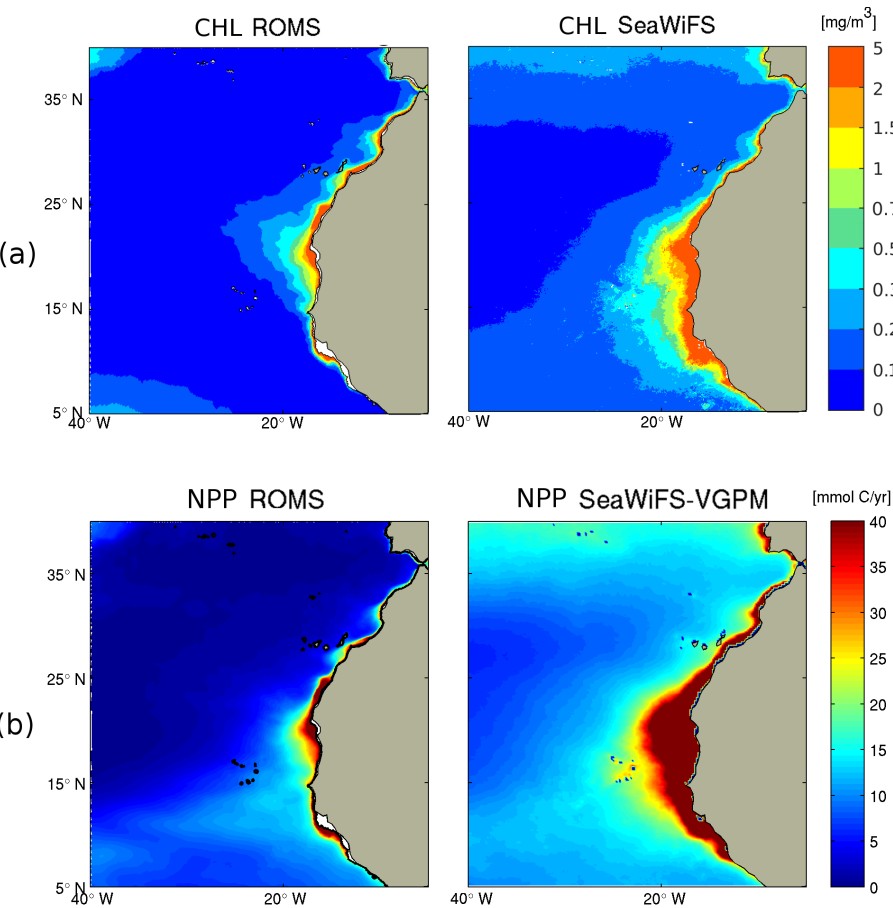

**Figure 4.** Maps evaluating the modeled chlorophyll and primary production in the CanUS. (a) Comparison of annual mean surface chlorophyll (CHL) between ROMS (left column) and SeaWiFS (right column). (b) Comparison of annual vertically integrated Net Primary Production (NPP) between ROMS (left column) and the VGPM estimate on the basis of the SeaWiFS data (right column). A detailed description of the data used for the evaluation is provided in Appendix A: Datasets, Table A3.

Our analysis of the vertical CHL distribution reveals that in the southern CanUS the modeled CHL has a deep maximum located between 20 m and 50 m depth, while little of the CHL is found in the surface layer (see Appendix B: Supplementary figures, Figure B2). The depth of the modeled CHL maximum is deeper than what can be expected for this region, according to local surveys. This deep bias of the chlorophyll maximum, and therefore of production, may be connected to the too deep

modeled mixed layer in the surroundings of the Cape Verde Islands, a region in which the observed MLD is very shallow, or to a too fast depletion of the nutrients at the upper edge of the nutricline. This deeper than observed primary production may result in a less intense lateral transport of CHL in this subregion given the decline of the advective currents with depth. This potential limitation will be discussed in depth in throughout the paper.

Modeled vertically integrated chlorophyll (not shown) as well as total annual Net Primary Production (NPP) (Figure 4b),

show instead intense biological activity in the southern CanUS, where NPP reaches values higher than in the northern sub-region, especially offshore. However, modeled values of NPP are lower than the SeaWiFS VGPM estimates by about 3-fold. Other NPP estimates such as those based on SeaWiFS CbPM and Modis-Aqua VGPM (a detailed descriptions of all the used datasets is provided in Appendix A: Datasets, Table A3) provide substantially lower estimates of primary production that are slightly closer to our modeled values, but the comparison does not substantially change the picture. Despite this underestima-

tion, the pattern and the offshore gradient of the modeled NPP agree with the estimates and allow us to discuss the impact of the organic carbon fluxes in terms of relative changes in the local carbon availability. A nearly homogeneous 3-fold increase of the modeled NPP would in fact not affect this analysis, even though it would likely change the absolute values of the fluxes of organic carbon that may exceed those found by our study.

Modeled Particulate Organic Carbon (POC) concentrations have annual mean values between 5 mmolC/m$^3$ and over 20

mmolC/m$^3$ in the first 100m depth of the very productive shelf areas laying therefore in the range of in situ observations (Alonso-González et al., 2009; Arístegui et al., 2003; Santana-Falcón et al., 2016; Fischer et al., 2009). Concentrations decline in the offshore direction with a pattern similar to that of NPP and have maximum values located between 20 m depth in the shelf area and 70 m depth offshore. The modeled POC compares well to the limited in situ data (see Appendix B: Supplementary figures, Figure B3) especially with regard to the vertically-integrated stocks in the first 100m. However, due to our coastally-

confined production combined with the fact that cruise data were mostly collected offshore, and due to the deepening of the chlorophyll maximum in the southern productive subregion, we observe a deeper-than-expected POC maximum in the model, in agreement with the vertical bias in CHL. Due to the absence of sediment resuspension and of a mechanism of disaggregation of the large detrital particles in the model, deep peaks of POC such as those present in Alonso-González et al. (2009) and Álvarez-Salgado and Arístegui (2015) are not observed in the annual mean modeled POC concentration.

A further important evaluation concerns the seasonal cycle, especially since the CanUS is characterized by the most intense seasonal variability among all EBUS (Chavez and Messié, 2009). The first 2 columns of Figure 5 show a comparison between model and observations of the seasonal variations of the circulation in the CanUS averaged in the first 15m, the depth of integration of the drifters. The plot reveals that the modeled CanUS circulation agrees well with the data collected by the drifters on the seasonal scale. As expected, an enhanced offshore flow is visible in summer in the northern Moroccan subregion

and in winter and spring in the southern Mauritanian-Senegalese subregion. The alongshore Canary Current is clearly visible

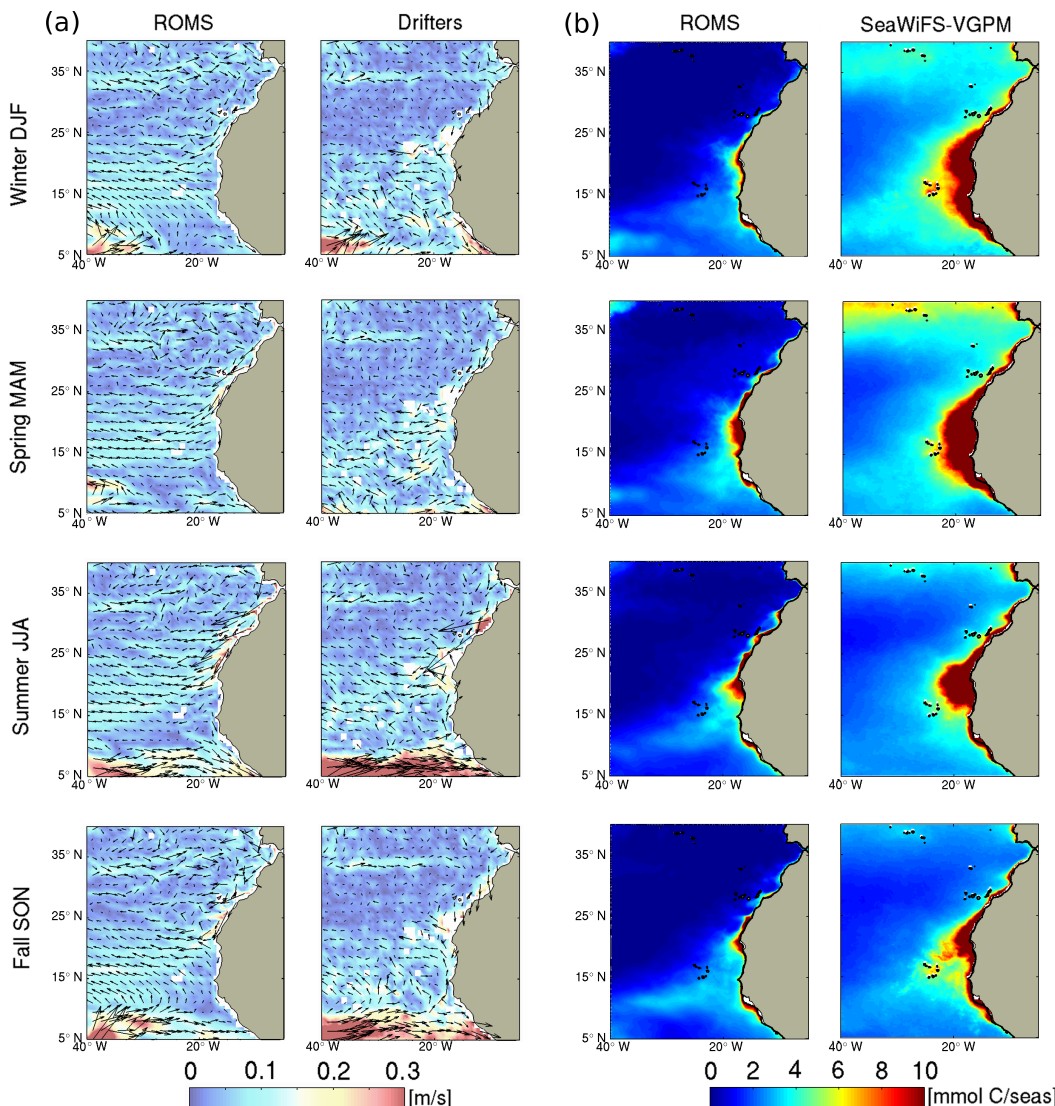

**Figure 5.** Panel (a): Circulation in the CanUS by season. ROMS (left column) and drifters (right column)]. ROMS output was integrated in the first 15m depth to be comparable with the drifter data. Panel (b): Vertically integrated Net Primary Production (NPP) from ROMS (right) and SeaWiFS VGPM estimate. A detailed description of the data used for the evaluation is provided in Appendix A: Datasets, Table A3.

in the northern sector of the CanUS. The Mauritanian Current seems to be weaker than observed especially in summer and to a smaller extent in fall. However, the Mauritanian Current is clearly visible in Figure 3b, in which the simulated flow is vertically integrated over the first 100m depth. Vertical sections of the modeled meridional flow (not shown) also show a clear northward flow corresponding to the Mauritanian Current below 10m depth. The modeled Mauritanian current is therefore slightly deeper than observed, possibly due to a deeper MLD observed in the southern CanUS coast (see Figure 2). The seasonality of the

Equatorial Counter Current (NECC) is well represented, even though its modeled flow in summer and fall is less intense than in the drifters data. The weaker than observed circulation in the southern sector of the CanUS will be taken into account in the discussion of the model results.

Since we are interested in quantifying the offshore fluxes of organic carbon throughout the upper few hundreds meters, vertically-integrated NPP is a better measure than surface Chlorophyll for evaluating the capacity of our model to reproduce the expected pattern of organic carbon. NPP in the CanUS is strongly influenced by the pattern of currents: the Cape Verde Frontal Zone separates a southern area of extended offshore production from a northern subregion in which productivity declines offshore (Arístegui et al., 2009). Both previous studies and SeaWiFS estimates show that productivity in the northern CanUS is dominated by a summer peak, while productivity in the southern CanUS shows a peaks in late winter and spring (Pelegrí et al., 2005; Pastor et al., 2013). This seasonality and subregional variability is well represented by our model at all latitudes of the CanUS. In both model and observations, the southern Mauritanian-Senegalese subregion is the most productive area of the CanUS and is characterized by a reduced offshore gradient of NPP, while the northern Moroccan subregion is characterized by a sharp offshore gradient of production. The convergence of the coastal currents in the region of Cape Blanc fuels a persistent offshore bloom (Auger et al., 2016) that clearly appears both in the SeaWiFS product and in the model.

As visible from the Taylor Diagrams (Appendix B: Supplementary figures, Figure B4 and Figure B5) the agreement between the pattern of the physical and biological variables of interest is also confirmed by the good correlation between modeled and observed fields for both the annual and the seasonal means. All the variables have a correlation of 0.7 or higher with the observations in the annual mean (except cruise data POC) and 0.68 or higher in the seasonal. In the annual mean, the values of the normalized standard deviations are particularly high for annual mean MLD (1.5), which is, as discussed above, due to a combination of too low variations in the Argo-based observational product and overestimation of the MLD variations by the model. Low values of the normalized standard deviations (STD) are observed for surface POC (0.65), CHL (0.6) and for net primary production (NPP1) (0.35), the latter corresponding to NPP from the SeaWiFS VGPM product. This is likely due to the weaker intensity of the modeled blooms. Interestingly, if modeled NPP is compared to the SeaWiFS CbPM product (NPP2), the normalized STD increases to 0.75, reflecting the rather large uncertainties in the NPP inferred from observations . In the annual mean, values of correlation and normalized STD for MLD, SST and CHL are comparable to those presented for the CanUS in the ROMS+NPZD study by Lachkar and Gruber (2011), despite the the boundaries of our grid being much further away, and therefore providing much less constraints on the modeled physics and biology in the region of interest. When compared to studies that used ROMS+NPZD in other upwelling systems such as the California Upwelling System (Gruber et al. (2011), whole domain), our Taylor diagram shows a slightly worse correlation and comparable normalized STD of surface CHL in the annual mean but a better seasonal representation, while modeled NPP has comparable performances.

In summary, the evaluation revealed that our modeling system is well suited to investigate the offshore transport of organic matter from the nearshore regions of the CanUS into the North Atlantic. It also showed a couple of shortcomings, especially with regard to our lack of explicit consideration of the role of DOC, and a model biases in a few regions, especially in the southern part of the CanUS. We will investigate and discuss the impact of these shortcomings in the discussion section below.

# 4 Results

## 4.1 NCP: Linking sources and sinks of Organic Carbon

The simulation reveals in the long term mean a strong onshore-offshore difference in the vertically integrated NCP, here $\int$NCP, i.e., primary production minus respiration/remineralization integrated from the bottom of the ocean up to the surface including remineralization in the sediments (Figure 6a).

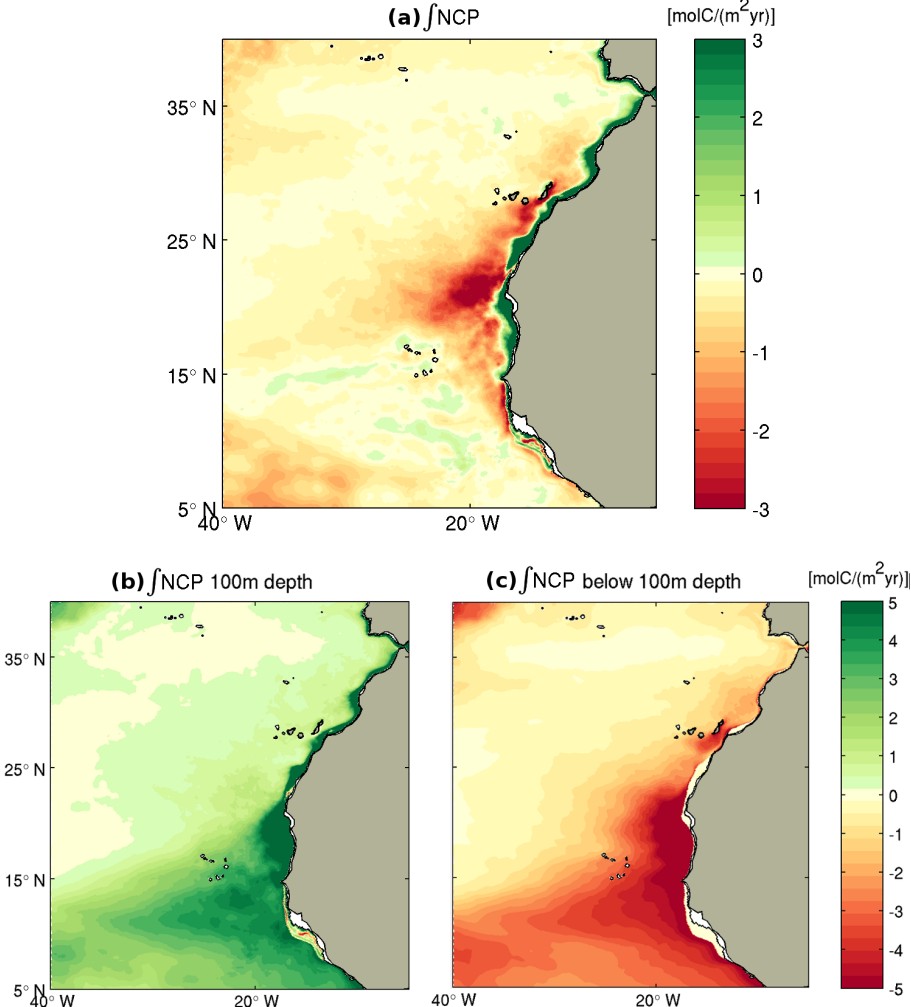

**Figure 6.** Maps of vertically integrated net community production including sediment remineralization. (a) NCP integrated over the full water column ($\int$NCP). This represents the net amount of organic carbon available for lateral redistribution. (b) NCP vertically integrated over the top 100 m only, including sediment remineralization. (c) As (b), for the depth range from 100 m to the bottom. Green indicates positive $\int$NCP (net source of organic carbon) while red means negative $\int$NCP (net sink of organic carbon).

The full water column ∫NCP is negative across nearly the entire eastern subtropical North Atlantic, while only a narrow strip of less than 100 km along the north-western African coast and a few offshore regions in the southern part of the domain have positive values. This implies that the majority of the region is net heterotrophic, as within each column of water including the sediments more organic matter is being consumed than what is being produced locally. In contrast, the shelf regions of the

CanUS are characterized by high levels of organic carbon production that exceed local consumption in the total water column and sediments, leading to a positive ∫NCP and therefore an excess of organic carbon, which is available for lateral export.

A very different offshore gradient exists if NCP is just integrated over the top 100 m (Figure 6b). Despite the full water column heterotrophy of the offshore waters, the top 100 m of the CanUS have a positive NCP at every latitude and distance from the coast, i.e., are, on average, a net source of organic carbon. In contrast, NCP integrated from 100 m depth downward

(Figure 6c) is everywhere negative, i.e., this part of the water column and the underlying sediments are net heterotrophic. Thus, the negative ∫NCP of the offshore waters (Figure 6a) arises from the excess of respiration at depth over the net production in the overlying surface ocean.

This switch between positive and negative NCP in the CanUS happens on average at ∼60 m in the nearshore regions deepening to ∼100 m in the offshore region, separating a layer of high net production from a layer of intense net respiration

(Figure 7a). This depth corresponds very closely to the euphotic zone depth, here defined by the level at which the light intensity at the surface is attenuated to 1%. Furthermore, this is also just below the depth of the maximum organic carbon concentration (Figure 7b). In the upper 100 m, the majority of this organic carbon stems from small detritus and phytoplankton, while below that depth and particularly in the nearshore areas, the large particles dominate. The contribution of zooplankton to the total organic carbon pool is substantial, but never dominant (see Appendix B: Supplementary figures, Figure B6).

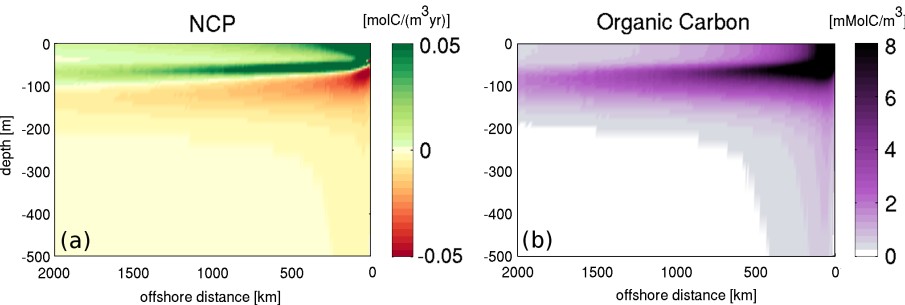

**Figure 7.** Mean vertical sections of (a) NCP and (b) POC in the Canary EBUS, averaged meridionally along lines of equal distance from the coast between 9.5°N and 32°N.

Since there is no substantial accumulation of organic carbon in the long term mean in any of the reservoirs, this onshore-offshore gradient in ∫NCP requires a considerable amount of organic carbon that is transported from the shelf region into the open subtropical North Atlantic. To understand this complex spatial pattern of autotrophic and heterotrophic activity in the region we next quantify the lateral and vertical fluxes of organic carbon in the CanUS.

## 4.2 Long-range offshore transport of organic carbon

The dominant nature of the offshore transport of organic carbon from the northwestern African shelf becomes clear by inspecting the annual mean and meridionally averaged section of the zonal flux of organic carbon (Figure 8a). This transport is nearly everywhere negative, indicating a westward, i.e., offshore transport, with the exception of the very nearshore region, where the narrow upwelling cell recirculates the organic carbon back onshore. This offshore flux spans the entire 2000 km range of distances from the coast in the first 200 m of depth, resulting in a continuous displacement of the organic carbon from the nearshore waters to the open sea. Only in the very nearshore region, the narrow upwelling cell causes the zonal flux to recirculate the organic carbon onshore.

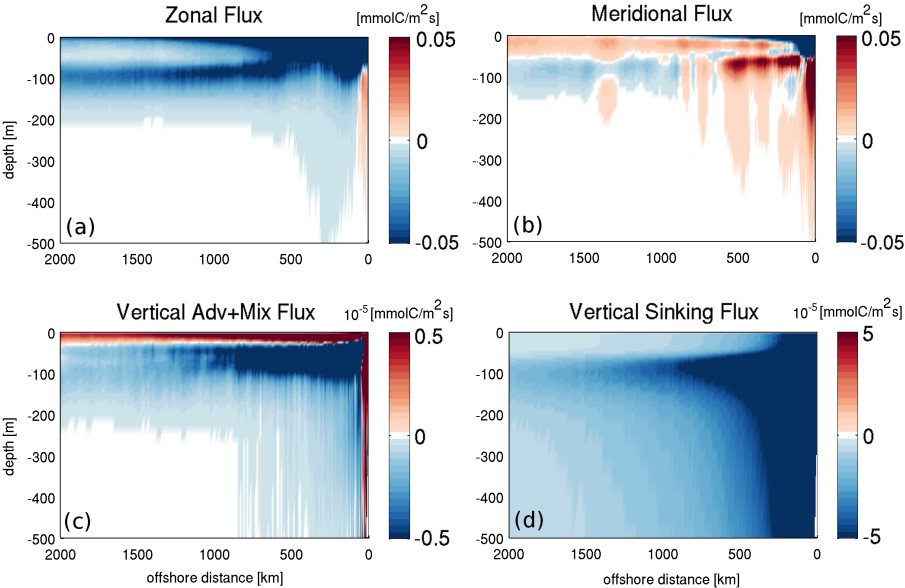

**Figure 8.** Mean vertical sections of the physical fluxes of organic carbon in the Canary EBUS, averaged meridionally along lines of equal distance from the coast between 9.5°N and 32°N. (a) Zonal flux of organic carbon with positive values indicating eastward (onshore) transport. (b) Meridional flux of organic carbon with positive values indicating northward transport. (c) Sum of vertical advective and mixing (eddy-diffusive) fluxes with positive values meaning upward transport. (d) Vertical sinking flux with negative values indicating downward transport. Note the different scales in the different panels.

The intensity of the offshore flux is maximum at the surface, and generally decreases with depth, except for the offshore regions, where a secondary maximum of zonal offshore transport occurs at around 100 m. Below that depth, the transport decreases rapidly and tapers off to very low values below 200 m with the exception of first 500 km from the coast. While the surface maximum of the offshore flux is mainly driven by the intense mean zonal velocity, the intensification of the flux around 100 m in the offshore waters is strongly influenced by the pattern of the organic carbon concentration (Figure 7b).

The lateral meridional flux (Figure 8b) shows a complex alternation of northward and southward fluxes, emerging from the integration of the meridional transport across a wide meridional band. Even though this flux is weaker than the zonal flux, this does not imply the absence of substantial alongshore currents within the domain. In fact, many of these currents get averaged out by the meridional integration. Despite this, the intense southward flowing Canary Current is still visible as a negative signature of the mean meridional flux near the coast. Northward fluxes, probably linked to an influx from the organic carbon-rich near equatorial region, are dominant further offshore.

The vertical advective and mixing (eddy-diffusive) fluxes (Figure 8c) are overall much weaker than the vertical sinking flux (Figure 8d). The latter, as expected from the fact that we employ constant sinking speeds has a pattern that reflects directly that of the organic carbon concentration (Figure 7b). In contrast, the fact that the vertical advective and mixing fluxes depend on the mean circulation results in a more complex pattern. These fluxes are positive both near the coast in response to the strong upwelling and in the upper first tens of meters where the vertical mixing redistributes the organic carbon against its vertical gradient. Below this shallow layer, subduction and downward mixing are dominant and contribute to the export of organic carbon to depth.

Reflecting the relative contribution of the different pools to the total organic carbon, the fluxes below the first 200 m as well as the vertical sinking flux are dominated by the contribution of the large detritus that reaches deep into the water column declining in concentration in the offshore direction. In the first 200 m, abundant small detritus, phytoplankton and to a smaller extent zooplankton shape the organic carbon fluxes up to the farthest boundary of the analysis domain.

## 4.3   The organic carbon budget

The annual mean budget of organic carbon for the upper waters of the whole CanUS highlights the key contribution of the offshore flux to the enhancement of the organic carbon pool and the maintenance of the heterotrophic activity in the open waters (Figure 9a and Figure 10a). In the upper 100 m, corresponding roughly to the euphotic layer, the offshore flux is the dominant lateral flux at all distances with a magnitude that always exceeds 10 % of the integrated NCP within the box (Figure 9a). More specifically, at 100 km from the coast the offshore flux of organic carbon transports as much as 1.6 Tmol C yr$^{-1}$ (18.7 Tg C yr$^{-1}$), a quantity that amounts to more than a third of the integrated NCP in the 0 km-100 km range, i.e. the first coastal box, and to 18 % of the Net Primary Production, NPP$_{100km}$ = 8.6 TmolC yr$^{-1}$.

A good measure for the magnitude and impact of the lateral redistribution of organic carbon is the difference between the organic carbon that is produced locally through NCP, and the amount of organic carbon that is exported vertically out of the euphotic zone (here 100 m). In the absence of any lateral redistribution, this difference, termed excess export, i.e., $\Delta E$ = Vertical Export - NCP is zero, while in the case of a strong lateral export of the organic matter, $\Delta E$ is negative since less carbon is available for the vertical export at depth. Conversely, if a particular region is importing a large amount of organic carbon through lateral transport and then exporting this carbon to depth, then the excess export $\Delta E$ is positive.

The analysis of $\Delta E$ as a function of offshore distance reveals that all regions have a positive $\Delta E$ with the exception of the nearshore one, whose $\Delta E$ is instead negative (Figure 9a). Thus, this supports the notion that the net heterotrophic activity over the whole water column in the offshore direction is fueled by a strong net growth in the very nearshore region of the CanUS.

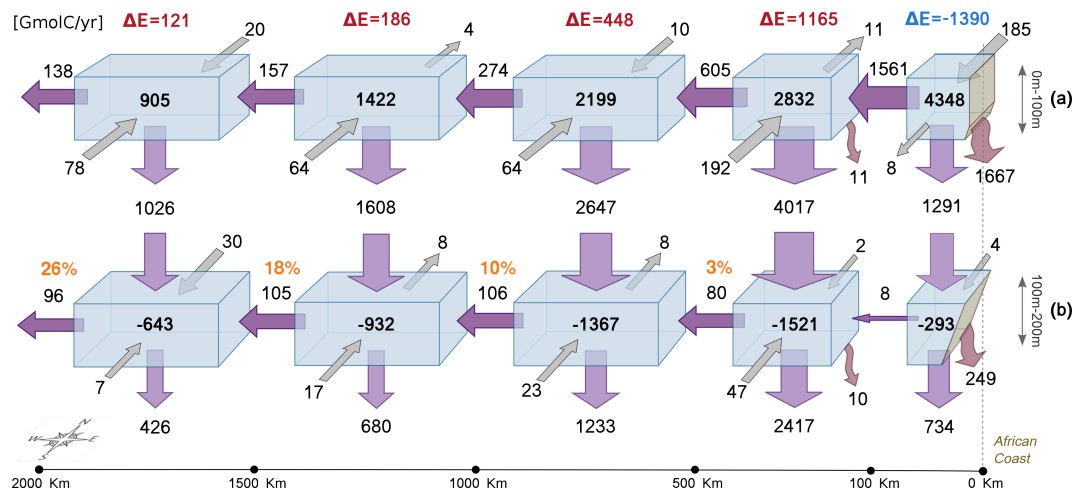

**Figure 9.** Annual mean Organic carbon budget for CanUS as a whole in units of GmolC yr$^{-1}$ for (a) the top 100 m, and (b) for the 100 m-200 m depth range. The lateral extension of the budget analysis boxes is shown in Figure 3b. The African coast is located on the right edge of the x-axis, with the offshore distance indicated at the bottom. Numbers inside the boxes represent the net biological flux in the volume (integrated NCP). The Arrows between boxes represent physical fluxes with the dimension of the arrows being scaled according to the magnitude of the fluxes. Lateral fluxes are advective, vertical fluxes are divided into fluxes between boxes (straight arrows, advective+mixing+sinking) and sinking fluxes to the sediments (bent arrows). The symbol $\Delta E$ = Vertical Export - NCP in (a) is a measure of the excess vertical export in each box with $\Delta E > 0$ indicating that the vertical export exceeds local NCP. The orange percentages in (b) represent the fraction of non-respired influx from above that is exported offshore.

The magnitude of the excess export at depth $\Delta E$ ranges between 41 % of NCP in the 100 km-500 km range to 13 % of NCP in the most distant region (Figure 10a, orange solid line) accounting for hundreds to thousands of Gmol of organic carbon per year. This excess export of organic carbon below 100m is explained by the divergence of the lateral fluxes. In particular, the divergence of the offshore flux (Figure 10a, purple solid line) releases in each region an amount of organic carbon that

constitutes between 8 % and 34 % of the local NCP, with the highest value in the 100 km-500 km offshore range, and explains with this organic carbon accumulation from 62 % to 80 % of the excess export at depth in the first 1500km offshore. In the most distant analysis region (1500 km-2000 km range) the offshore flux divergence drops, resulting in a significant export of organic carbon through the 2000 km offshore boundary and little accumulation; this flux may impact the biological activity even farther in the North Atlantic Gyre.

The alongshore lateral fluxes also positively contribute to the total budget with a net influx of organic carbon in the euphotic layer (Figure 9a, blue solid line). However, the divergence of the alongshore flux exceeds 10 % of the local NCP only in the most distant analysis region and represents therefore a minor contribution to the excess export at depth.

    The fate of the vertically exported carbon in the very biologically active 100 m-200 m depth layer is still strongly influenced by the offshore transport (Figure 9b), except for the nearshore, where the upwelling cell recirculates the organic carbon onshore.

The offshore flux intensifies moving away from the coast, reaching its maximum at 1500 km distance from the coast, where its

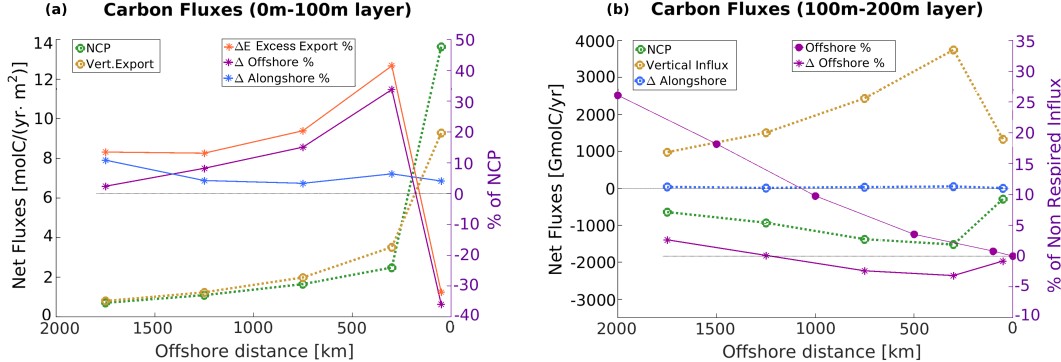

**Figure 10.** Main flux trends in the CanUS as a function of offshore distance for (a) the top 100 m and (b) for the 100 m-200 m depth range. In both panels: dotted lines refer to the left y-axis, solid lines refer to the right y-axis. Quantities represented by solid lines are expressed as percent of the significant carbon source for the layer, respectively: NCP for the top 100 m in (a), and the non-respired influx of carbon for the 100 m-200 m depth layer in (b). Δ fluxes represent the divergence of the fluxes: net amount of carbon accumulated or removed by the flux in each box. All net fluxes are binned at the center of the box of reference (e.g., fluxes in the 0 km-100 km region are binned to 50 km offshore), except for the offshore flux in panel (b), which refers to the boundaries of the boxes. Note that in panel a, the vertical export is a negative flux and that the yellow dotted line refers to its magnitude, and that ΔE is the excess vertical export as in Figure 9a). In panel b, the non respired influx is computed by summing the vertical influx, the alongshore flux, and NCP.

intensity becomes comparable to that of the top 100 m. Given the negative contribution of NCP to the organic carbon budget in this layer, the magnitude and divergence of the offshore flux at this depth can be compared to the non-respired influx of organic carbon, i.e., the amount of incoming carbon that is available in each box after remineralization. The main sources of organic carbon at this depths are the incoming vertical flux from the euphotic layer and to a very small extent the divergence of the

lateral alongshore flux (respectively yellow and blue dotted lines of Figure 10b). Therefore, the non-respired influx is defined as the sum of the incoming vertical flux, the divergence of the alongshore flux and the negative NCP.

If we compare the offshore flux at the boundary of the boxes to the non-respired influx in each box we see that the offshore flux moves a substantial amount of the organic carbon available, reaching 26% at 2000 km distance from the shore (purple solid line with circles of Figure 10b). At the same time, due to its intensification in the direction of the open sea, the offshore

flux does not release carbon in the boxes as confirmed by its negative divergence up to 1500km offshore distance (purple solid line with stars of Figure 10b and percentages in orange in Figure 9b), but it traps it and transports it even farther toward the open waters. The 100 m-200 m depth layer of the CanUS is therefore still characterized by a significant offshore transport that moves the organic carbon towards the oligotrophic center of the North Atlantic Gyre, furthering water column heterotrophy there.

As the offshore fluxes are small below 200 m (Figure 8a), we omitted them from the plot. In fact, the maximum contribution to the total offshore transport of the water column below 200 m, accounting for a few kilometers of depth, is 12 % reached at

500 km of distance from the shore. This fraction quickly declines offshore to a minimum of only 0.4 % at 2000 km of distance from the shore. Thus, the vast majority of the transport occurs in the top 200 m of the water column in our model.

## 4.4 Subregional variability of the organic carbon fluxes

Substantial meridional differences in both biological activity and circulation characterize the CanUS and influence the pattern of vertically integrated NCP (see Figure 6) and the implied lateral organic carbon fluxes. In the euphotic layer the pattern of production changes with latitude transitioning from a sharp offshore NCP gradient in the northern CanUS to an wide offshore extent of high NCP in the southern CanUS. These gradients can be explained by the pattern of the nutrients fluxes (see Appendix B: Supplementary figures, Figure B7). In the northern CanUS, nutrients are in fact mostly provided by coastal upwelling, while the positive signature of the wind-stress curl in the southern CanUS favors Ekman pumping of nutrients also offshore (Figure B7c). Intense production in the surroundings of Cape Blanc is likely due to the convergence of the alongshore nutrient fluxes (Figure B7b), in agreement with Auger et al. (2016) and Pastor et al. (2013).

Below 100m, the northern CanUS is characterized by a weak offshore gradient of deep respiration which, combined with a sharp offshore gradient of production in the layer above, results in an extended net water column heterotrophy in the open waters. In contrast, the southern CanUS is characterized by a widespread vertical correspondence between shallow sources and deep sinks of organic carbon that result in a vertically integrated NCP of nearly zero ($\int$NCP$\sim 0$), with negative values of $\int$NCP confined only between the African shelf and the Cape Verde archipelago. Between these two zonal bands with distinct $\int$NCP signatures, the central CanUS located in the surroundings of Cape Blanc (21°N) and the whole Cape Verde frontal zone are hot-spots for the respiration of the organic carbon. Here, the region of deep intense remineralization extends farther offshore than the area of intense near-surface productivity, resulting in a vast peak of negative $\int$NCP that reaches far into the North Atlantic Gyre. To identify what processes drive the organic carbon redistribution that give rise to these $\int$NCP gradients and quantify the contribution of the different zonal bands to the total transport we analyze in detail the spatial patterns of the physical fluxes of organic carbon.

Subregional differences in the organic carbon transport are visible in all of the components of the physical fluxes (Figure 11). Both zonal and meridional fluxes integrated over the top 100 m are clearly influenced by the regional pattern of currents (see also Figure 3b) and change sign in the proximity of the Cape Verde frontal zone, the crucial boundary between the northern anticyclonic and the southern cyclonic circulation. North of the Cape Verde front, the zonal flux is mostly offshore and intensifies moving towards Cape Blanc likely due to both the intense coastal mesoscale activity that culminates at 21°N with the giant Cape Blanc filament and to the formation of the Cape Verde front (Arístegui et al., 2009). South of Cape Blanc, ocean striations appear in the form of alternate onshore and offshore flux bands. The meridional transport converges around Cape Blanc, again with a sharp inversion of sign that reflects the direction of flow of the Canary Current and the Mauritanian Current.

The 100 m horizontal section of the vertical advective transport of organic carbon reflects the signature of the wind stress curl that is negative north of the Cape Verde front and positive to the south. As a consequence, the highest values of advective export at depth are found in the northern regions of low offshore production, while vertical advective export of organic carbon

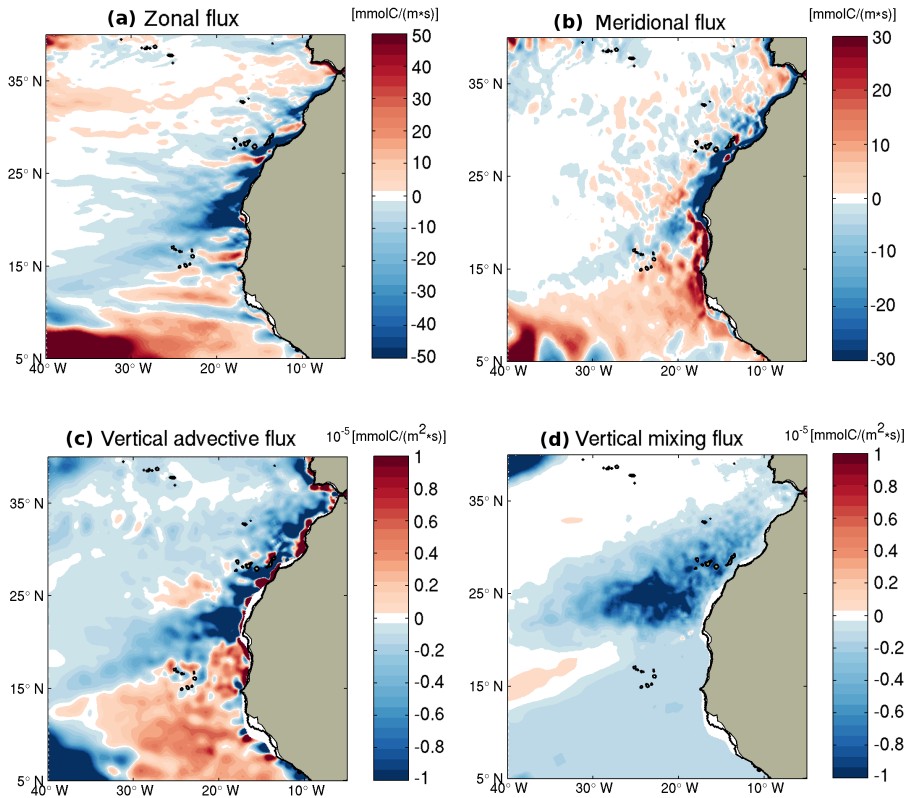

**Figure 11.** Maps of the organic carbon flux components in the top 100 m corresponding to the euphotic layer in the CanUS. (a) Zonal flux vertically integrated over the top 100 m with positive values indicating eastward (onshore) transport; (b) as (a), but for the meridional flux with positive values indicating northward transport; (c) vertical advective flux across 100 m with positive values indicating upward transport; (d) as (c) but for vertical mixing. Plotted vertical components were smoothed with a 7x7 grid points 2-dimensional filter.

is not favored in the very productive southern subregion. As for the offshore transport, also the vertical advective export in the northern CanUS sector and on the Cape Verde frontal zone is likely enhanced in the first few hundreds of kilometers by the abundant coastal filaments that quickly channel and downwell the coastally produced organic carbon. The vertical mixing fluxes of organic carbon across 100 m shows that this component is important only in the northern subregion characterized by
5   a much deeper MLD, and declines offshore with the decrease of the organic carbon concentration. Sinking fluxes through the 100m depth are not shown as their pattern is mostly proportional to the 100 m-integrated NCP (Figure 6b) with high export in regions of high production and do not add substantial complexity to the discussion. However, it is worth remarking that their intensity is about one order of magnitude higher than that of the vertical advective and vertical mixing fluxes, reaching very intense peaks ($\sim$-40 molC/(m$^2$day)) in the surroundings of the most productive region of Cape Blanc, and high values of
10   about -10 molC/(m$^2$day) in the Mauritanian-Senegalese subregion. The southern sector of the CanUS is therefore dominated

by intense sinking fluxes of organic carbon, while both non-sinking vertical components of the export have a very limited role in the organic carbon export in the region south of Cape Blanc.

The analysis of the spatial pattern of the organic carbon fluxes also remarks the special role of the central zonal band located between the Canary Archipelago and the Cape Verde Islands, characterized by the most intense biological and physical fluxes. This central zonal band is characterized by the strongest heterotrophic activity offshore, a persistent and intense offshore transport, a convergence of the lateral alongshore fluxes in the shelf, strong vertical advective export at depth, intense vertical mixing and a peak of the sinking flux. To highlight how differently the physical fluxes impact the organic carbon budget at different latitudes and to study the interaction between significant zonal bands in the CanUS we divided the region into southern, central and northern subregions (defined as in Figure 3b) and carried out a subregional box budget analysis.

Among the CanUS subregions, the Southern subregion (pink line of Figure 12) is characterized by very high levels of NCP in the euphotic layer also in the open waters (Figure 12a) accompanied by a relatively low impact of the physical fluxes of carbon on the local budget. In fact, despite both the offshore transport and the vertical advective+mixing export below 100 m have high intensities in absolute terms, their divergences are low when compared to the high values of NCP in each box and therefore they do not have a substantial impact on biology (Figure 12b and Figure 12d). On the one side, the little accumulation of organic carbon due to the offshore flux explains the large portion of biologically neutral water column in this region ($\int$NCP$\sim$0, see Figure 6a). On the other side, even though the sinking fluxes (Figure 12e) still export substantial amounts of carbon in this subregion, the low efficiency of the advective+mixing export points to a potentially limited capacity of this subregion to export at depth dissolved and suspended material (not modeled in our study). The comparatively higher impact of the alongshore flux is explained by the strong coupling of the southern subregion with the equatorial carbon rich area that allows a net influx of organic carbon through the southern boundary.

The impact of the lateral offshore transport in the Northern subregion (blue line of Figure 12) is particularly high in the first 500 km offshore (Figure 12b). This is the result of a combination of both strong export fluxes on the shelf and of the fast decline of NCP in the offshore direction, with a consequent high ratio of the offshore flux divergence to NCP and an important influence of the flux on the local budget. The intense mesoscale activity in this northern subregion, especially in the form of persistent filaments that detach from the coast and quickly channel water and tracers offshore typically for some hundreds km, has an important role in this intense nearshore export. The northern subregion is also the most efficient in the vertical advective+mixing export up to 1000 km offshore as a consequence of the deep MLD and the abundant mesoscale coastal filaments, with a consequently high capacity to export light organic carbon species below the euphotic layer. However, both the lateral and the vertical export efficiency decline quickly moving offshore in the northern subregion. This decline is likely due to the low organic carbon concentration of the offshore waters, to the limited offshore extension of the filaments and to the incoming flux of the Azores Current counteracting the offshore transport at the northern edge of the domain.

As anticipated, the most active area in terms of the organic carbon transport and export is the central CanUS subregion (green line of Figure 12), which includes the Cape Verde frontal zone. The central subregion collects the lateral alongshore fluxes from the northern and southern subregions in the nearshore area, with a net increase of the carbon availability in the first coastal box (0 km-100 km range) of more than 1/3 of the local NCP (Figure 12c). The southward flowing Canary Current

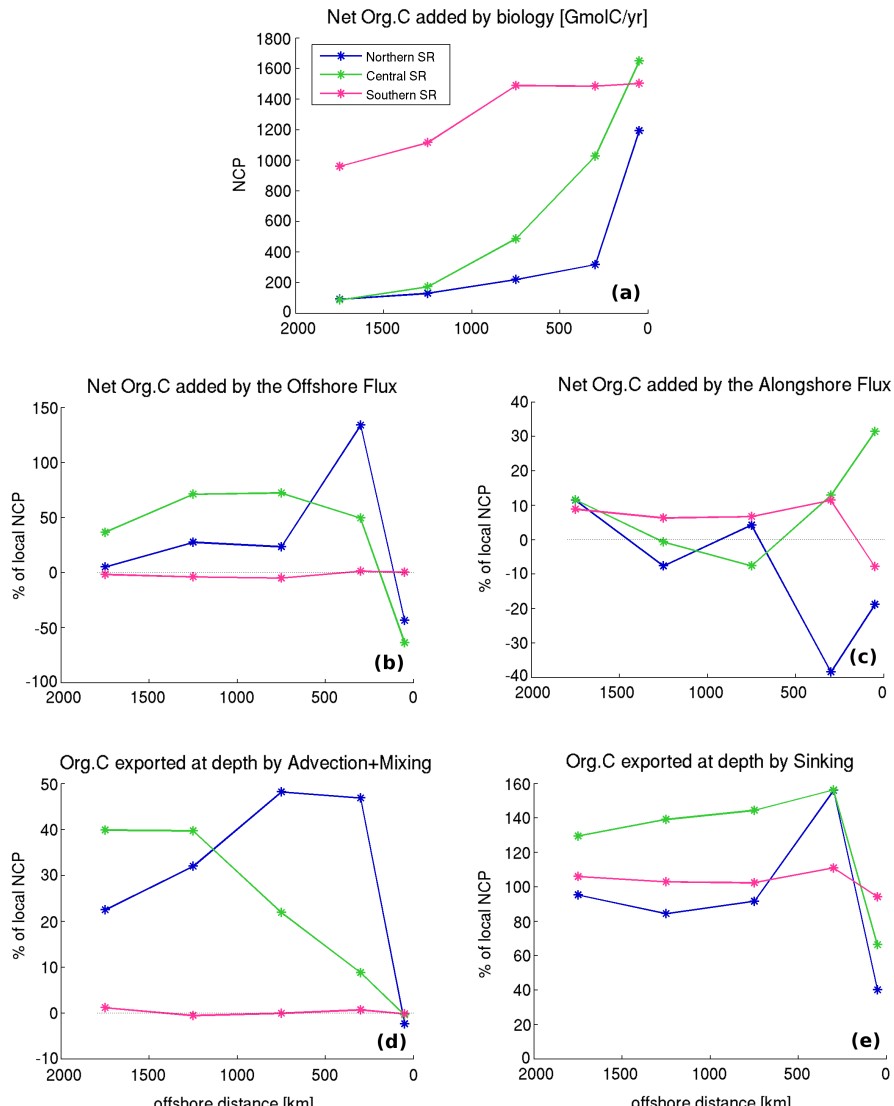

**Figure 12.** Trends of NCP and impact of the organic carbon fluxes by subregion and offshore distance in top 100 m as a function of offshore distance. (a) NCP [GmolC/yr]; (b) Divergence of the offshore transport in % of NCP; (c) Divergence of thealongshore transport in % of NCP; (d) Export below 100m by advection and mixing in % of NCP; (d) Export below 100 m by sinking in % of NCP. Fluxes in the plots are binned at the center of the box of reference (eg, fluxes referring to the 0 km-100 km box are binned to 50 km offshore on the x-axis).

and northward flowing Mauritanian Current that converge in this zonal band are the main contributors to this organic carbon influx on the coast around Cape Blanc. The carbon collected on the shelf is likely exported offshore together with the locally produced carbon by a very intense zonal flux characterized also by a large divergence that exceeds the values of the northern subregion in the offshore waters (Figure 12b). The carbon accumulation due to the divergence of the offshore transport over

the central zonal band is on average as high as 57 % of the local NCP, reaching peaks of more than 70 % of NCP in the 500 km-1500 km range and still accounting for 37 % of NCP in the farthest offshore box (1500 km-2000 km range). Both the mean circulation characterized by the westward flowing currents along the Cape Verde front and the mesoscale activity in the form of the giant Cape Blanc filament are expected to contribute to this intense offshore transport. As a consequence of this increased carbon availability, the central subregion allows very high values of the total vertical export of carbon below the euphotic layer (advective+mixing+sinking transport), which almost doubles the local production. Among these vertical components, the advective+mixing vertical export becomes particularly important in the offshore waters. The alongshore convergence of the organic carbon on the shelf and the high lateral mobility of the organic carbon in the offshore direction of the central subregion not only explain the peak of net water column heterotrophy in the offshore waters around Cape Blanc (see Figure 6a), but they characterize the Cape Verde frontal zone as a key region of the CanUS for the collection and export of the coastally produced organic carbon far into the North Atlantic Gyre.

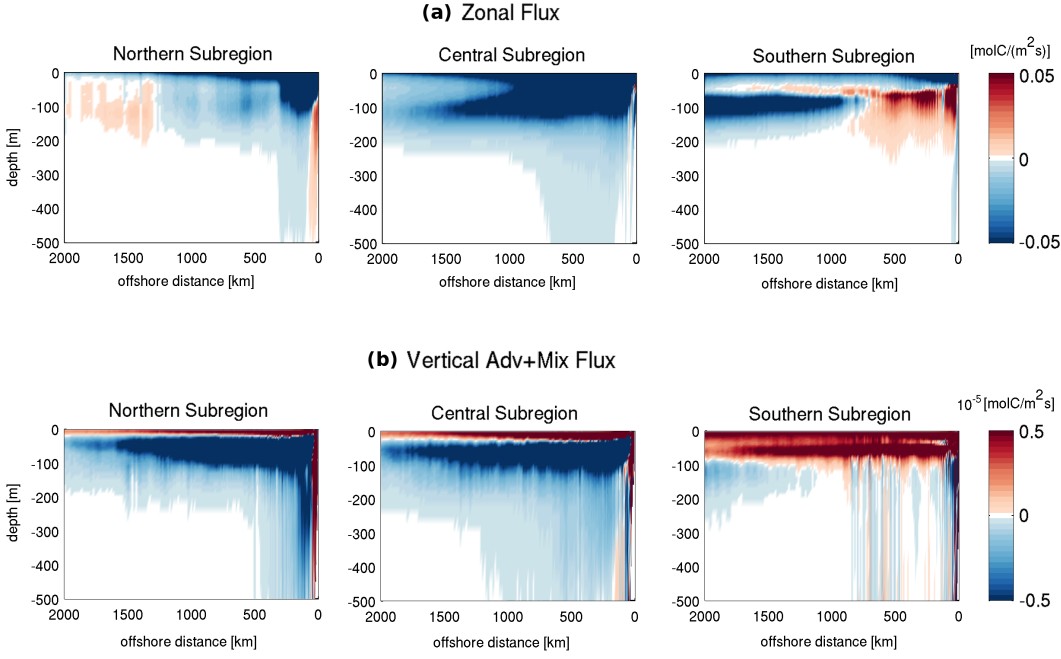

**Figure 13.** Vertical sections of the organic carbon offshore flux (a) and vertical advective+mixing flux (b), averaged meridionally along lines of equal distance from the coast in each subregion in accordance to the zonal bands defined by the budget analysis boxes. In all the plots the horizontal x-axis represents the distance from the coast (km); the vertical y-axis represents the depth (m). Subplot (a) Zonal flux: positive means eastward (onshore); Subplot (b) Vertical advective + mixing (eddy-diffusive) flux: positive means upward.

Further insights in the zonal differences of the organic carbon transport below the euphotic layer are given by the zonally averaged mean vertical profiles of the offshore and vertical advective+mixing fluxes for each subregion (Figure 13). The southern subregion is confirmed to be the least efficient in the offshore transport and vertical advective+mixing export also

at depth, despite an intensification of the offshore flux below the surface in the farthest region of analysis. This offshore intensification is however probably connected to the intersection of our southern zonal band with the Cape Verde frontal region which crosses the northern boundary of this subregion at about 1000 km offshore. The vertical advective+mixing export in the southern subregion is also remarkably different from the other subregions: not only the shallow MLD limits vertical mixing but the positive signature of the wind stress curl favors upwelling in this subregion. The mean vertical profiles of the zonal flux for the northern subregion shows clearly in the offshore waters below the surface a weak onshore flux that confirms the important influence of the incoming Azores current in the limitation of the offshore transport away from the coast. The deep extension of both the offshore flux and of the vertical advective+mixing downwelling in the first few kilometers from the coast of the northern subregion suggests again the link with the recurrent coastal filaments that characterize this sector of the CanUS and are known to enhance the fluxes through a depth of several hundreds of meters (Arístegui et al., 2009). The central subregion presents also at depth the most intense and persistent offshore flux that reaches 2000 km from the coast in the whole first 200 m layer, remarking the important role of this subregion for the offshore redistribution of the organic carbon. Both offshore and advective+mixing fluxes extend particularly deep into the water column in the first several hundreds of kilometers from the coast in the central subregion, likely due to both the intense mean circulation and the powerful giant Cape Blanc filament renown for its remarkable offshore extension (Gabric et al., 1993; Fischer et al., 2009). In both the northern and central subregions the offshore gradient of the zonal flux is the cause of the important accumulation of organic carbon that allows an enhanced respiration at depth.

The results of our subregional analysis show how physical forcing, mean and mesoscale circulation drive the lateral and vertical redistribution of the organic carbon in the CanUS, giving rise to a persistent offshore transport of organic carbon that shapes the $\int$NCP pattern and can reach as far as 2000 km into the North Atlantic Gyre. Further insights into the special role of mesoscale activity in the lateral redistribution of organic carbon in the CanUS and a quantification of this component of the transport will be provided in detail in a dedicated publication.

## 5 Discussion

### 5.1 Implications and comparison with previous work

Our results highlight the importance of the lateral transport of organic carbon from coastal regions of intense production to the oligotrophic open waters and its key role in fueling the offshore heterotrophic activity. We thus confirm the predictions of several in situ observations and estimates from multiple independent local surveys (Arístegui et al., 2009; Pelegrí et al., 2005). Our modeled fraction of the coastal production of organic carbon that is transported offshore beyond 100 km amounts to 18% of NPP (36 % of NCP) and lies in the range of previous estimates (Duarte and Cebrián, 1996). Our results also reveal that the offshore transport extends far offshore, having important consequences for the biological activity: the enhancement of the carbon availability due to the offshore influx can still be as large as 37% of the local net community production between 1500 km and 2000 km offshore. The offshore transport is particularly relevant below the euphotic layer, especially in the 100m-200m

layer, where the lateral transport of the organic carbon can extend even farther into the North Atlantic gyre, i.e., beyond 2000 km offshore.

This long-range offshore transport of organic matter involves two possible, not mutually exclusive pathways. In the first "direct" pathway, the offshore transport is sustained entirely by the excess organic carbon produced on the African shelf, which then gets advected offshore without the addition of "new", i.e., locally formed organic carbon along the way. In the second "recycling" pathway, the offshore flux of organic carbon is sustained by the production of organic carbon that happens along the way to the open sea: new and regenerated offshore production replaces the incoming organic carbon that gets remineralized and sinks to depth in a continuous recycling of the organic carbon that is advected offshore. The nutrients required to fuel the production stem from either from the upwelling along the coast and subsequent offshore transport and/or from the local remineralization of the incoming organic matter. The vertical pumping of the nutrients from the deeper waters offshore would be significant only in regions of positive wind stress curl (in our case only the southern CanUS) or abundant in mesoscale eddies. A simple analysis of the residence times of the modeled organic carbon pools as well as the fact that alongside the organic carbon, also substantial amounts of nutrients are transported offshore favors a dominance of the second "recycling" hypothesis. In fact, a small detritus particle resides, on average, only about 200 days in the top 200 m depth layer due to its sinking speed if we disregard downwelling and other loss terms such as remineralization; given typical lateral transport velocities of less than 0.05 m/s when averaged over the top 200 m, this particle would be able to travel at best only about 800km offshore. This distance is only 80 km for large detritus and 1600 km in the case of phytoplankton. If we further include the loss terms (coagulation and remineralization) of these organic carbon pools, which substantially reduce the life time, it is highly unlikely for the coastally produced organic carbon to reach as far as 2000 km in the open sea. Even though zooplankton is the only organic carbon species that does not sink in the model, its modest contribution to the offshore fluxes, compared to those of phytoplankton and of the abundant small detritus, cannot justify the magnitude of the observed lateral transport. Further, the inorganic nutrients fluxes (see Appendix B: Supplementary figures, Figure B7) are of sufficient magnitude to refuel new growth of organic matter to replace that part that is lost by sinking. Further analyses including Lagrangian experiments are necessary to gain a quantitative understanding of the succession of transformations happening along the way to the open waters.

The meridional alongshore transport also contributes to the organic carbon redistribution, especially on the shelf, where we find the maximum intensity of the coastal currents. In line with the results of Auger et al. (2016) and Pastor et al. (2013) we find that the area around Cape Blanc, corresponding to the region of convergence of the coastal flows and formation of the Cape Verde frontal zone, is a key region of the CanUS. The relevance of the central Cape Verde frontal zone in the CanUS was discussed in Auger et al. (2016) and Pastor et al. (2013) in terms of chlorophyll and nutrient convergence on the shelf and of their subsequent offshore advection, visible as a persistent bloom in offshore of the Cape Blanc region. Here we confirm and strengthen these results, affirming that this sector of the CanUS has a central role in collecting the organic carbon produced on the shelf and then transporting it offshore. This offshore flux of carbon against the gradient of productivity extends far away from the coast and feeds the heterotrophic activity of the deep open waters for at least 2000 km of kilometers offshore, generating a long tail of net heterotrophy. Deep offshore transport and subduction in this region are likely enhanced by the

persistent Cape Blanc filament, which is known from local surveys for being able to transport an estimated 50 % of the coastally produced carbon both in the surface and at depth, extending several hundreds of km offshore (Gabric et al., 1993; Ohde et al., 2015).

The partition of the CanUS into a northern anticyclonic and a southern cyclonic circulation regime and the differences in mesoscale activity and wind stress curl (Arístegui et al., 2009) are well reflected by the differences in the transport and cycling of the organic carbon north and south of the Cape Verde frontal zone. The portion of CanUS located north of the Cape Verde front can be regarded as the very eastern edge of the North Atlantic Gyre (Pelegrí et al., 2005). Several studies highlight the way in which the abundant coastal filaments of this northern CanUS sector substantially enhance the offshore transport and the downwelling of organic carbon in the first hundreds of kilometers from the coast (Álvarez-Salgado et al., 2007; Gabric et al., 1993; Fischer et al., 2009; García-Muñoz et al., 2005; Ohde et al., 2015). These structures together with a strong mean offshore flow explain the intense zonal transport and the vertical downwelling in the shelf region of the northern CanUS and the sharp decline of these two fluxes in the open waters. The range of influence of the offshore transport in this zonal band is further enhanced by the eddies spun off by the filaments (Barton et al., 2004), while the negative signature of the wind stress curl maintains the vertical downwelling in the offshore. The strength of the vertical transport by downwelling and mixing suggests that the northern subregion is potentially efficient in exporting to depth the dissolved, suspended and slowly settling material, i.e., the organic carbon species that are difficult if not impossible to measure with sediment traps, but may still constitute the key component for the closure of the organic carbon budget at depth (Hopkinson and Vallino, 2005; Alonso-González et al., 2010). The portion of CanUS located south of the Cape Verde front is instead mostly coupled to the southern equatorial circulation and to a much smaller extent to the Northern Atlantic gyre. Here, the net water column biological activity shows a dominantly neutral water column and little water column heterotrophy, the latter mostly confined to a region between the African coast and the Cape Verde archipelago. The intense near-surface production, the much smaller offshore gradient in productivity (NASA-OBPG, 2010) and to some extent the transitory nature of the filaments that form on the shelf at these latitudes (Arístegui et al., 2009) result in a small impact of the organic carbon lateral fluxes. Sinking dominates the vertical export at these latitudes while mixing and vertical advection are impeded by a shallow MLD and the positive wind stress curl leading to upwelling.

Overall, in a large part of the CanUS the lateral redistribution of organic carbon from the shelf to the open waters results in a very substantial lateral shift of the region of remineralization from the region of production. This is in contrast with the representation of the organic carbon pump as a pure vertical process and highlights the fundamental importance of the lateral transport of organic carbon for the maintenance of the biological activity. However, despite the very large lateral input of organic carbon in the upper 100 m across much of the offshore region of the CanUS, our model does not show evidence for net heterotrophic conditions in the near-surface waters of these regions. Thus the shallow open sea is everywhere a net source of organic carbon for the deeper layers. This is the case irrespective of whether the vertical integration is performed over the mean euphotic layer depth (100m), over the local euphotic layer depth or over the local MLD (generally shallower than the euphotic layer). Thus, our model provides strong support for the net autotrophic surface ocean hypothesis (Williams et al., 2013). The spatial pattern of modeled near-surface autotrophy (Figure 6b) agrees with the calculated global distribution of NCP (Williams et al., 2013, Figure 1) once the net heterotrophic regions are substituted by weakly autotrophic low productive

waters. The depth at which vertically integrated sinks and sources from top to bottom compensate each other in the model is almost everywhere located at more than 200m depth and can be deeper than 1000 m in regions of nearly neutral water column, confirming the importance of the respiration in deep waters (Del Giorgio and Duarte, 2002).

Both the mean Ekman transport and the turbulent mesoscale activity contribute to the total lateral fluxes of organic carbon

connecting coastal sources to deep offshore sinks. The two concur also in determining the vertical downwelling and mixing that increase the organic carbon transport to depth. The magnitude of the relative contribution of these two terms to the organic carbon fluxes and their different role in fueling the heterotrophic activity offshore must be detangled through further analysis.

## 5.2   Limitations and caveats

We discuss here how our quantification of the offshore transport of organic carbon at the surface and at depth may be affected

by a few shortcomings of the model. The first set of shortcomings involve our modeling of the organic matter pool, especially our lack of consideration of the an explicitly modeled DOC pool and the representation of POC at depth. The second set of shortcomings involve a few biases in our modeled physical/biogeochemical fields. We discuss the potential impact of these shortcomings in turn.

Regarding DOC, the pool that matters is that of semi-labile DOC as it has a life time of beyond a few days, implying that

it can be transported substantial distances before it gets remineralized. As a result, it has the potential to enhance our modeled lateral export of organic carbon. This is especially the case since DOC is readily produced in the surface ocean and contributes also substantially to the export of organic matter from the near-surface ocean (Hansell, 2002; Arístegui et al., 2002; Hansell et al., 2009; Hansell and Carlson, 2015), in particular in subtropical regions such as the North Atlantic gyre (Torres-Valdés et al., 2009; Roussenov et al., 2006). Even though DOC is not explicitly modeled, the small detritus, with its sinking speed of

$w_{SD}$=1 m day$^{-1}$, represents essentially a suspended POC pool with some similarity to a semi-refractory DOC, particularly regarding its susceptibility to lateral transport. But differing from DOC, the small detritus coagulates to large detritus resulting in a shorter lifetime than DOC in the surface ocean. At the same time, the rate of production of DOC is likely smaller than that of the small detritus, likely leading to a situation where the small detrital pool likely has a behavior that is rather close to that of DOC. Thus, we would argue that the impact of our shortcoming of not representing the dynamics of DOC explicitly

is smaller than possibly inferred at first sight. In order to explore more quantitatively the potential impacts of our lack of explicit consideration of DOC, we ran a sensitivity study, in which we set the vertical sinking of the small detritus, $w_{SD}$, to zero and reduced the coagulation time scale for small detritus to 40% of its baseline value. No adjustments were made to the parameterization of the large detritus. This sensitivity study needs to be considered as an extreme scenario - i.e., it is meant to explore the potential contribution of DOC rather than an attempt to quantify it in detail. We spun up the model with the

new biological parameters from year 24 of the baseline run (6 years of spinup) and used years 30-35 for the analysis, as for the baseline run. The results show, as expected, an intensification of the lateral fluxes of organic carbon in the euphotic layer. The standing stock of suspended POC increases about twofold, largely due to its longer average lifetime in the surface ocean, stimulating the local recycling of organic matter. This increases both primary production and heterotrophic activity in the near-surface layer, leaving the $\int$NCP pattern basically unchanged and preserving the net autotrophy of the near-surface waters. In

fact, even though the lateral transport of small detritus is much larger in this sensitivity study and reaching further out into the open North Atlantic, the net horizontal divergence of the lateral flux remains roughly the same. Thus, for the key question at hand, i.e., can the offshore transport fuel net heterotrophic conditions in the offshore regions of the Canary CS, the answer essentially remains the same.

5    Another potential caveat of our study regards the lateral redistribution of the organic carbon at depths larger than the first few hundred meters. On average, our modeled offshore transport below 200 m is very small, and never larger than 12 % of the total transport. However, model limitations in the representation of the offshore transport below this depth should be discussed taking into account three potential and partially contrasting caveats. First, the model does not include the process of sediment resuspension, therefore impeding the formation of high POC concentration spikes near the shelves (Inthorn et al., 10   2006a; Alonso-González et al., 2009) and limiting the bottom transport along the slopes (Inthorn et al., 2006b; Hwang et al., 2008). Second, and with a similar effect, the dynamics of our particulate pools only allows the aggregation of small particles into bigger and heavier ones, while it does not consider disaggregation of heavy particles into lighter ones as a consequence of degradation or partial grazing (Alonso-González et al., 2010), resulting in a one-way path to fast sinking that cannot be reversed. Due to these two factors that preclude the existence of deep local maxima of suspended POC, our study may underestimate 15   the lateral transport of organic carbon at depth. Third, sinking velocities in our model are fixed at every depth to moderate values (maximum of 10 m day$^{-1}$ for large detritus), while sinking velocities have been observed to be able to reach relatively high values, increasing by roughly an order of magnitude between the mesopelagic and bathypelagic regions (Fischer and Karakaş, 2009; Berelson, 2002) with a consequent fast vertical export at depth of the particles by sinking. Heavy particles at depths below 1000 m have been shown to have mean sinking velocities of 100-300 m day$^{-1}$ (Fischer and Karakaş, 2009) and 20   to be often accompanied by a pool of slow sinking material with mean sinking velocities of 1 to a few m day$^{-1}$, resulting in a bimodal distribution of the sinking speeds (Alonso-González et al., 2010). Viewing these three caveats together, we have two missing processes that would cause our model to represent a lower bound estimate, and one process that would cause the correct offshore transport to be smaller. We cannot assess the implication of this finding in full, but submit that at least with regard to the offshore transport in the upper waters, i.e., upper 200 m, our model is likely in the right range, perhaps on the 25   lower side. We have much less confidence in the offshore transport below 200 m, where the observed concentrations of organic carbon can be quite high, although the offshore velocities are substantially smaller.

We also need to assess the potential impact of the physical/biogeochemical biases that we diagnosed in the Evaluation section. In the northern CanS our model overestimates the MLD depth; however our modeled MLD shows a meridional gradient that has the same trend as the observed one, with an extremely shallow mixed layer in the southern region below 30   the Cape Verde front and deeper mixed layer in the north. This suggests that, even though we may potentially overestimate vertical mixing in the northern CanUS, this subregion would still be expected to be the only one in which this process is relevant. In the southern CanUS, our model shows a weaker than observed circulation and a deeper than observed chlorophyll and NPP maximum, which may lead to an underestimation of the lateral transport and therefore of the net heterotrophy of the water column. Both a shoaling of the biological production towards the surface characterized by more intense currents and an 35   intensification of the circulation can in fact result in the strengthening of the lateral zonal and meridional organic carbon fluxes.

However, an increase of the offshore zonal fluxes in the southern subregion could favor a more heterotrophic water column only if accompanied by an increase of the divergence of the flux, resulting in a substantial accumulation of organic carbon compared to the local production. In the meridional direction, an intensification of the alongshore Mauritanian current may instead increase the influx of organic carbon from the south into the Cape Verde frontal zone, fueling even further the deep

respiration in the already strongly heterotrophic central CanUS.

To summarize, we believe that the above-discussed caveats do not substantially affect our main findings. If anything they possibly strengthen our conclusion regarding the importance and long-range nature of the offshore transport of organic carbon. In fact our model may, if anything, underestimate the total lateral transport of organic carbon both at the surface and at depth. For this reason we believe that it is of fundamental importance to take into account the three-dimensionality of the marine

organic carbon cycle and the essential role of the productive coastal ocean in the global biogeochemical cycles.

## 6   Conclusions

This paper provides a first comprehensive quantification of the lateral and vertical fluxes of organic carbon in the Canary Upwelling System (CanUS) up to 2000 km offshore.

The long-range lateral fluxes of organic carbon in the euphotic layer (0 m-100 m depth) of the CanUS are dominated by the

offshore flux that extends on average as far as 1500 km into the North Atlantic Gyre. Along its way, the offshore flux adds to the euphotic layer an amount of organic carbon that corresponds from 8 % to 34 % of the alongshore average NCP, explaining from 62 % to 80 % of the excess vertical export, i.e., the export below the euphotic layer that exceeds the local production, and fueling extra heterotrophic activity at depth. In the 100 m-200 m layer the offshore transport of organic carbon continues to dominate the lateral fluxes, transporting always >8 % of the available organic carbon and intensifies away from the coast with

potential repercussions on the biological activity of the North Atlantic Gyre interior.

This massive redistribution of organic carbon from the nearshore to the offshore makes the vertically-integrated net community production, i.e., $\int$NCP, strongly positive in the nearshore regions and strongly negative in the offshore. This implies, when viewed over the whole water column, that the nearshore regions are net autotrophic and the offshore regions are net heterotrophic. However, the upper ocean (down to more than 100 m) acts everywhere as an organic carbon source, i.e., remains

autotrophic. Thus, our model demonstrates how critical it is to consider the depth interval over which the trophic state of a system is evaluated.

Strong subregional differences in the fluxes characterize the CanUS. North of the Cape Verde frontal zone, coastal production quickly declines offshore, while strong offshore transport by filaments fuels strong remineralization in the offshore regions, causing strong heterotrophic conditions downward of 200m. Mixing and vertical downwelling play an important role in the

vertical export in this subregion, enhancing the export of small detrital material below the euphotic layer.

South of the Cape Verde frontal zone high levels of near-surface production extend far offshore. Despite being the most productive area, the southern subregion has a much more modest impact on the offshore flux and also on the offshore rates of

remineralization. This results in a water column that has almost a neutral trophic state ($\int$NCP$\sim$0). Vertical export at depth in this subregion is driven mostly by sinking fluxes due to the shallow MLD and the positive wind stress curl signature.

The central zonal band of the CanUS, which includes the Cape Verde frontal zone bridging the northern and southern subregions, is characterized by an alongshore convergence of organic carbon on the shelf. The accumulated organic carbon is laterally exported from the shelf by an intense offshore flux that along the way releases on average as much organic carbon as 57 % of the local NCP, fueling the most intense peak of water column heterotrophy of the entire CanUS. The offshore transport is pronounced also at depth especially in the first 500 km from the coast, while advective and mixing fluxes have an important role in the vertical export in this subregion. Both the intense offshore transport and downwelling of organic carbon may be enhanced by the very large and persistent Cape Blanc filament.

Our study highlights the strength of the coupling between the productive CanUS region and the adjacent oligotrophic open North Atlantic. Lateral fluxes, especially the offshore transport, are influenced by mean circulation, mesoscale activity and physical forcings and have an essential role in the fueling of the heterotrophic activity in the open seas. Their impact on the local carbon availability fully explains the complex pattern of net sources and sinks of organic carbon of the CanUS region.

## Appendix A: Datasets

**a) Forcing datasets**

| Data source | Ref. time | Resolution | Variables | Reference |
|---|---|---|---|---|
| Era-Interim | 1979-2010 | N128 reduced Gaussian grid | freshwater flux, wind stress, net heat flux, net shortwave radiation | Dee et al. (2011) |
| GLOBALVIEW 2011 | 1998-2011 | 1°x 1°grid | atmospheric $pCO_2$ | GLOBALVIEW-$CO_2$ (2011) |

**b) Datasets used for corrections to the forcing datasets**

| Data source | Ref. time | Resolution | Variables | Reference |
|---|---|---|---|---|
| DFS 5.2 | 1979-2011 | 0.7°x 0.7°grid | downward longwave radiation, downward shortwave radiation | Brodeau et al. (2010) |
| Era-Interim | 1979-2011 | 0.75°x 0.75°grid | downward longwave radiation, downward shortwave radiation | Dee et al. (2011) |
| Era-Interim | 1989-2009 | 1.5°x 1.5°grid | sea-ice fraction | Dee et al. (2011) |
| NSIDC Sea Ice Motion Vectors | 1979-2006 | 25 km EASE-Grid | sea-ice drift | Fowler (2003) |

**Table A1.** Description of the datasets used for (a) the model run main forcing, (b) datasets used for calculating stratus cloud an sea ice corrections to the main forcing. DFS: Drakkar Frocing Set; NSIDC: National Snow and Ice Data Center.

**Boundary conditions**

| Data source | Ref. time | Resolution | Variables | Reference |
|---|---|---|---|---|
| WOA 2013 | 1955-2012 | 0.25°x 0.25°grid | temperature, salinity, nitrate | Locarnini et al. (2013), Zweng et al. (2013) |
| SODA v1.4.2 | 1958-2001 | 0.5°x 0.5°grid | momentum components, sea surface height | Carton and Giese (2008) |
| SeaWiFS | 1997-2010 | 9km grid | sea surface chlorophyll | NASA-OBPG (2010) |
| GLODAP | - | 1°x 1°grid | sea surface alkalinity, sea surface dissolved inorganic carbon | Key et al. (2004) |

**Table A2.** Description of the datasets used for the model run lateral boundary conditions. WOA: World Ocean Atlas; SODA: Simple Ocean Data Assimilation; SeaWiFS: Sea-viewing Wide Field-of-view Sensor; GLODAP: GLobal Ocean Data Analysis Project.

**Model evaluation**

| Data source | Ref. time | Resolution | Variables | Reference |
|---|---|---|---|---|
| Aviso CMDT Rio05 | 1993-1999 | 0.5°x 0.5°grid | sea surface height | Rio and Hernandez (2004) |
| AVHRR | 1981-2014 | 0.25°x 0.25°grid | sea surface temperature | Reynolds et al. (2007) |
| CARS | 1955-2003 | 0.5°x 0.5°grid | sea surface salinity | Ridgway et al. (2002) |
| Argo DT-0.2 | 1941-2008 | 2°x 2°grid | mixed layer depth | Montégut et al. (2004) |
| Drifters | 1979-2012 | 0.5°x 0.5°grid | sea surface height | Lumpkin and Johnson (2013) |
| SeaWiFS | 1997-2010 | 9km grid | sea surface chlorophyll | NASA-OBPG (2010) |
| SeaWiFS VGPM | 1997-2010 | 9km grid | extrapolated net primary production (NPP) | Behrenfeld and Falkowski (1997) |
| SeaWiFS CbPM | 1997-2010 | 9km grid | extrapolated net primary production (NPP) | Westberry et al. (2008) |
| SeaWiFS POC | 1997-2010 | 9km grid | extrapolated surface particulate organic carbon (POC) | NASA-OB.DAAC (2010) |
| Modis-Aqua VGPM | 2002-2016 | 9km grid | extrapolated net primary production | Behrenfeld and Falkowski (1997) |
| WOD09 | - | rebinned to 0.5°x 0.5°grid | chlorophyll | Johnson et al. (2009) |
| AMT | (2004-2014) | in-situ [0m,200m] depth | particulate organic carbon | BODC-NERC (2014) |
| Geotraces | (2010) | in-situ surface | particulate organic carbon | GEOTRACES (2010) |
| ANT | (2005) | in-situ [0m,200m] depth | particulate organic carbon | ANT (2005) |

**Table A3.** Description of the datasets used for the model evaluation. CMDT: Combined Mean Dynamic Topography; AVHRR: Advanced Very High Resolution Radiometer; CARS: CSIRO Atlas of Regional Seas; SeaWiFS: Sea-viewing Wide Field-of-view Sensor; WOD09: World Ocean Database 2009.

# Appendix B:  Supplementary figures

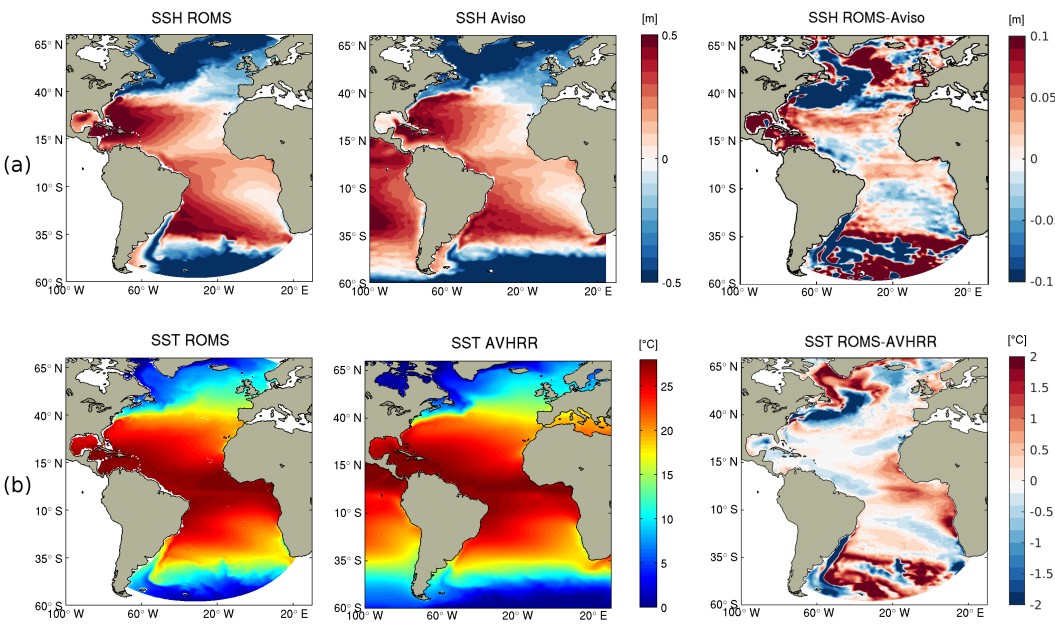

**Figure B1.** Mean Sea Surface Height SSH (a) and Sea Surface Temperature SST (b) from model and observational data, accompanied by a model-data difference plot in the full Atlantic Telescopic Grid domain. A detailed description of the data used for the evaluation is provided in Appendix A: Datasets, Table A3.

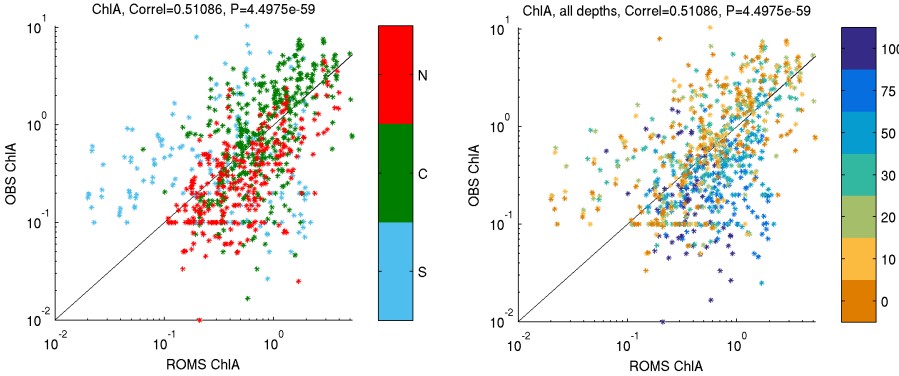

**Figure B2.** Evaluation of the modeled annual mean Chlorophyll (CHL) by subregion and by depth for the first 500km offshore as defined by the first two budget analysis boxes, see Figure 4. The spread of the dots is maximum for the southern subregion, in which modeled CHL is too low at small depths and too high at large depths. Observational dataset: WOD09, annual mean CHL. A detailed dscription of the used data is provided in Appendix A: Datasets, Table A3.

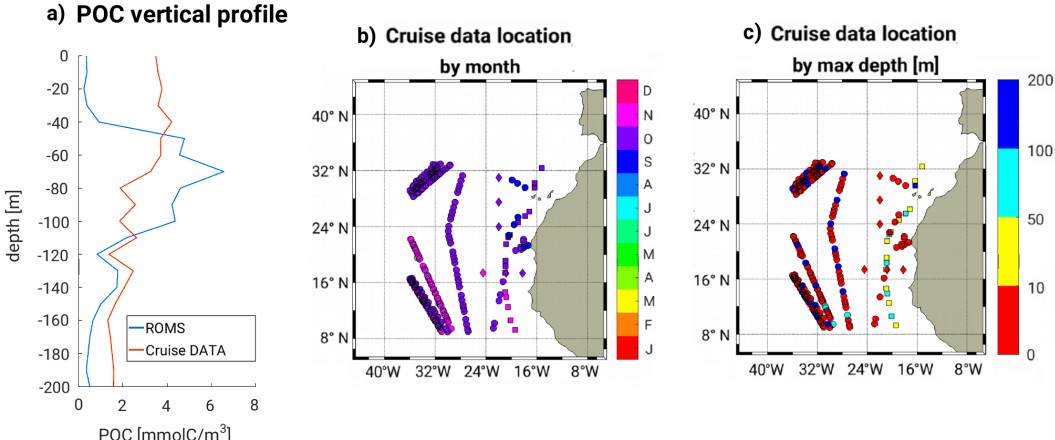

**Figure B3.** Panel a) Comparison of the modeled mean POC profile with the measured POC mean profile through co-location in space and time of modeled and cruise data POC. Data were re-binned in depth to 10m depth intervals. We used data contained in the first 2000 km from the coast as defined by the Budget Analysis boxes, see Figure 4. Panel b) Location of the cruise data colored by sampling month. Panel c) Location of the cruise data colored by max depth of the samples. A detailed description of the used data is provided in Appendix A: Datasets, Table A3.

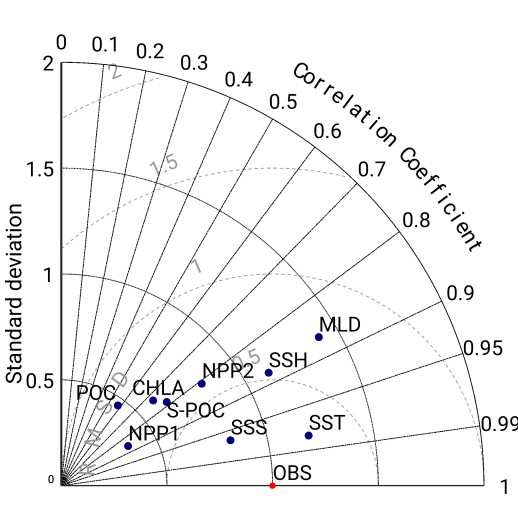

**Figure B4.** Taylor diagrams for the Canary EBUS region of analysis ([9.5N,32.5N]x[5W,35W]), climatological annual mean fields. Used datasets - Sea Surface Temperature (SST): AVHRR, Sea Surface Salinity (SSS): CARS, Sea Surface Height (SSH): Aviso CMDT Rio05, Mixed Layer Depth (MLD): Argo DT-0.2, Chlorophyll (CHLA): SeaWiFS, Net Primary Production dataset 1 (NPP1): SeaWiFS VGPM, Net Primary Production dataset 2 (NPP2): SeaWiFS CbPM, Surface Particulate Organic Carbon (S-POC): SeaWiFS POC, Particulate Organic Carbon (POC): cruise POC data (AMT, ANT, Geotraces). A detailed description of the data used for the evaluation is provided in Appendix A: Datasets, Table A3.

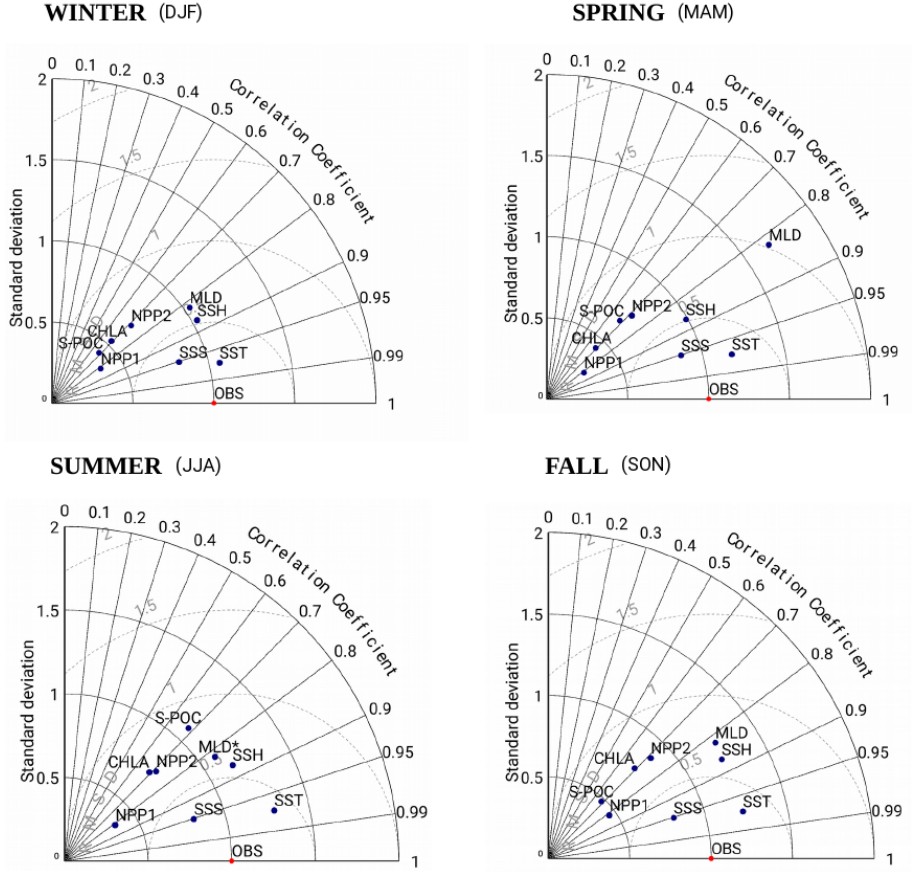

**Figure B5.** Taylor diagrams for the Canary EBUS region of analysis ([9.5N,32.5N]x[5W,35W]), climatological seasonal mean fields. In the Summer diagram MLD was rescaled to MLD*=MLD/2, the summer $MLD_{STD}$ is therefore 2 times as big as the plotted value, while the correlation remains unchanged. Used datasets - Sea Surface Temperature (SST): AVHRR, Sea Surface Salinity (SSS): CARS, Sea Surface Height (SSH): Aviso CMDT Rio05, Mixed Layer Depth (MLD): Argo DT-0.2, Chlorophyll (CHLA): SeaWiFS, Net Primary Production dataset 1 (NPP1): SeaWiFS VGPM, Net Primary Production dataset 2 (NPP2): SeaWiFS CbPM, Surface Particulate Organic Carbon (S-POC): SeaWiFS POC. A detailed description of the data used for the evaluation is provided in Appendix A: Datasets, Table A3.

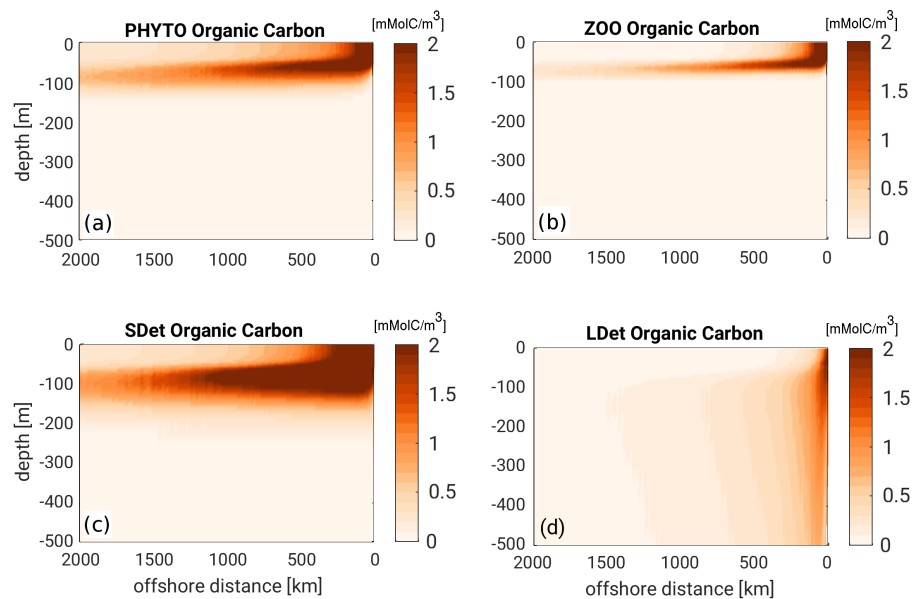

**Figure B6.** Mean vertical sections of the concentration of the modeled organic carbon (POC) components in the Canary EBUS, averaged meridionally along lines of equal distance from the coast between 9.5°N and 32°N. (a) Phytoplankton (PHYTO); (b) Zooplankton (ZOO); (c) Small detritus (SDet); (d) Large detritus (LDet).

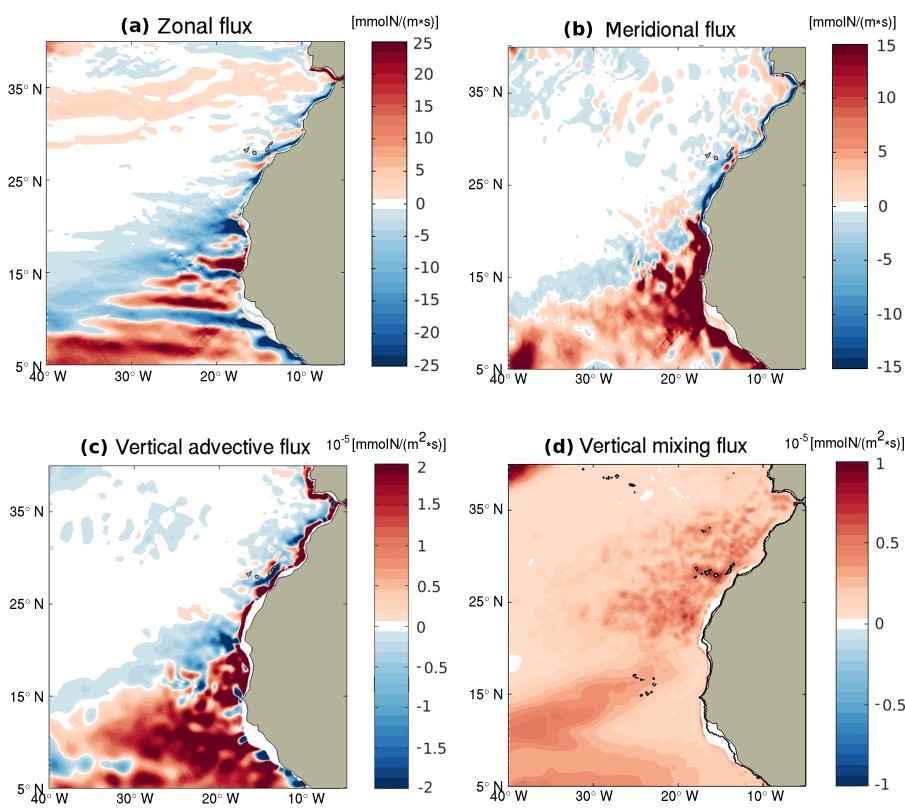

**Figure B7.** Maps of the inorganic nutrients (total inorganic nitrogen = nitrate + ammonia) flux components in the top 100 m corresponding to the euphotic layer in the CanUS. (a) Zonal flux vertically integrated over the top 100 m with positive values indicating eastward (onshore) transport; (b) as (a), but for the meridional flux with positive values indicating northward transport; (c) vertical advective flux across 100 m with positive values indicating upward transport; (d) as (c) but for vertical mixing. The plotted vertical component was smoothed with a 7x7 grid points 2-dimensional filter.

*Author contributions.* N.G., Z.L. and E.L. conceived the study. E.L. and M.M. set up the experiment and improved the model. E.L. performed the analysis. E.L. and N.G. wrote the manuscript. All authors contributed to the interpretation of the results and to the manuscript. N.G. and M.M. supervised this study.

"The authors declare that they have no conflict of interest."

5   *Acknowledgements.* We would like to thank Martin Frischknecht for his relevant comments on the work and during the preparation of the manuscript, Cara Nissen and Meike Vogt for their valuable feedback and Damian Loher for the technical support. We thank Referee nr.1 and Dr. Josep Pelegrí for their thoughtful review and their valuable comments that have helped us to improve this manuscript. We also thank the group of the Faculty of Marine Sciences at the University of Las Palmas de Gran Canaria in particular Javier Arístegui for allowing a fruitful exchange of ideas and information and Bàrbara Barceló for her kind support. A special thought goes to the late professor Pablo Sangrà

10   whose generosity and dedication to science will always be a source of inspiration. This research was financially supported by the Swiss Federal Institute of Technology Zürich (ETH Zürich) and the Swiss National Science Foundation (Project CALNEX, grant No.149384). The simulations were performed at the HPC cluster of ETH Zürich, Euler, which is located in the Swiss Supercomputing Center (CSCS) in Lugano and operated by ETH ITS Scientific IT Services in Zürich. Model output is available upon request. Please contact the corresponding author, Elisa Lovecchio (elisa.lovecchio@usys.ethz.ch), in that matter.

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
