# Peer review of "On the long-range offshore transport of organic carbon from the Canary Upwelling System to the open North Atlantic"

_Biogeosciences, 2016_

## Referee Comment (RC1) · Anonymous Referee #1 · 15 Feb 2017

Review of the manuscript entitled "On the long-range offshore transport of organic carbon from the Canary Upwelling System to the open North Atlantic" by Lovecchio et al.

This study focuses on the offshore transport of organic carbon from the very productive regions of the Canary Upwelling System (Can US) to the central regions of the Atlantic Ocean. This study is based on a model configuration of the whole Atlantic Ocean using a telescopic grid which allows to have an eddy-resolving scale over the CanUS region. The authors use a simple NPZD model with two sinking POC pools (a slow sinking and a fast sinking pool). The model does not represent dissolved organic carbon. The model predicts an intense offshore transport of organic carbon which extends

beyond 1500 km into the interior of the subtropical Atlantic Ocean. Nevertheless, the upper ocean remains autotrophic all over the domain. When integrated, NCP is net heterotrophic everywhere except in a very narrow coastal band. The central region near Cape Blanc is very efficient at collecting organic carbon and at transporting it offshore. This explains the large net heterotrophic characteristics of the region, which extends far offshore into the interior of the subtropical Atlantic Ocean (even further offshore than 2000 km).

This study is really interesting and clearly demonstrates the critical role organic matter plays in the lateral transport of carbon and nutrients from productive coastal upwelling systems to offshore more oligotrophic areas. Furthermore, this study emphasizes the regional differences in the CanUS area with the central part being an efficient collector of organic matter which is exported very far into the subtropical Atlantic Ocean. I really appreciated the detailed budget as well as the quite insightful diagnostics. So, I definitely think that this study deserves publication in Biogeosciences. However, I have two major concerns that would need to be addressed, or at least discussed.

The NPZD model is very simple which is not a problem by itself. It performs enough well to be suitable for that study. This is quite clearly shown in the validation section of the paper. However, according to me, it lacks a critical reservoir especially concerning the objectives of that study: DOC or more precisely semi-labile and semi-refractory DOC. Concentrations of semi-labile DOC range from typically 20 to 40 umol/L in the upper ocean (Hansell et al., 2009; Hansell and Carlson, 2014). Its lifetime is also quite long and ranges from weeks to years which makes it possible for that pool to be transported far away from its production region. It has been shown to potentially play an important role in the subtropical gyres (e.g., Roussenov et al., 2006; Torres-Valdés et al., 2009). In the present study, this pool is omitted and thus, a potentially large contribution to the lateral export of organic carbon is not represented. This needs at least to be discussed in the discussion section.

In this study, the importance of mesoscale features is emphasized several times but

never clearly quantified. It would have been nice to have such a quantification. I would suggest two possible means to do that: 1) to perform a classical separation technique between the mean and eddy components of the transport; 2) to perform a simulation in which the non linear terms in the Navier-Stokes equations for momentum iare cancelled such as in Gruber et al. (2011). Otherwise, any discussion of the effect of the mesoscale circulation remains quite speculative and qualitative.

I have some more specific comments which are listed below:

Page 2, line 1 - "resuspension of bottom sediments and can create ... " I guess something is missing in this sentence.

Page 4, line 33 - In the list of state variables that are listed, you should add O2.

Page 5, lines 11-12 - Phytoplankton can coagulate with small POC to form large POC. Is it also the case for small POC with small POC?

Page 7, lines 28-33 - You should refer to figure 3 to illustrate the different regions.

Page 10, lines 8-13 - Almost everywhere, except near Cape Blanc, high values of Chlorophyll are too narrow and too much trapped near the coast. As mentionned by the authors, this bias is especially strong in the Southern part of the CanUS domain.

Page 11, lines 1-6 - The authors here discuss the characteristics of the modeled subsurface maximum of Chl (DCM) and they refer to Figure B2. This is not always easy to see from Figure B2. The most obvious bias that emerges from the figure is the too high values of Chl at depth below 50m. Otherwise, it is hard to quantify from that plot the depth of ithe DCM in the model and in the data.

Page 17, lines 3-6 - For sure in the interior of the ocean, the contribution of small POC to the vertical sinking flux of organic matter should drop very quickly with depth. A figure showing the contribution of the different pools of organic matter to total organic carbon would be nice.

Page 18, Figure 9 - The fluxes in the different boxes are not balanced (the imbalance is however small). Is it because the model is not fully at steady state or because of the internal variability related to the mesoscale activity?

Page 18, lines 1-6 - The DeltaE diagnostics is interesting. It accounts for two processes that can increase the export without changing the NCP: 1) The organic matter that is being transported laterally and that sinks out of the upper ocean increases the export and thus DeltaE. 2) The organic matter that is being transported laterally and that remineralizes in the upper box. This stimulates the biological activity which produces more organic matter which is sinks out of the upper ocean. In that case, NCP is not changed (the increase in PP compensates for the remineralization of the laterally supplied organic carbon) and export is increased which increases DeltaE. This two mechanisms should be explained here, especially because in the discussion section it is shown that the second process dominates.

Page 20, line 22 "and quantify the contribute of the different zonal bands ..." I guess it should be contribution

Page 22, line 25 "and quickly channell water ..." It should be channel.

Page 24, line 8 "becomes particularly important the offshore waters" Some words are missing here.

Page 25, lines 18-20 - The splitting between the contribution of the mean flow and of the eddy transport is not really clear here. See my second major concern above.

---

## Referee Comment (RC2) · J. L. Pelegrí (Referee) · 26 Feb 2017

This study uses a coupled physical-biogeochemical numerical model to investigate the export of organic carbon off NW Africa, from the coastal upwelling band into the deep ocean. The numerical data is used to carefully assess sources and sinks of organic carbon in the coastal and offshore regions, including the horizontal and vertical fluxes as well as primary production and remineralization. The exhaustive data analysis provides insight into the main mechanisms that control the sign of net community production (NCP) in a very extensive region off NW Africa. This manuscript represents a comprehensive study that should eventually become a work of reference for future research on coastal upwelling systems.

[Figure]

In my opinion, however, there are a number of important issues data need to be either solved or clarified before the paper may be accepted for publication. These relate to the following: (1) evaluation of the model's performance, (2) latitudinal partition of the domain, and (3) the role played by the supply of subsurface inorganic nutrients in the coastal and offshore upwelling regions. The final paper will also benefit from both better revision of literature, particularly on the circulation patterns in the Canary upwelling system (CanUS), and more concise and less speculative writing. I have no doubts that the authors can address these points, I encourage them to carry out the additional effort.

I will next explain in more detail the above major issues and later will comment on some other minor points that also require attention.

**Major issues**

(1) Evaluation of the model's performance

This is a very critical aspect and the authors dedicate a significantly long section, including an appendix, to evaluate the performance of the model. They compare the numerical output with observations using different datasets: the near-surface seasonal circulation as inferred from surface drifters; the annual-mean sea surface height, sea surface temperature (SST) and sea surface salinity; the annual-mean mixed layer depth (MLD); the annual-mean surface chlorophyll; and the annual-mean and seasonal-mean net primary production (NPP).

I value this effort very much but, honestly, at the end of the Evaluation section I have important doubts on how good the model's performance is. Throughout this section the authors recognize the existence of substantial differences between model and field data, and also talk about model bias. In Figure 2 they show the spatial distribution of model-data differences for several surface fields. The differences are not negligible at all, as clearly seen by the range of values in the mean fields and the differences, e.g. SST (range of values is 12°C and range in deviations is 4°C) and MLD (range of values
is 100 m and range in deviations is 60 m). I particularly miss a comparison between the depth distribution of the modelled and observed particulate organic carbon (POC), which is of capital importance for this study. The seasonal results (Figure 5) show very large differences, possibly too large.

The authors end this section referring to a Taylor diagram presented in Appendix B (Figure B3), concluding that there is a "good correlation between the modelled and observed fields both in the annual and in the seasonal means." They show the Taylor diagram for the annual-mean results and for the mean of the seasonal results. The authors argue that the Taylor diagram shows results comparable to other studies for upwelling systems. Rather than comparing with other studies, it would be better to look at the statistics and discuss whether the results are convincing or not. For the annual-mean, for example, SST, CHLA and MLD respectively have a (normalized) standard deviation of about 1.2, 0.6 and 1.4, and a (normalized) root-mean-square difference of 0.3, 0.7 and 0.8. The authors should discuss whether these values are reasonable or not. I am particularly confused by Figure B3b: how is this calculated, just an average mean? What is the meaning? Wouldn't it be much better to show all four seasonal diagrams? It would also help to include, as supplementary materials, diagrams for each subregion.

(2) Latitudinal partition of the domain

In several places of the Introduction and Discussion the authors recognize that the Cape Verde frontal zone is a natural boundary between the subtropical and tropical domains. Nevertheless, for most of their analysis on latitudinal variability they use a partition in three areas or subregions, as shown in Figure 3b, which is not properly justified. I imagine this is done as an attempt to grasp the character of the meridionally convergent region near Cape Blanc but, as it is clear from the velocity fields in Figures 3 and 5, this is not correct. In my opinion only the southern subregion would comprise an area with approximately coherent dynamics.

My suggestion here is to use four subregions of different size: the northern one (25-32°N) would correspond to an area with substantial mesoscalar activity, with eddies and filaments generated both south of the Canary Archipelago and at the upwelling front; the second area would represent the permanent and intense central upwelling region (21-25°N); the third are would concentrate on the convergent region immediately south of Cape Blanc, which is the root of the Cape Blanc giant filament (about 17-21°N, though these limits change with longitude); the southern area (9.5-17°N) would correspond to the tropical region. Right now most of the discussion is either on the results for the latitudinal-average picture or (to a lesser degree) for the three proposed regularly-spaced subregions. With this alternative partition, the paper would certainly become much more informative.

I value very much the authors' efforts to provide bulk figures for the entire region but I think that plotting these results may be very misleading. For example, the data in Figure 8 suggests that the zonal flux of organic carbon is more intense than meridional one. I doubt this very much: in my opinion this is only an artefact that the latitudinal average tends to cancel the contributions of the southward Canary Upwelling Current and northward Mauritania Current and Poleward Undercurrent (please see references below regarding the main currents in the CanUS). My suggestion is to produce fewer plots on the results for the entire region (Figure 9 is fine but some other plots may be replaced by tables) and instead show what is happening in each area: the CanUS is so large that it surely deserves a closer view for each subregion.

(3) Upwelling of coastal and offshore inorganic nutrients

The coastal upwelling region is a source of inorganic nutrients to the surface layers in the coastal transition zone that are later exported offshore (e.g. Pelegrí et al., 2006; Pastor et al., 2008, 2013). Such a flux of inorganic nutrients is a prime element in the offshore net primary production and the sign of the NCP north of Cape Blanc. However, this issue is not mentioned in the manuscript until the Discussion. The subject is important enough to deserve careful attention when examining the sources and sinks for

NPP, it is the difference between new production using the subsurface load of inorganic nutrients or production after remineralization.

The offshore waters in the southern subregion are also largely affected by the presence of upward Ekman pumping, i.e. offshore upwelling resulting from positive wind-stress curl. Again this is an important aspect in the dynamics and NPP balance of this subregion, which is again acknowledged very late in the manuscript and only partly discussed.

The model could be used to assess these different contributions. Perhaps this was not the objective of the authors, which is fine, but then the potential relevance of the upwelling and transport of inorganic nutrients on the NPP and NCP within the entire region should be properly discussed since early in the manuscript.

**Minor points**

(4) p 1, l 10: divergence or convergence?

(5) p 2, l 11: replace "and can create" by "can create."

(6) p 2, l 15: "Arístegui."

(7) p 3, l 5: other relevant references are Pelegrí et al. (2006) and Pastor et al. (2013).

(8) p 3, l 24: also Pastor et al. (2013).

(9) section on Methods: how does the model calculate the vertical velocity?

(10) caption of Figure 1: Gran Canaria is cited in the caption but not located in the map.

(11) p 10, l 10-13: please clarify.

(12) caption of Figure 4: VGPM is first mentioned here but it is defined nowhere in the manuscript.

(13) p 12, l 21-24: asides the Canary Current and the Mauritanian Current you should also probably refer to the Canary Upwelling Current (associated to the coastal upwelling jet) and the Poleward Undercurrent (please see references below).

(14) p 12, l 31-32: ". . . NPP is a better measure than chlorophyll for evaluating. . ."

(15) p 13, l 1: Pastor et al. (2013) is probably a better reference.

(16) caption Figure 6: panel b also includes sediment remineralization?

(17) p 18, l 3-5: are you using two different definitions for excess export?

(18) p 20, l 1-2: here and elsewhere it is best to not refer to lines, they should be defined in the figure's caption or legend (otherwise you would have to define them everywhere).

(19) p 20, l 27 and 33: "north of the Cape Verde front. . ."

(20) p 22, l 2: ". . .south of Cape Blanc."

(21) please revise caption of Figure 12.

(22) Figure 13: I suggest that you also show the meridional fluxes.

(23) caption Figure 13: "vertical" rather than "vertcal."

(24) p 24, l 14: "(Figure 13)."

(25) p 25, l 6-8: this is likely an artefact of the SW-NE orientation of the coast.

(26) p 25, l 10 and 15: please include references.

(27) p 27, l 8: see also Pastor et al. (2013).

(28) p 28, l 7: "these."

(29) p 28, l 10-18: usage of so many conditionals raises doubts on the reader.

(30) p 28, l 27: remove "from Section 4.1."

(31) p 28, l 28: is this the right way to cite a figure within a reference?

(32) p 29, l 3: here and elsewhere separate numbers from units, i.e. "2000 km" rather

than "2000km."

(33) p 29, l 16-17: please revise writing.

(34) additional references: asides those mentioned above, there are other works that would help better describe the circulation patterns in the CanUS, such as Mason et al. (2011), Peña-Izquierdo et al. (2012, 2015), Pelegrí and Peña-Izquierdo (2015), Pelegrí and Benazzouz (2015).

**References**

Mason, E., Colas, F., Molemaker, J., Shchepetkin, A. F., Troupin, C., McWilliams, J. C., and Sangrà, P.: Seasonal variability of the Canary Current: a numerical study. Journal of Geophysical Research, 116, C06001, 2011.

Pastor, M. V., Pelegrí, J. L., Hernández‐Guerra, A., Font, J., Salat, J., and Emelianov, M.: Water and nutrient fluxes off Northwest Africa. Continental Shelf Research, 28, 915‐936, 2008.

Pastor, M. V., Palter, J. B., Pelegrí, J. L., and Dunne J. P.: Physical drivers of interannual chlorophyll variability in the eastern subtropical North Atlantic. Journal of Geophysical Research: Oceans, 118, 3871‐3886, 2013.

Pelegrí, J. L. and Benazzouz, A.: Coastal upwelling off North‐West Africa. In: Valdés, L. and Déniz‐González, I. (eds). Oceanographic and biological features in the Canary Current Large Marine Ecosystem. IOC‐UNESCO, Paris. IOC Technical Series, 115, 93‐103, 2015.

Pelegrí, J. L. and Peña‐Izquierdo, J.: Eastern boundary currents off North‐West Africa. In: Valdés, L. and Déniz‐González, I. (eds). Oceanographic and biological features in the Canary Current Large Marine Ecosystem. IOC‐UNESCO, Paris. IOC Technical Series, 115, 81‐92, 2015.

Pelegrí, J. L., Marrero‐Díaz, A., and Ratsimandresy, A. W.: Nutrient irrigation of the

North Atlantic. Progress in Oceanography, 70, 366–406, 2006.

Peña‐Izquierdo, J., Pelegrí, J. L., Pastor, M. V., Castellanos, P., Emelianov, M., Gasser, M., Salvador, J., and Vázquez‐Domínguez, E.: The continental slope current system between Cape Verde and the Canary Islands. Scientia Marina, 76 (S1), 65–78, 2012.

Peña‐Izquierdo, J., Van Sebille, E., Pelegrí, J. L., Sprintall, J., Mason, E., Llanillo, P., and Machín, F.: Water mass pathways to the North Atlantic Oxygen Minimum Zone. Journal of Geophysical Research: Oceans, 120, 3350–3372, 2015.

---

## Author Comment (AC1) · 12 Apr 2017

**Answer to Referee #1**

We thank Referee nr.1 for the time spent on reviewing our manuscript and for his/her thoughtful comments that have helped us to better understand the role of our small detritus pool and the sensitivity of our results with regard to our treatment of organic matter. This will improve the quality of our manuscript. We include below our detailed answers to all the raised questions/comments.

**Answers to Major comments**

**Major Comment nr.1:**
*The NPZD model is very simple which is not a problem by itself. It performs enough well to be suitable for that study. This is quite clearly shown in the validation section of the paper. However, according to me, it lacks a critical reservoir especially concerning the objectives of that study: DOC or more precisely semi-labile and semi-refractory DOC. Concentrations of semi-labile DOC range from typically 20 to 40 umol/L in the upper ocean (Hansell et al., 2009; Hansell and Carlson, 2014). Its lifetime is also quite long and ranges from weeks to years which makes it possible for that pool to be transported far away from its production region. It has been shown to potentially play an important role in the subtropical gyres (e.g., Roussenov et al., 2006; Torres-Valdés et al., 2009). In the present study, this pool is omitted and thus, a potentially large contribution to the lateral export of organic carbon is not represented. This needs at least to be discussed in the discussion section.*

**Answer to MC1:**
As correctly stated by this reviewer, our NPZD model does not include an explicit DOC pool, which at first sight could be considered as a serious shortcoming given the potentially substantial contribution of DOC to the lateral transport of organic carbon. However, our model includes, in addition to the standard pool of fast sinking (large) particulate organic carbon (Large Detritus, LDet), also a pool of very slowly sinking particles (Small Detritus, SDet). Given its sinking speed of 1 m day$^{-1}$ SDet represents essentially a suspended POC pool. Thus, this pool has some similarity to a (semi-refractory) DOC, particularly regarding its susceptibility to being subject to strong lateral transport. The important difference is that SDet coagulates to LDet, while this is not the case for DOC, i.e., SDet has a somewhat shorter lifetime in the surface ocean than the semi-refractory DOC. At the same time, the rate of production of DOC is likely smaller than that of SDet, since most of the organic matter produced in the surface ocean is routed first through SDet, while this is not the case for DOC. Thus, while we are clearly not representing DOC in our model simulations, we do not expect the explicit consideration of DOC to completely change our results. Or in other words, we would argue that the impact of this shortcoming is smaller than possibly inferred at first sight.

In order to explore the potential impacts of our lack of consideration of DOC more quantitatively, we ran a sensitivity study where we altered the behavior of SDet to become like DOC. Specifically, we set the sinking speed of the SDet pool to zero, i.e., $w_{SD}=0$, and reduced the coagulation time scale $t_{coag}$ to 3/5 of its baseline value to mimic as closely as possible a dissolved organic carbon pool. No

adjustments were made to the parameterization of the LDet pool to compensate for the strong reduction in the routing of organic carbon toward this pool. This sensitivity study thus needs to be considered as an extreme scenario - i.e., is meant to explore the potential contribution of DOC rather than an attempt to quantify it in detail. We spun up the model with the new biological parameters from year 24 of the baseline run (6 years of spinup) and used years 30-35 for the analysis, as we did for the baseline run.

The results of this sensitivity simulation (see Figures MC1-1-3 below) suggests that a dissolved pool of organic carbon would tend to intensify the lateral fluxes of organic carbon in the euphotic layer and stimulate the local recycling of organic matter, increasing both primary production and heterotrophic activity in the near-surface layer, but not alter net community production in a major manner. These apparently contradictory conclusions can be rationalized by our modifications resulting in a substantial increase in the average lifetime of SDet. Rather than becoming subject to sinking and coagulation, SDet now remains in the surface ocean, increasing the standing stock of POC there substantially, which increases also the offshore transport. However, due to the reduced reactivity of SDet resulting in a longer lifetime, the net horizontal divergence of SDet remains roughly the same, even though the transport is larger and reaching further out into the open North Atlantic. The roughly unchanged horizontal divergence of organic matter transport implies a roughly unchanged net community production as well. Thus, for the key question at hand, i.e., can the offshore transport fuel net heterotrophic conditions in the offshore regions of the Canary CS, the answer essentially remains unchanged.

[Figure]

**Figure MC1-1:** Map of Community Production including sediment remineralization in the sensitivity study with reduced sinking and coagulation of SDet: (a) vertically integrated in the whole watercolumn; (b) vertically integrated in the first 100m depth; (c) vertically integrated below 100m depth. Compare to Figure 6 in the main text.

[Figure]

**Figure MC1-2:** Map of horizontal transport of POC in the sensitivity case with a non-sinking and very slowly aggregating SDet pool. (a) zonal transport of total POC in the top 100 m. (b) as (a), but for the meridional transport. Contrast this to Figure 11 in the main text.

In response to this comment, we will clarify the role of SDet and our lack of consideration of an explicit DOC pool in the text. Concretely, we propose to include a dedicated paragraph in the discussion section to examine the potential contribution of DOC to the lateral redistribution of organic carbon. This paragraph will include some literature-based discussion on the base of the relevant papers kindly suggested by this referee, as well as the results of this sensitivity simulations (w/o figures). We further will make sure throughout the text that the reader remains aware that our model-based study deals with the lateral transport of POC only, and not of total organic carbon. A comment to this effect will also be added to the abstract.

**Major Comment nr.2:**
*In this study, the importance of mesoscale features is emphasized several times but never clearly quantified. It would have been nice to have such a quantification. I would suggest two possible means to do that: 1) to perform a classical separation technique between the mean and eddy components of the transport; 2) to perform a simulation in which the non linear terms in the Navier-Stokes equations for momentum are cancelled such as in Gruber et al. (2011). Otherwise, any discussion of the effect of the mesoscale circulation remains quite speculative and qualitative.*

**Answer to MC2:**
We agree with Referee nr.1 that mesoscale processes play an important role for the lateral redistribution of organic carbon in the region and that their contribution needs to be discussed more quantitatively. However, we are of the opinion that a full in-depth analysis goes well beyond the scope

of this paper, which is already quite detailed and long. Our preferred strategy is to leave this aspect to a second, dedicated publication that focuses exclusively on the role of mesoscale processes for the long-range transport of organic carbon in the region. This follow-up study will include an analysis of the decomposition of the fluxes into their mean and turbulent components, some sensitivity studies and a study of the influence of mesoscale eddies on the offshore transport and transformation of organic matter. The strategy we propose for this present paper is to strengthen the discussion of the mesoscale contribution with more concrete references to previous literature and also mentioning our knowledge obtained with the analysis that we are currently developing. We also propose to add in the present paper a reference to the follow-up study that we are currently working on.

**Answers to Detailed comments**

**DC1:** Page 2, line 1 - *"resuspension of bottom sediments and can create ... " I guess something is missing in this sentence.*
Thank you. The "and" is a typo, we will correct it to: "resuspension of bottom sediments can create..."

**DC2:** Page 4, line 33 - *In the list of state variables that are listed, you should add O2.*
Thank you, we will add it.

**DC3:** Page 5, lines 11-12 - *Phytoplankton can coagulate with small POC to form large POC. Is it also the case for small POC with small POC?*
Yes.  To clarify this we will  mention the smallPOC-smallPOC coagulation in the text.

**DC4:** Page 7, lines 28-33 - *You should refer to figure 3 to illustrate the different regions.*
Thanks for the suggestion. We will add a reference to the Figure.

**DC5:** Page 10, lines 8-13 - *Almost everywhere, except near Cape Blanc, high values of Chlorophyll are too narrow and too much trapped near the coast. As mentionned by the authors, this bias is especially strong in the Southern part of the CanUS domain.*
Yes, we acknowledge the limitation of the modeled surface Chlorophyll. However, along the whole northern coastline from 32°N down to Cape Blanc (21°N), surface Chlorophyll is not narrower than in the satellite product. Below Cape Blanc, Chlorophyll is underestimated at the surface due to a deepening of the chlorophyll maximum, as discussed in pages 10 and 11.

**DC6:** Page 11, lines 1-6 - *The authors here discuss the characteristics of the modeled sub-surface maximum of Chl (DCM) and they refer to Figure B2. This is not always easy to see from Figure B2. The most obvious bias that emerges from the figure is the too high values of Chl at depth below 50m. Otherwise, it is hard to quantify from that plot the depth of the DCM in the model and in the data.*
We take note of this comment. In response, we will add a better description of the figure to the paper. In particular, we will ensure to better explain the pattern of latitudes and depths.

**DC7:** Page 17, lines 3-6 - *For sure in the interior of the ocean, the contribution of small POC to the vertical sinking flux of organic matter should drop very quickly with depth. A figure showing the contribution of the different pools of organic matter to total organic carbon would be nice.*
We have added a plot of the mean vertical profiles of the four pools of organic carbon in the CanUS, as visible in the following Figure DC7-1. This figure will be included  the Appendix of the paper.

[Figure]

**Figure DC7-1**: Mean vertical offshore sections of the organic carbon components in mmolC/m$^3$; x-axis:offshore distance [km], y-axis: depth [m]

**DC8:** Page 18, Figure 9 - *The fluxes in the different boxes are not balanced (the imbalance is however small). Is it because the model is not fully at steady state or because of the internal variability related to the mesoscale activity?*
There are a few reasons why the fluxes are not completely balanced. The first reason is indeed the lack of a complete steady-state, which leads to changes in the size of the standing stocks, which we computed, but did not add to the figures. In addition, we also did not include in our analysis the contribution of horizontal and vertical mixing fluxes associated with the background diffusivity. However, these fluxes are very small. Another small source of error is the fact that our 3D analysis boxes are defined by horizontal boundaries that correspond to the position of the long-term mean sigma-levels, where the sigma levels define the terrain-following coordinate used in ROMS. However,

sigma-levels slightly move due to relatively small differences in SSH at each time step and this can result in slight miss-matches of the mean flux calculation.

In response to this comment, we will add some text to the figure caption to explain the reasons for the lack of closure.

**DC9:** Page 18, lines 1-6 - *The DeltaE diagnostics is interesting. It accounts for two processes that can increase the export without changing the NCP: 1) The organic matter that is being transported laterally and that sinks out of the upper ocean increases the export and thus DeltaE. 2) The organic matter that is being transported laterally and that remineralizes in the upper box. This stimulates the biological activity which produces more organic matter which is sinks out of the upper ocean. In that case, NCP is not changed (the increase in PP compensates for the remineralization of the laterally supplied organic carbon) and export is increased which increases DeltaE. This two mechanisms should be explained here, especially because in the discussion section it is shown that the second process dominates.*

We thank Referee nr.1 for his/her comment and we agree that it would be relevant to introduce this discussion before. In response we will use this suggestion and already explain the two possible mechanisms in the Results section, and then reconnect to this passage in the Discussion section where we discuss their relative contribution.

**DC10:** Page 20, line 22 *"and quantify the contribute of the different zonal bands ..." I guess it should be contribution*

Thanks. Will be corrected.

**DC11:** Page 22, line 25 *"and quickly channel water ..." It should be channel.*

Thanks. Will be corrected.

**DC12:** Page 24, line 8 *"becomes particularly important the offshore waters" Some words are missing here.*

Thanks, we will correct it to "important in the offshore waters"

**DC13:** Page 25, lines 18-20 - *The splitting between the contribution of the mean flow and of the eddy transport is not really clear here. See my second major concern above.*

We will be more specific and add a more detailed discussion of the mesoscale contribution as mentioned in our answer to the second major comment.

---

## Author Comment (AC2) · 12 Apr 2017

**Answer to Referee #2, Josep L. Pelegrí**

We thank Dr. Josep L. Pelegrí for his careful review of our manuscript and for his thoughtful comments that will surely help to improve its quality. We tried to address all his comments and we include below a detailed answer to all the questions.

**Answers to Major comments**

**Major comment nr.1:**
*(1) Evaluation of the model's performance*
*This is a very critical aspect and the authors dedicate a significantly long section, including an appendix, to evaluate the performance of the model. They compare the numerical output with observations using different datasets: the near-surface seasonal circulation as inferred from surface drifters; the annual-mean sea surface height, sea surface temperature (SST) and sea surface salinity; the annual-mean mixed layer depth (MLD); the annual-mean surface chlorophyll; and the annual-mean and seasonal-mean net primary production (NPP).*
*I value this effort very much but, honestly, at the end of the Evaluation section I have important doubts on how good the model's performance is. Throughout this section the authors recognize the existence of substantial differences between model and field data, and also talk about model bias.*

**Answer to MC1:**
As stated by Dr. Pelegrí, we have invested quite some effort to carefully evaluate many aspects of our model. As a result, we feel that we are well aware of its strengths and its limitations. While there are clearly some issues, the results of our model evaluation are in line with most state of the art models – in many respects the fidelity of the model simulated fields is even better than that of most models. However, it is clear that models are never perfect, so the question we have to answer is to what degree biases and other types of errors will affect the results and the conclusions drawn from them. Our overall assessment is that, despite the biases, that the performance of our model is more than adequate to answer the main scientific question regarding the magnitude and the importance of the long-range lateral fluxes of organic carbon. Thanks to the information provided by a detailed model comparison with observations, we are also able to discuss in the paper how our results are affected by the observed biases, especially in the southern subregion of the Canary Upwelling System (CanUS), where we see the largest and most relevant differences from the observations. We address the different elements of this first main comment in sequence:

*A) In Figure 2 they show the spatial distribution of model-data differences for several surface fields. The differences are not negligible at all, as clearly seen by the range of values in the mean fields and the differences, e.g. SST (range of values is 12° C and range in deviations is 4°C) and MLD (range of valuesis 100 m and range in deviations is 60 m).*

The SST biases are clearly significant but actually quite a bit smaller than implied by Dr. Pelegrí's comment, i.e., ±2°C.  The SST plot (Figure 2b) shows that differences between model and observations

lay in the interval [-0.75ºC,1ºC] in the large majority of the domain, with a large fraction of this bias having a range of only ±0.5ºC. Larger differences are confined to a very narrow coastal band. The region located south of Cape Blanc has the extensive bias. But also here, the (positive) bias has a range of only [0.5ºC,0.75ºC]. This warm bias is accompanied by a positive bias in salinity of about 0.5 (Figure 2c), leading to a near complete compensating with respect to their impact on density. Overall, we consider these biases to be small relative to the spatial and temporal variations. They are also too small to affect substantially primary production or the lateral export of organic carbon. Therefore we expect that these SST and salinity biases have a minor impact on our study. In response to this comment, we will discuss the SST and salinity biases and their impact on the study more explicitly.

The biases in the mixed layer depth are likely more relevant for our study. As highlighted by Dr. Pelegrí, while the modeled distribution agrees overall reasonably well with the observed one based on Argo-floats, our modeled MLD shows sharper gradients than the observed pattern resulting in rather large differences in the northern nearshore and the central offshore region. This could be a true bias of our model, but we also note that the Argo DT-0.2 MLD product was gridded on a relatively low-resolution 2ºx2º grid and that it has a rather limited coverage in the nearshore areas. As a result this product may not be able to properly capture strong gradients and overly smooth distribution relative to reality in regions with strong variations, such as ours. Given our MLD bias structure, it is feasible that some fraction of it could be attributed to biases in the Argo-based product.

In response to this comment, we will extend our already existing discussion in the Results and Discussion sections with a more in depth analysis. In particular, we propose to add a short paragraph to the discussion section to assess the potential impact of these biases in more detail. We also plan to add some material with regard to the impact on productivity (e.g., alteration of the light limitation) and lateral carbon transport.

*B) I particularly miss a comparison between the depth distribution of the modelled and observed particulate organic carbon (POC), which is of capital importance for this study. The seasonal results (Figure 5) show very large differences, possibly too large.*

We agree with Dr. Pelegrí regarding the necessity of having a comparison of modeled and observed POC and we thank him for this suggestion. To this end, we have conducted an evaluation of the modeled POC using 2 datasets: 1) the MODIS satellite estimate of surface POC (S-POC in the diagram); 2) the cruise POC measurements from AMT, ANT and Geotraces in the upper 200 m (POC in the diagram) located in the 0km-2000km offshore range of our analysis domain. Most of the in-situ data were collected in fall, especially in October, often in the far offshore region of our domain (cf. Figure MC1-1).

[Figure]

circle=AMT, square=ANT-XXIII/1, diamond=Geotraces

**Figure MC1-1:** Retrieved cruise POC measurements in the region of analysis corresponding to the Budget Analysis boxes. Data points are colored by sampling month and by maximum depth of the measurement. Circles=AMT (15-23), Squares=ANT, Diamonds=Geotraces.

Modeled POC and data were co-located in space and time using a daily ROMS climatology for the same 6 years. As visible from the resulting plot (Figure MC1-2), the magnitude of modeled and observed POC is the same and the vertically-integrated POC in the first 100m also corresponds. Due to our coastally-confined production (largely discussed in the model evaluation) combined with the fact that cruise data are mostly located offshore, and due to the deepening of the chlorophyll maximum in the southern productive subregion, we observe a deeper-than-expected POC maximum in the model. As also discussed in the paper, this may mean that if anything our model may underestimate the offshore transport in the CanUS (and especially in its southern sector), therefore implying that the already large magnitude of the offshore transport that we find may be a low estimate.

In response to this comment, we will include in the Taylor diagram to a comparison with POC (see Figure MC1-3) to both satellite estimates and in situ measurements. We will also highlight that in our paper we already provide a plot (Figure 7) of the mean vertical profile of the total organic carbon for the whole Canary Upwelling System.

[Figure]

**Figure MC1-2:** mean POC profile in the CanUS compared to cruise data, from co-located POC, binned in depth to 10m depth intervals.

[Figure]

**Figure MC1-3**: Annual mean Taylor diagram including:
1) an evaluation of surface particulate organic carbon (**S-POC**) using SeaWiFS satellite estimates;
2) a comparison with depth profiles of **POC** from cruise data through co-location of ROMS output in space and time.

For additional discussion of the implications of having a shallower POC distribution we refer also to our answer to Anonymous Referee 1, in which we discuss the results of some sensitivity studies in terms of both transport and impact on NCP.

*C) The authors end this section referring to a Taylor diagram presented in Appendix B (Figure B3), concluding that there is a "good correlation between the modelled and observed fields both in the annual and in the seasonal means." They show the Taylor diagram for the annual-mean results and for the mean of the seasonal results. The authors argue that the Taylor diagram shows results comparable to other studies for upwelling systems. Rather than comparing with other studies, it would be better to look at the statistics and discuss whether the results are convincing or not. For the annual-mean, for example, SST, CHLA and MLD respectively have a (normalized) standard deviation of about 1.2, 0.6 and 1.4, and a (normalized) root-mean-square difference of 0.3, 0.7 and 0.8. The authors should discuss whether these values are reasonable or not.*

Following Dr. Pelegrí's comment, we will be more specific in the description of our Taylor diagrams in the Evaluation section. Regarding our Taylor diagrams included in Figure B3 and here in Figure MC1-3 and Figure MC1-4, all the variables show a correlation of 0.7 or higher with the observations in the annual mean (except cruise data POC) and 0.68 or higher in the seasonal. Among all variables considered, the values of the normalized standard deviations are particularly high for annual mean MLD (1.5), due to the too sharp gradients and high peaks discussed in paragraph (A) of this document. Low values of the normalized standard deviations are instead observed for chlorophyll (0.65) and for NPP1 (0.35) that corresponds to NPP compared to the SeaWiFS VGPM product. This is a consequence

of the fact that, even though the representation of the pattern of the variable in the model is close enough to the observations, the magnitude of the modeled blooms is not as intense. Interestingly, if modeled NPP is compared to NPP from the SeaWiFS CbPM product (shown in the Taylor diagram as NPP2), a normalized standard deviation of about 0.75 emerges. This implies a rather large differences in the estimated NPP in the two products.

*D) I am particularly confused by Figure B3b: how is this calculated, just an average mean? What is the meaning? Wouldn't it be much better to show all four seasonal diagrams? It would also help to include, as supplementary materials, diagrams for each subregion.*

The Taylor diagram in Figure B3b is calculated as the simple mean of the seasonal Taylor diagrams. However, as suggested by Dr. Pelegrí, we have decided to substitute this figure, and explicitly include in the appendix of the paper the four seasonal Taylor diagrams, here visible in Figure MC1-4. We have now included surface POC (S-POC) compared against the SeaWiFS satellite estimates also in these diagrams.

[Figure]

**Figure MC1-4:** Seasonal Taylor diagrams, including surface particulate organic carbon (S-POC) through a comparison with SeaWiFS satellite product estimate.

In the Summer diagram MLD was rescaled to MLD*=MLD/2. The summer $MLD_{STD}$ is therefore 2 times as big as the one represented in the plot, while the correlation remains unchanged.

**Major comment nr.2:**

*(2) Latitudinal partition of the domain*

*A) In several places of the Introduction and Discussion the authors recognize that the Cape Verde frontal zone is a natural boundary between the subtropical and tropical domains. Nevertheless, for most of their analysis on latitudinal variability they use a partition in three areas or subregions, as shown in Figure 3b, which is not properly justified. I imagine this is done as an attempt to grasp the character of the meridionally convergent region near Cape Blanc but, as it is clear from the velocity fields in Figures 3 and 5, this is not correct. In my opinion only the southern subregion would comprise an area with approximately coherent dynamics. My suggestion here is to use four subregions of different size: the northern one (25- 32◦ N) would correspond to an area with substantial mesoscalar activity, with eddies and filaments generated both south of the Canary Archipelago and at the upwelling front; the second area would represent the permanent and intense central upwelling region (21-25◦ N); the third are would concentrate on the convergent region immediately south of Cape Blanc, which is the root of the Cape Blanc giant filament (about 17-21◦ N, though these limits change with longitude); the southern area (9.5-17◦ N) would correspond to the tropical region. Right now most of the discussion is either on the results for the latitudinal-average picture or (to a lesser degree) for the three proposed regularly-spaced subregions. With this alternative partition, the paper would certainly become much more informative.*

*B) I value very much the authors' efforts to provide bulk figures for the entire region but I think that plotting these results may be very misleading. For example, the data in Figure 8 suggests that the zonal flux of organic carbon is more intense than meridional one. I doubt this very much: in my opinion this is only an artefact that the latitudinal average tends to cancel the contributions of the southward Canary Upwelling Current and northward Mauritania Current and Poleward Undercurrent (please see references below regarding the main currents in the CanUS). My suggestion is to produce fewer plots on the results for the entire region (Figure 9 is fine but some other plots may be replaced by tables) and instead show what is happening in each area: the CanUS is so large that it surely deserves a closer view for each subregion.*

**Answer to MC2:**

**A)** We agree with Dr. Pelegrí that other choices for the subregional Budget Analysis domain were also possible. Our partition serves to quantify both the alongshore convergence of particulate organic carbon from both north and south of Cape Blanc and the subsequent intense offshore flow that takes place along the Cape Verde front. The use of wide domains allows us to have a more robust measure of the fluxes in a region of high mesoscale variability. This partition also avoids us to place boundaries in critical regions such as around the Cape Verde convergence; placing boundaries in such flux-intense regions would make the results of our budget analysis very sensitive to the exact latitude of the boundary.

However, we have considered the latitudinal partition proposed by Dr. Pelegrí, and repeated our analysis on his proposed domains, as shown in Figure MC2-1. The changes basically consists in a sub-division of the central domain into two smaller zonal bands. Our northern and southern zonal bands

already satisfied Dr. Pelegrí's definitions, corresponding to a northern subregion rich in mesoscale activity (now only displaced by half degree) and a southern tropical subregion. The results of the new budget analysis are displayed in Figure MC2-2. As expected, northern and southern subregions are characterized by the same pattern of fluxes as those presented in the paper, since moving the southern boundary of the northern subregion by half a degree north does not affect the budget. The central subregion is split in a "central north" and "central south" zonal bands (green and orange lines). The impact of the offshore flux in these two zonal bands is very similar (Figure MC2-2, panel b). The flow of the Cape Verde front crosses the boundary between the "central north" and "central south" zonal band at about 1000km offshore, adding to the offshore flux in the "central south" subregion at this distance from the coast. However, this effect is an artifact generated by the split of the front in two segments. It thus does not add much to our understanding of the magnitude or the impact of the long-range offshore flux at these latitudes. As regard to the alongshore fluxes, we find that dividing the central subregion in two zonal bands does not clarify the source of the organic carbon that is exported offshore along the Cape Verde front. In fact, while before we could clearly identify the central subregion as a region of alongshore convergence of the organic carbon, now the budget for the "central north" and "central south" subregions depends strongly on the exact location of the intermediate boundary and the exact pathway of the Canary and Mauritanian currents near Cape Blanc. For example, in the 0km-100km offshore range, the "central north" subregion still exports southward more carbon than what it receives from its northern boundary, resulting in a net alongshore export to the "central south" subregion. In contrast, in the 100km-500km offshore range of the "central north" subregion a local recirculation of the carbon from the "central south" zonal band is visible in the meridional fluxes plot of Figure 11 of the paper. This recirculation induces a large net influx of organic carbon in the "central north" subregion. All these effects strongly depend on the exact location of the intermediate boundary in this region of intense flux convergence. As a consequence, we believe that the use of just one large central subregion for the budget analysis is more appropriate for our main purpose of understanding the magnitude and possibly the sources of the lateral offshore flux of organic carbon in the CanUS. To clarify the reasons that lay behind our choice of the domains used for the budget analysis, we will therefore include in the paper a clear and detailed explanation.

[Figure]

**Figure MC1-1:** Comparison between the latitudinal domains proposed by Dr. Pelegrí (red lines) and the domains used in the paper (blue lines).

[Figure]

**Figure MC1-2:** Budget analysis results based on the subregions proposed by Dr. Pelegrí. Trends of NCP and impact of the organic carbon fluxes (divergence of the flux / NCP) by subregion.

**B)** We thank Dr. Pelegrí for this comment, and we agree with him on the fact that alongshore fluxes are locally more intense than the offshore fluxes in the nearshore. We also agree that, since Figure 8b shows the meridional fluxes averaged over the whole CanUS, the contribution of the northward and southward alongshore currents mostly cancel each other. In this specific case, will reconsider the picture and its description and we will discuss more explicitly the relative contribution of the zonal and meridional components of the fluxes. In general, we still believe that an in depth discussion of the bulk fluxes is very relevant for our purpose of quantifying the overall magnitude of the lateral fluxes from the North African coast on the North Atlantic gyre. For this reason, we plan to keep the original figures.

**Major comment nr.3:**
*(3) Upwelling of coastal and offshore inorganic nutrients*
*The coastal upwelling region is a source of inorganic nutrients to the surface layers in the coastal transition zone that are later exported offshore (e.g. Pelegrí et al., 2006; Pastor et al., 2008, 2013). Such a flux of inorganic nutrients is a prime element in the offshore net primary production and the sign of the NCP north of Cape Blanc. However, this issue is not mentioned in the manuscript until the Discussion. The subject is important enough to deserve careful attention when examining the sources and sinks for NPP, it is the difference between new production using the subsurface load of inorganic nutrients or production after remineralization.*
*The offshore waters in the southern subregion are also largely affected by the presence of upward Ekman pumping, i.e. offshore upwelling resulting from positive wind-stress curl. Again this is an important aspect in the dynamics and NPP balance of this subregion, which is again acknowledged very late in the manuscript and only partly discussed.*
*The model could be used to assess these different contributions. Perhaps this was not the objective of the authors, which is fine, but then the potential relevance of the upwelling and transport of inorganic nutrients on the NPP and NCP within the entire region should be properly discussed since early in the manuscript.*

We agree with Dr. Pelegrí on the importance of including a discussion of sources and sinks of nutrients in the region. For this reason we have decided to include in the paper a figure showing the lateral and vertical fluxes of inorganic nutrients (Figure MC3-1). This will highlight the importance of these fluxes and improve the discussion of the pattern and of the subregional variability of NCP, underlining in particular the importance of the coastal upwelling of nutrients in the northern and of the Ekman pumping in the southern Canary Upwelling System.

[Figure]

**Figure MC3-1:** Inorganic Nitrogen fluxes in the first 100m. Horizontal fluxes were integrated in the first 100m, while vertical fluxes were sliced at 100m depth.

**Answers to Minor comments**

(4) p 1, l 10: *divergence or convergence?*
The divergence is negative, which means flow is convergent.

(5) p 2, l 11: *replace "and can create" by "can create."*
Thank you, it was a typo. We will correct it.

(6) p 2, l 15: *"Arístegui."*
Thank you, it was a typo. We will correct it.

(7) p 3, l 5: *other relevant references are Pelegrí et al. (2006) and Pastor et al. (2013).*
Thank you for your suggestion.

(8) p 3, l 24: *also Pastor et al. (2013).*
Thank you.

(9) section on Methods: *how does the model calculate the vertical velocity?*
The vertical velocity is computed through integration of the mass-conservation equation of an incompressible fluid ($\vec{\nabla} \cdot \vec{u} = 0$) from the ocean floor upwards. For detail about the equations solved by ROMS and its numerical solution technique we refer to Shchepetkin and McWilliams (2005) and the ROMS Wiki (https://www.myroms.org/wiki/Equations_of_Motion).

(10) caption of Figure 1: *Gran Canaria is cited in the caption but not located in the map.*
Thank you, we will add the name in the figure.

(11) p 10, l 10-13: *please clarify.*
We will revise this sentence.

(12) caption of Figure 4: *VGPM is first mentioned here but it is defined nowhere in the manuscript.*
The used SeaWiFS VGPM product is described in detail in Table A3 (Appendix) among the products used for the Model evaluation. We will add a reference also in the text.

(13) p 12, l 21-24: *asides the Canary Current and the Mauritanian Current you should also probably refer to the Canary Upwelling Current (associated to the coastal upwelling jet) and the Poleward Undercurrent (please see references below).*
Thank you, we will include it in the description.

(14) p 12, l 31-32: *". . . NPP is a better measure than chlorophyll for evaluating. . ."*
Thank you, we will correct the sentence.

(15) p 13, l 1: *Pastor et al. (2013) is probably a better reference.*
Thank you for your suggestion.

(16) caption Figure 6: *panel b also includes sediment remineralization?*
Yes, all panels include remineralization. We will adjust the figure caption accordingly.

(17) p 18, l 3-5: *are you using two different definitions for excess export?*
No, $\Delta E$ is always defined as $\Delta E$= vertical export - NCP

(18) p 20, l 1-2: *here and elsewhere it is best to not refer to lines, they should be defined in the figure's caption or legend (otherwise you would have to define them everywhere).*
Thank you, we will revise the references.

(19) p 20, l 27 and 33: *"north of the Cape Verde front. . ."*
Thank you, we will correct it.

(20) p 22, l 2: *". . .south of Cape Blanc."*
Thank you.

(21) *please revise caption of Figure 12.*
Thank you, we will revise it.

(22) Figure 13: *I suggest that you also show the meridional fluxes.*
Thank you for your suggestion. We evaluated these figures, and we have decided to add a discussion of the vertical sections and depth trend of the meridional fluxes by subregion in the text. However, we prefer not to include this plot, since it does not add any substantial information that cannot be inferred from the 2D plot of the meridional fluxes, Figure 11b.

(23) caption Figure 13: *"vertical" rather than "vertcal."*
Thank you. We will correct the typo.

(24) p 24, l 14: *"(Figure 13)."*
Thank you. We will correct the typo.

(25) p 25, l 6-8: *this is likely an artefact of the SW-NE orientation of the coast.*
Thank you, we will revise the paragraph to take this effect into account.

(26) p 25, l 10 and 15: *please include references.*
Thank you, we will add references .

(27) p 27, l 8: *see also Pastor et al. (2013).*
Thank you for your suggestion.

(28) p 28, l 7: *"these."*
Thank you, we will correct it.

(29) p 28, l 10-18: *usage of so many conditionals raises doubts on the reader.*
Thank you, we will revise the writing.

(30) p 28, l 27: *remove "from Section 4.1."*
Thank you, we will remove it.

(31) p 28, l 28: *is this the right way to cite a figure within a reference?*
We will revise the reference.

(32) p 29, l 3: *here and elsewhere separate numbers from units, i.e. "2000 km" rather than "2000km."*
Thank you, we will correct it.

(33) p 29, l 16-17: *please revise writing.*
We will revise these sentences.

(34) additional references: *asides those mentioned above, there are other works that would help better describe the circulation patterns in the CanUS, such as Mason et al.(2011), Peña-Izquierdo et al. (2012, 2015), Pelegrí and Peña-Izquierdo (2015), Pelegrí and Benazzouz (2015).*
Thank you for the suggestions, we plan to add these references.

---

## Author Response (AR1)

**Author's Response**

Dear Dr. Jack Middelburg,
thank you very much for taking our manuscript into further consideration for publication in Biogeosciences.

We have carefully revised our manuscript following your suggestions and we include below:

- our point-by-point answers to the two Referees including the detailed discussion of the major and minor comments, and (marked in color) the actual modifications made on the manuscript for each point;

- a marked-up version of the manuscript.

We look forward to your response.
Best regards,
Elisa Lovecchio

**Answer to Referee #1**

We thank Referee nr.1 for the time spent on reviewing our manuscript and for his/her thoughtful comments that have helped us to better understand the role of our small detritus pool and the sensitivity of our results with regard to our treatment of organic matter. This will improve the quality of our manuscript. We include below our detailed answers to all the raised questions/comments.

**Answers to Major comments**

**Major Comment nr.1:**
*The NPZD model is very simple which is not a problem by itself. It performs enough well to be suitable for that study. This is quite clearly shown in the validation section of the paper. However, according to me, it lacks a critical reservoir especially concerning the objectives of that study: DOC or more precisely semi-labile and semi-refractory DOC. Concentrations of semi-labile DOC range from typically 20 to 40 umol/L in the upper ocean (Hansell et al., 2009; Hansell and Carlson, 2014). Its lifetime is also quite long and ranges from weeks to years which makes it possible for that pool to be transported far away from its production region. It has been shown to potentially play an important role in the subtropical gyres (e.g., Roussenov et al., 2006; Torres-Valdés et al., 2009). In the present study, this pool is omitted and thus, a potentially large contribution to the lateral export of organic carbon is not represented. This needs at least to be discussed in the discussion section.*

**Answer to MC1:**
As correctly stated by this reviewer, our NPZD model does not include an explicit DOC pool, which at first sight could be considered as a serious shortcoming given the potentially substantial contribution of DOC to the lateral transport of organic carbon. However, our model includes, in addition to the standard pool of fast sinking (large) particulate organic carbon (Large Detritus, LDet), also a pool of very slowly sinking particles (Small Detritus, SDet). Given its sinking speed of 1 m day$^{-1}$ SDet represents essentially a suspended POC pool. Thus, this pool has some similarity to a (semi-refractory) DOC, particularly regarding its susceptibility to being subject to strong lateral transport. The important difference is that SDet coagulates to LDet, while this is not the case for DOC, i.e., SDet has a somewhat shorter lifetime in the surface ocean than the semi-refractory DOC. At the same time, the rate of production of DOC is likely smaller than that of SDet, since most of the organic matter produced in the surface ocean is routed first through SDet, while this is not the case for DOC. Thus, while we are clearly not representing DOC in our model simulations, we do not expect the explicit consideration of DOC to completely change our results. Or in other words, we would argue that the impact of this shortcoming is smaller than possibly inferred at first sight.

In order to explore the potential impacts of our lack of consideration of DOC more quantitatively, we ran a sensitivity study where we altered the behavior of SDet to become like DOC. Specifically, we set the sinking speed of the SDet pool to zero, i.e., $w_{SD}=0$, and reduced the coagulation time scale $t_{coag}$ to 2/5 of its baseline value to mimic as closely as possible a dissolved organic carbon pool. No

adjustments were made to the parameterization of the LDet pool to compensate for the strong reduction in the routing of organic carbon toward this pool. This sensitivity study thus needs to be considered as an extreme scenario - i.e., is meant to explore the potential contribution of DOC rather than an attempt to quantify it in detail. We spun up the model with the new biological parameters from year 24 of the baseline run (6 years of spinup) and used years 30-35 for the analysis, as we did for the baseline run.

The results of this sensitivity simulation (see Figures MC1-1-3 below) suggests that a dissolved pool of organic carbon would tend to intensify the lateral fluxes of organic carbon in the euphotic layer and stimulate the local recycling of organic matter, increasing both primary production and heterotrophic activity in the near-surface layer, but not alter net community production in a major manner. These apparently contradictory conclusions can be rationalized by our modifications resulting in a substantial increase in the average lifetime of SDet. Rather than becoming subject to sinking and coagulation, SDet now remains in the surface ocean, increasing the standing stock of POC there substantially, which increases also the offshore transport. However, due to the reduced reactivity of SDet resulting in a longer lifetime, the net horizontal divergence of SDet remains roughly the same, even though the transport is larger and reaching further out into the open North Atlantic. The roughly unchanged horizontal divergence of organic matter transport implies a roughly unchanged net community production as well. Thus, for the key question at hand, i.e., can the offshore transport fuel net heterotrophic conditions in the offshore regions of the Canary CS, the answer essentially remains unchanged.

[Figure]

**Figure MC1-1:** Map of Community Production including sediment remineralization in the sensitivity study with reduced sinking and coagulation of SDet: (a) vertically integrated in the whole watercolumn; (b) vertically integrated in the first 100m depth; (c) vertically integrated below 100m depth. Compare to Figure 6 in the main text.

[Figure]

**Figure MC1-2:** Map of horizontal transport of POC in the sensitivity case with a non-sinking and very slowly aggregating SDet pool. (a) zonal transport of total POC in the top 100 m. (b) as (a), but for the meridional transport. Contrast this to Figure 11 in the main text.

In response to this comment:

- We have mentioned this model limitation and its potential implications directly in the Abstract (p.1, ll.18-21):

> "Our modeled offshore transport of organic carbon is likely a lower bound estimate due to our lack of full consideration of the contribution of dissolved organic carbon and that of particulate organic carbon stemming from the resuspension of sediments. But even in the absence of these contributions, our results emphasize the fundamental role of the lateral redistribution of the organic carbon for the maintenance of the heterotrophic activity in the open sea."

- We have modified our manuscript in order to refer to our modeled total organic carbon explicitly as Particulate Organic Carbon POC.
In particular, we have added the next passage in the Methods section (pag.6 ll.2-6):

> "As the model does not include an explicit DOC pool, our modeled total organic carbon corresponds to POC only. However, the small detritus, given its very small sinking speed, behaves essentially as a suspended POC pool, i.e., shares many similarities to DOC. To assess the possible implications of our neglecting DOC, we run a sensitivity experiment where we turned the small detritus pool into essentially DOC by setting its sinking velocity to zero and by reducing it coagulation rate."

- We have divided the Discussion section into two subsections (namely "Implications and comparison with previous work" and "Limitation and caveats") and in the latter subsection we have introduced a new paragraph to examine the potential contribution of DOC to the lateral redistribution of organic carbon, based on the suggested literature and on the results our sensitivity experiment (p.29 l.14 – p.30 l.4):

> "Regarding DOC, the pool that matters is that of semi-labile DOC as it has a life time of beyond a few days, implying that it can be transported substantial distances before it gets remineralized. As a result, it has the potential to enhance our modeled lateral export of organic carbon. This is especially the case since DOC is readily produced in the surface ocean and contributes also substantially to the export of organic matter from the near-surface ocean (Hansell, 2002; Arístegui et al., 2002; Hansell et al., 2009; Hansell and Carlson, 2015), in particular in subtropical regions such as the North Atlantic gyre (Torres-Valdés et al., 2009; Roussenov et al., 2006). Even though DOC is not explicitly modeled, the small detritus, with its sinking speed of $w_{SD}$ =1 m day$^{-1}$ , represents essentially a suspended POC pool with some similarity to a semi-refractory DOC, particularly regarding its susceptibility to lateral transport. But differing from DOC, the small detritus coagulates to large detritus resulting in a shorter lifetime than DOC in the surface ocean. At the same time, the rate of production of DOC is likely smaller than that of the small detritus, likely leading to a situation where the small detrital pool likely has a behavior that is rather close to that of DOC. Thus, we would argue that the impact of our shortcoming of not representing the dynamics of DOC explicitly is smaller than possibly inferred at first sight. In order to explore more quantitatively the potential impacts of our lack of explicit consideration of DOC, we ran a sensitivity study, in which we set the vertical sinking of the small detritus, $w_{SD}$, to zero and reduced the coagulation time scale for small detritus to 40% of its baseline value. No adjustments were made to the parameterization of the large detritus. This sensitivity study needs to be considered as an extreme scenario - i.e., it is meant to explore the potential contribution of DOC rather than an attempt to quantify it in detail. We spun up the model with the new biological parameters from year 24 of the baseline run (6 years of spinup) and used years 30-35 for the analysis, as for the baseline run. The results show, as expected, an intensification of the lateral fluxes of organic carbon in the euphotic layer. The standing stock of suspended POC increases about twofold, largely due to its longer average lifetime in the surface ocean, stimulating the local recycling of organic matter. This increases both primary production and heterotrophic activity in the near-surface layer, leaving the NCP pattern basically unchanged and preserving the net autotrophy of the near-surface waters. In fact, even though the lateral transport of small detritus is much larger in this sensitivity study and reaching further out into the open North Atlantic, the net horizontal divergence of the lateral flux remains roughly the same. Thus, for the key question at hand, i.e., can the offshore transport fuel net heterotrophic conditions in the offshore regions of the Canary CS, the answer essentially remains the same."

**Major Comment nr.2:**
*In this study, the importance of mesoscale features is emphasized several times but never clearly quantified. It would have been nice to have such a quantification. I would suggest two possible means to do that: 1) to perform a classical separation technique between the mean and eddy components of the transport; 2) to perform a simulation in which the non linear terms in the Navier-Stokes equations for momentum are cancelled such as in Gruber et al. (2011). Otherwise, any discussion of the effect of the mesoscale circulation remains quite speculative and qualitative.*

**Answer to MC2:**
We agree with Referee nr.1 that mesoscale processes play an important role for the lateral redistribution of organic carbon in the region and that their contribution needs to be discussed more quantitatively. However, we are of the opinion that a full in-depth analysis goes well beyond the scope of this paper, which is already quite detailed and long. Our preferred strategy is to leave this aspect to a second, dedicated publication that focuses exclusively on the role of mesoscale processes for the long-range transport of organic carbon in the region. This follow-up study will include an analysis of the decomposition of the fluxes into their mean and turbulent components, some sensitivity studies and a study of the influence of mesoscale eddies on the offshore transport and transformation of organic matter.

- For this present paper we have strengthened the discussion of the mesoscale contribution with more concrete references to previous literature such as Arístegui et al. (2009), Gabric et al. (1993), Fischer et al. (2009), Álvarez-Salgado and Arístegui (2015). We have also added in the present paper a reference to the follow-up study that we are currently working on (p.26 l.20-22):

> "Further insights into the special role of mesoscale activity in the lateral redistribution of organic carbon in the CanUS and a quantification of this component of the transport will be provided in detail in a dedicated publication."

**Answers to Detailed comments**

**DC1:** Page 2, line 1 - *"resuspension of bottom sediments and can create ... " I guess something is missing in this sentence.*
Thank you. We have corrected it to: "resuspension of bottom sediments can create..."

**DC2:** Page 4, line 33 - *In the list of state variables that are listed, you should add O2.*
Thank you, we have added the variable (p.5, ll.1-3):

"An additional four state variables have been added to reflect the cycling of carbon and oxygen, namely dissolved inorganic carbon (DIC), alkalinity, mineral $CaCO_3$ and dissolved oxygen ($O_2$) (Hauri et al., 2013; Turi et al., 2014; Lachkar and Gruber, 2013)."

**DC3:** Page 5, lines 11-12 - *Phytoplankton can coagulate with small POC to form large POC. Is it also the case for small POC with small POC?*
Yes, thank you. We have added the smallPOC-smallPOC coagulation in the text (p.5, ll.12-13):
"Coagulation of phytoplankton with small detritus as well as coagulation of small detritus with small detritus also forms large detritus..."

**DC4:** Page 7, lines 28-33 - *You should refer to figure 3 to illustrate the different regions.*
Thanks for the suggestion. We have added a reference to Figure 3 (p.8 ll.14-15):
"The lateral extension of the full CanUS boxes and of the subregional boxes is presented in Figure 3 together with the pattern of the modeled currents."

**DC5:** Page 10, lines 8-13 - *Almost everywhere, except near Cape Blanc, high values of Chlorophyll are too narrow and too much trapped near the coast. As mentionned by the authors, this bias is especially strong in the Southern part of the CanUS domain.*
Yes, we acknowledge the limitation of the modeled surface Chlorophyll. However, along the whole northern coastline from 32°N down to Cape Blanc (21°N), surface Chlorophyll is not narrower than in the satellite product. Below Cape Blanc, Chlorophyll is underestimated at the surface due to a deepening of the chlorophyll maximum, as discussed in pages 10 and 11.

**DC6:** Page 11, lines 1-6 - *The authors here discuss the characteristics of the modeled sub-surface maximum of Chl (DCM) and they refer to Figure B2. This is not always easy to see from Figure B2. The most obvious bias that emerges from the figure is the too high values of Chl at depth below 50m. Otherwise, it is hard to quantify from that plot the depth of the DCM in the model and in the data.*
We take note of this comment. In response, we have added a short description of the figure in the caption (p.35, Figure B2):
"Evaluation of the modeled annual mean Chlorophyll (CHL) by subregion and by depth for the first 500km offshore as defined by the first two budget analysis boxes, see Figure 4. The spread of the dots is maximum for the southern subregion, in which modeled CHL is too low at small depths and too high at large depths."

**DC7:** Page 17, lines 3-6 - *For sure in the interior of the ocean, the contribution of small POC to the vertical sinking flux of organic matter should drop very quickly with depth. A figure showing the contribution of the different pools of organic matter to total organic carbon would be nice.*
We have added a plot of the mean vertical profiles of the four pools of modeled organic carbon in the CanUS in the appendix of the paper as Figure B6, p.38. The plot is here visible as visible in the following Figure DC7-1.

[Figure]

**Figure DC7-1**: Mean vertical offshore sections of the organic carbon components in mmolC/m[3]; x-axis:offshore distance [km], y-axis: depth [m]

**DC8:** Page 18, Figure 9 - *The fluxes in the different boxes are not balanced (the imbalance is however small). Is it because the model is not fully at steady state or because of the internal variability related to the mesoscale activity?*
We have added a passage in our Methods section to explain the reasons of the lack of closure of our budget fluxes (p.7 ll.33 – p.8 l.2):
"In our analysis we disregarded the contribution of the horizontal and vertical mixing associated with the background diffusivity. We also used a fixed depth for the sigma layers that define the box boundaries, disregarding their vertical oscillations. Both approximations can result in small residuals in the budget analysis."

**DC9:** Page 18, lines 1-6 - *The DeltaE diagnostics is interesting. It accounts for two processes that can increase the export without changing the NCP: 1) The organic matter that is being transported laterally and that sinks out of the upper ocean increases the export and thus DeltaE. 2) The organic matter that is being transported laterally and that remineralizes in the upper box. This stimulates the biological activity which produces more organic matter which is sinks out of the upper ocean. In that case, NCP is not changed (the increase in PP compensates for the remineralization of the laterally supplied organic carbon) and export is increased which increases DeltaE. This two mechanisms should*

*be explained here, especially because in the discussion section it is shown that the second process dominates.*

We thank Referee nr.1 for his/her comment. In the end, we have decided to maintain the (extended) discussion of these two possible mechanisms in the Discussion section, "Implications and comparison with previous work" subsection.

**DC10:** Page 20, line 22 *"and quantify the contribute of the different zonal bands ..." I guess it should be contribution*

Thanks. We have corrected it as suggested.

**DC11:** Page 22, line 25 *"and quickly channel water ..." It should be channel.*

Thanks. We have corrected it as suggested.

**DC12:** Page 24, line 8 *"becomes particularly important the offshore waters" Some words are missing here.*

Thanks, we have corrected it to (p.25, ll.6-7):

"Among these vertical components, the advective+mixing vertical export becomes particularly important in the offshore waters."

**DC13:** Page 25, lines 18-20 - *The splitting between the contribution of the mean flow and of the eddy transport is not really clear here. See my second major concern above.*

As anticipated in our previous answer to MC2, we plan to illustrate in detail and detangle the mean and mesoscale contributions to the transport in a dedicated publication, currently in preparation. We have added references to: Arístegui et al. (2009), Gabric et al. (1993), Fischer et al. (2009).

**Answer to Referee #2, Josep L. Pelegrí**

We thank Dr. Josep L. Pelegrí for his careful review of our manuscript and for his thoughtful comments that will surely help to improve its quality. We tried to address all his comments and we include below a detailed answer to all the questions.

**Answers to Major comments**

**Major comment nr.1:**
*(1) Evaluation of the model's performance*
*This is a very critical aspect and the authors dedicate a significantly long section, including an appendix, to evaluate the performance of the model. They compare the numerical output with observations using different datasets: the near-surface seasonal circulation as inferred from surface drifters; the annual-mean sea surface height, sea surface temperature (SST) and sea surface salinity; the annual-mean mixed layer depth (MLD); the annual-mean surface chlorophyll; and the annual-mean and seasonal-mean net primary production (NPP).*
*I value this effort very much but, honestly, at the end of the Evaluation section I have important doubts on how good the model's performance is. Throughout this section the authors recognize the existence of substantial differences between model and field data, and also talk about model bias.*

**Answer to MC1:**
As stated by Dr. Pelegrí, we have invested quite some effort to carefully evaluate many aspects of our model. As a result, we feel that we are well aware of its strengths and its limitations. While there are clearly some issues, the results of our model evaluation are in line with most state of the art models – in many respects the fidelity of the model simulated fields is even better than that of most models. However, it is clear that models are never perfect, so the question we have to answer is to what degree biases and other types of errors will affect the results and the conclusions drawn from them. Our overall assessment is that, despite the biases, that the performance of our model is more than adequate to answer the main scientific question regarding the magnitude and the importance of the long-range lateral fluxes of organic carbon. Thanks to the information provided by a detailed model comparison with observations, we are also able to discuss in the paper how our results are affected by the observed biases, especially in the southern subregion of the Canary Upwelling System (CanUS), where we see the largest and most relevant differences from the observations. We address the different elements of this first main comment in sequence:

**A)** *In Figure 2 they show the spatial distribution of model-data differences for several surface fields. The differences are not negligible at all, as clearly seen by the range of values in the mean fields and the differences, e.g. SST (range of values is 12∘ C and range in deviations is 4∘C) and MLD (range of valuesis 100 m and range in deviations is 60 m).*

The SST biases are clearly significant but actually quite a bit smaller than implied by Dr. Pelegrí's comment, i.e., ±2°C.  The SST plot (Figure 2b) shows that differences between model and observations

lay in the interval [-0.75ºC,1ºC] in the large majority of the domain, with a large fraction of this bias having a range of only ±0.5ºC. Larger differences are confined to a very narrow coastal band. The region located south of Cape Blanc has the extensive bias. But also here, the (positive) bias has a range of only [0.5ºC,0.75ºC]. This warm bias is accompanied by a positive bias in salinity of about 0.5 (Figure 2c), leading to a near complete compensating with respect to their impact on density. Overall, we consider these biases to be small relative to the spatial and temporal variations. They are also too small to affect substantially primary production or the lateral export of organic carbon. Therefore we expect that these SST and salinity biases have a minor impact on our study. In response to this comment, we will discuss the SST and salinity biases and their impact on the study more explicitly.

The biases in the mixed layer depth are likely more relevant for our study. As highlighted by Dr. Pelegrí, while the modeled distribution agrees overall reasonably well with the observed one based on Argo-floats, our modeled MLD shows sharper gradients than the observed pattern resulting in rather large differences in the northern nearshore and the central offshore region. This could be a true bias of our model, but we also note that the Argo DT-0.2 MLD product was gridded on a relatively low-resolution 2ºx2º grid and that it has a rather limited coverage in the nearshore areas. As a result this product may not be able to properly capture strong gradients and overly smooth distribution relative to reality in regions with strong variations, such as ours. Given our MLD bias structure, it is feasible that some fraction of it could be attributed to biases in the Argo-based product.

In response to this comment, we have extended our discussion of the model biases in the Results and Discussion sections with a more in depth analysis.

- In particular, to be more precise in the discussion of our biases we have modified the Evaluation section as follows (p.8, l.30 – p.10, l.7):

"Despite the stratus cloud correction, the modeled SSTs are still a bit too warm in the southern sector of the CanUS. However, differences between model and observations are limited to the interval [-0.75°C,1°C] over the large majority of the domain, with a large fraction of this bias having a range of only ±0.5°C. Larger differences are confined to a very narrow coastal band. The model also captures the observed Sea Surface Salinity (SSS, Figure 2c) well; relevant negative differences are only observed in the southern CanUS, in connection with the warm SST bias, resulting in a compensation of the density. Overall, we consider these biases to be small relative to the spatial and temporal variations, therefore we expect these SST and SSS biases to have a minor impact on the conclusions of our study.

The modeled annual mean Mixed Layer Depth (MLD, Figure 2d) is consistent with the general pattern of the Argo-based MLD product, even tough the modeled pattern has sharper gradients. Deeper than observed values of the MLD are visible in the northern sector of the CanUS and in the nearshore waters of the southern sector of the CanUS. It is worth noting that the Argo dataset was generated on a relatively low resolution grid, i.e., 2°x 2°, and thus is likely underestimating lateral gradients. In addition, the float coverage in Eastern Boundary Current system is relatively low, owing to the strong currents and the offshore transport, making the

Argo-based MLD product vulnerable for biases in these regions. Nevertheless, some of the differences are likely real, as they appear also in other products. This is particularly the case for the overestimation of MLD in the nearshore region of the southern CanUS and in the long strip extending southwestward from the Canary Island, possibly due to biases in the position of the large-scale currents as evidenced by the differences in SSH (Figure 2a)."

- We have also divided the Discussion section into two subsections (namely "Implications and comparison with previous work" and "Limitation and caveats") and in the latter subsection we have created a dedicated paragraph in which we discuss the potential consequences of our biases in the model evaluation (p.30 l.27 – p.31 l.5):

"We also need to assess the potential impact of the physical/biogeochemical biases that we diagnosed in the Evaluation section. In the northern CanS our model overestimates the MLD depth; however our modeled MLD shows a meridional gradient that has the same trend as the observed one, with an extremely shallow mixed layer in the southern region below the Cape Verde front and deeper mixed layer in the north. This suggests that, even though we may potentially overestimate vertical mixing in the northern CanUS, this subregion would still be expected to be the only one in which this process is relevant. In the southern CanUS, our model shows a weaker than observed circulation and a deeper than observed chlorophyll and NPP maximum, which may lead to an underestimation of the lateral transport and therefore of the net heterotrophy of the water column. Both a shoaling of the biological production towards the surface characterized by more intense currents and an intensification of the circulation can in fact result in the strengthening of the lateral zonal and meridional organic carbon fluxes. However, an increase of the offshore zonal fluxes in the southern subregion could favor a more heterotrophic water column only if accompanied by an increase of the divergence of the flux, resulting in a substantial accumulation of organic carbon compared to the local production. In the meridional direction, an intensification of the alongshore Mauritanian current may instead increase the influx of organic carbon from the south into the Cape Verde frontal zone, fueling even further the deep respiration in the already strongly heterotrophic central CanUS."

*B) I particularly miss a comparison between the depth distribution of the modelled and observed particulate organic carbon (POC), which is of capital importance for this study. The seasonal results (Figure 5) show very large differences, possibly too large.*

We agree with Dr. Pelegrí regarding the necessity of having a comparison of modeled and observed POC and we thank him for this suggestion. To this end, we have conducted an evaluation of the modeled POC using 2 datasets: 1) the MODIS satellite estimate of surface POC (S-POC in the diagram); 2) the cruise POC measurements from AMT, ANT and Geotraces in the upper 200 m (POC in the diagram) located in the 0km-2000km offshore range of our analysis domain. Most of the in-situ data were collected in fall, especially in October, often in the far offshore region of our domain (cf. Figure MC1-1).

[Figure]

circle=AMT, square=ANT-XXIII/1, diamond=Geotraces

**Figure MC1-1:** Retrieved cruise POC measurements in the region of analysis corresponding to the Budget Analysis boxes. Data points are colored by sampling month and by maximum depth of the measurement. Circles=AMT (15-23), Squares=ANT, Diamonds=Geotraces.

[Figure]

Modeled POC and data were co-located in space and time using a daily ROMS climatology for the same 6 years. As visible from the resulting plot (Figure MC1-2), the magnitude of modeled and observed POC is the same and the vertically-integrated POC in the first 100m also corresponds. Due to our coastally-confined production (largely discussed in the model evaluation) combined with the fact that cruise data are mostly located offshore, and due to the deepening of the chlorophyll maximum in the southern productive subregion, we observe a deeper-than-expected POC maximum in the model. As also discussed in the paper, this may mean that if anything our model may underestimate the offshore transport in the CanUS (and especially in its southern sector), therefore implying that the already large magnitude of the offshore transport that we find may be a low estimate.

**Figure MC1-2:** mean POC profile in the CanUS compared to cruise data, from co-located POC, binned in depth to 10m depth intervals.

In response to this comment:

- We have added a comparison between the mean profiles of modeled and observed POC using cruise data, combining Figures MC1-1 and MC1-2 into Figure B3 in the Appendix of the manuscript, p.36.

- We have included a comparison with cruise and satellite POC in the annual mean Taylor diagram (Figure B4 of the manuscript, here visible as Figure MC1-3), both based on satellite estimates and on in situ measurements, p.36 and a comparison with satellite POC in the seasonal Taylor diagrams (Figure B5 in the manuscript, here visible as Figure MC1-4) p.37.

- We have introduced a paragraph dedicated to the evaluation of modeled POC in the Evaluation section (p.12 ll.19-29):

> "Modeled Particulate Organic Carbon (POC) concentrations have annual mean values between 5 mmolC/m$^3$ and over 20 mmolC/m$^3$ in the first 100m depth of the very productive shelf areas laying therefore in the range of in situ observations (Alonso-González et al., 2009; Arístegui et al., 2003; Santana-Falcón et al., 2016; Fischer et al., 2009). Concentrations decline in the offshore direction with a pattern similar to that of NPP and have maximum values located between 20 m depth in the shelf area and 70 m depth offshore. The modeled POC compares well to the limited in situ data (see Appendix B: Supplementary figures, Figure B3) especially with regard to the vertically-integrated stocks in the first 100m. However, due to our coastally-confined production combined with the fact that cruise data were mostly collected offshore, and due to the deepening of the chlorophyll maximum in the southern productive subregion, we observe a deeper-than-expected POC maximum in the model, in agreement with the vertical bias in CHL. Due to the absence of sediment resuspension and of a mechanism of disaggregation of the large detrital particles in the model, deep peaks of POC such as those present in Alonso-González et al. (2009) and Álvarez-Salgado and Arístegui (2015) are not observed in the annual mean modeled POC concentration."

- In the caption of Figure 7 from the manuscript, p.16, we have highlighted that the plot already presents the mean vertical profile of POC for the whole Canary Upwelling System.

[Figure]

**Figure MC1-3**: Annual mean Taylor diagram including:
1) an evaluation of surface particulate organic carbon (**S-POC**) using SeaWiFS satellite estimates;
2) a comparison with depth profiles of **POC** from cruise data through co-location of ROMS output in space and time.

For an additional discussion of the implications of having a shallower POC distribution we refer also to our answer to Anonymous Referee 1, in which we discuss the results of some sensitivity studies in terms of both transport and impact on NCP.

*C) The authors end this section referring to a Taylor diagram presented in Appendix B (Figure B3), concluding that there is a "good correlation between the modelled and observed fields both in the annual and in the seasonal means." They show the Taylor diagram for the annual-mean results and for the mean of the seasonal results. The authors argue that the Taylor diagram shows results comparable to other studies for upwelling systems. Rather than comparing with other studies, it would be better to look at the statistics and discuss whether the results are convincing or not. For the annual-mean, for example, SST, CHLA and MLD respectively have a (normalized) standard deviation of about 1.2, 0.6 and 1.4, and a (normalized) root-mean-square difference of 0.3, 0.7 and 0.8. The authors should discuss whether these values are reasonable or not.*

Following Dr. Pelegrí's comment, we have modified the description of our annual mean Taylor diagram in the Evaluation section in order to be more specific and we also have included the discussion of the new seasonal Taylor diagrams (see also point D of the Answers to MC1). The paragraph was modified as follows (p.14 ll.15-30):

> "As visible from the Taylor Diagrams (Appendix B: Supplementary figures, Figure B4 and Figure B5) the agreement between the pattern of the physical and biological variables of interest is also confirmed by the good correlation between modeled and observed fields for both the annual and the seasonal means. All the variables have a correlation of 0.7 or higher with the observations in the annual mean (except cruise data POC) and 0.68 or higher in the seasonal. In the annual mean, the values of the normalized standard deviations are particularly high for annual mean MLD (1.5), which is, as discussed above, due to a combination of too low variations in the Argo-based observational product and overestimation of the MLD variations by the model. Low values of the normalized standard deviations (STD) are observed for surface POC (0.65), CHL (0.6) and for net primary production (NPP1) (0.35), the latter corresponding to NPP from the SeaWiFS VGPM product. This is likely due to the weaker intensity of the modeled blooms. Interestingly, if modeled NPP is compared to the SeaWiFS CbPM product (NPP2), the normalized STD increases to 0.75, reflecting the rather large uncertainties in the NPP inferred from observations. In the annual mean, values of correlation and normalized STD for MLD, SST and CHL are comparable to those presented for the CanUS in the ROMS+NPZD study by Lachkar and Gruber (2011), despite the the boundaries of our grid being much further away, and therefore providing much less constraints on the modeled physics and biology in the region of interest. When compared to studies that used ROMS+NPZD in other upwelling systems such as the California Upwelling System (Gruber et al. (2011), whole domain), our Taylor diagram shows a slightly worse correlation and comparable normalized STD of surface CHL in the annual mean but a better seasonal representation, while modeled NPP has comparable performances."

*D) I am particularly confused by Figure B3b: how is this calculated, just an average mean? What is the meaning? Wouldn't it be much better to show all four seasonal diagrams? It would also help to include, as supplementary materials, diagrams for each subregion.*

The Taylor diagram in Figure B3b is calculated as the simple mean of the seasonal Taylor diagrams. As suggested by Dr. Pelegrí, we have decided to substitute this figure, and explicitly include in the appendix of the paper the four seasonal Taylor diagrams (Figure B5 of the Appendix in the manuscript, p.37), here visible in Figure MC1-4. We have now included surface POC (S-POC) compared against the SeaWiFS satellite estimates also in these diagrams.

[Figure]

**Figure MC1-4:** Seasonal Taylor diagrams, including surface particulate organic carbon (S-POC) through a comparison with SeaWiFS satellite product estimate. In the Summer diagram MLD was rescaled to MLD*=MLD/2. The summer $MLD_{STD}$ is therefore 2 times as big as the one represented in the plot, while the correlation remains unchanged.

**Major comment nr.2:**

*(2) Latitudinal partition of the domain*

*A) In several places of the Introduction and Discussion the authors recognize that the Cape Verde frontal zone is a natural boundary between the subtropical and tropical domains. Nevertheless, for most of their analysis on latitudinal variability they use a partition in three areas or subregions, as shown in Figure 3b, which is not properly justified. I imagine this is done as an attempt to grasp the character of the meridionally convergent region near Cape Blanc but, as it is clear from the velocity fields in Figures 3 and 5, this is not correct. In my opinion only the southern subregion would comprise an area with approximately coherent dynamics. My suggestion here is to use four subregions of different size: the northern one (25- 32◦ N) would correspond to an area with substantial mesoscalar activity, with eddies and filaments generated both south of the Canary Archipelago and at the upwelling front; the second area would represent the permanent and intense central upwelling region (21-25◦ N); the third are would concentrate on the convergent region immediately south of Cape Blanc, which is the root of the Cape Blanc giant filament (about 17-21◦ N, though these limits change with longitude); the southern area (9.5-17◦ N) would correspond to the tropical region. Right now most of the discussion is either on the results for the latitudinal-average picture or (to a lesser degree) for the three proposed regularly-spaced subregions. With this alternative partition, the paper would certainly become much more informative.*

*B) I value very much the authors' efforts to provide bulk figures for the entire region but I think that plotting these results may be very misleading. For example, the data in Figure 8 suggests that the zonal flux of organic carbon is more intense than meridional one. I doubt this very much: in my opinion this is only an artefact that the latitudinal average tends to cancel the contributions of the southward Canary Upwelling Current and northward Mauritania Current and Poleward Undercurrent (please see references below regarding the main currents in the CanUS). My suggestion is to produce fewer plots on the results for the entire region (Figure 9 is fine but some other plots may be replaced by tables) and instead show what is happening in each area: the CanUS is so large that it surely deserves a closer view for each subregion.*

**Answer to MC2:**

**A)** We agree with Dr. Pelegrí that other choices for the subregional Budget Analysis domain were also possible. Our partition serves to quantify both the alongshore convergence of particulate organic carbon from both north and south of Cape Blanc and the subsequent intense offshore flow that takes place along the Cape Verde front. The use of wide domains allows us to have a more robust measure of the fluxes in a region of high mesoscale variability. This partition also avoids us to place boundaries in critical regions such as around the Cape Verde convergence; placing boundaries in such flux-intense regions would make the results of our budget analysis very sensitive to the exact latitude of the boundary.

However, we have considered the latitudinal partition proposed by Dr. Pelegrí, and repeated our analysis on his proposed domains, as shown in Figure MC2-1. The changes basically consists in a sub-division of the central domain into two smaller zonal bands. Our northern and southern zonal bands

already satisfied Dr. Pelegrí's definitions, corresponding to a northern subregion rich in mesoscale activity (now only displaced by half degree) and a southern tropical subregion. The results of the new budget analysis are displayed in Figure MC2-2. As expected, northern and southern subregions are characterized by the same pattern of fluxes as those presented in the paper, since moving the southern boundary of the northern subregion by half a degree north does not affect the budget. The central subregion is split in a "central north" and "central south" zonal bands (green and orange lines). The impact of the offshore flux in these two zonal bands is very similar (Figure MC2-2, panel b). The flow of the Cape Verde front crosses the boundary between the "central north" and "central south" zonal band at about 1000km offshore, adding to the offshore flux in the "central south" subregion at this distance from the coast. However, this effect is an artifact generated by the split of the front in two segments. It thus does not add much to our understanding of the magnitude or the impact of the long-range offshore flux at these latitudes. As regard to the alongshore fluxes, we find that dividing the central subregion in two zonal bands does not clarify the source of the organic carbon that is exported offshore along the Cape Verde front. In fact, while before we could clearly identify the central subregion as a region of alongshore convergence of the organic carbon, now the budget for the "central north" and "central south" subregions depends strongly on the exact location of the intermediate boundary and the exact pathway of the Canary and Mauritanian currents near Cape Blanc. For example, in the 0km-100km offshore range, the "central north" subregion still exports southward more carbon than what it receives from its northern boundary, resulting in a net alongshore export to the "central south" subregion. In contrast, in the 100km-500km offshore range of the "central north" subregion a local recirculation of the carbon from the "central south" zonal band is visible in the meridional fluxes plot of Figure 11 of the paper. This recirculation induces a large net influx of organic carbon in the "central north" subregion. All these effects strongly depend on the exact location of the intermediate boundary in this region of intense flux convergence. As a consequence, we believe that the use of just one large central subregion for the budget analysis is more appropriate for our main purpose of understanding the magnitude and possibly the sources of the lateral offshore flux of organic carbon in the CanUS.

[Figure]

**Figure MC1-1:** Comparison between the latitudinal domains proposed by Dr. Pelegrí (red lines) and the domains used in the paper (blue lines).

[Figure]

**Figure MC1-2:** Budget analysis results based on the subregions proposed by Dr. Pelegrí. Trends of NCP and impact of the organic carbon fluxes (divergence of the flux / NCP) by subregion.

In response to this comment we have decided to explicitly illustrate the reasons behind our choice of the zonal partition of the domain into three subregions. This passage was added to the Methods section (p. 8, ll.9-14):

> "To highlight the different roles of the three fundamental zonal bands in the CanUS, we divided the EBUS into three subregions (Southern, Central, Northern), maintaining for each subregion the same five offshore domains as for the previous analysis and considering only the euphotic layer (0 m -100 m). Subregional boundaries were placed at 17°N and 24.5°N. This allows us to distinguish between three regimes of circulation and production of the CanUS: a northern subregion dominated by coastal upwelling and coastal filaments, a southern tropical subregion, and a central subregion where the Canary and Mauritanian Currents converge to form the Cape Verde front (Pelegrí and Peña-Izquierdo, 2015)."

**B)** We thank Dr. Pelegrí for this comment, and we agree with him on the fact that alongshore fluxes are locally more intense than the offshore fluxes in the nearshore. We also agree that, since Figure 8b shows the meridional fluxes averaged over the whole CanUS, the contribution of the northward and southward alongshore currents mostly cancel each other.

To address this comment, we have modified the text as follows (p.18, ll.1-6):

> "The lateral meridional flux (Figure 8b) shows a complex alternation of northward and southward fluxes, emerging from the integration of the meridional transport across a wide meridional band. Even though this flux is weaker than the zonal flux, this does not imply the absence of substantial alongshore currents within the domain. In fact, many of these currents get averaged out by the meridional integration. Despite this, the intense southward flowing Canary Current is still visible as a negative signature of the mean meridional flux near the coast. Northward fluxes, probably linked to an influx from the organic carbon-rich near equatorial region, are dominant further offshore."

In general, we still believe that an in depth discussion of the bulk fluxes is very relevant for our purpose of quantifying the overall magnitude of the lateral fluxes from the North African coast on the North Atlantic gyre.

For this reason, we have decided to keep the original bulk figures.

**Major comment nr.3:**
*(3) Upwelling of coastal and offshore inorganic nutrients*
*The coastal upwelling region is a source of inorganic nutrients to the surface layers in the coastal transition zone that are later exported offshore (e.g. Pelegrí et al., 2006; Pastor et al., 2008, 2013). Such a flux of inorganic nutrients is a prime element in the offshore net primary production and the sign of the NCP north of Cape Blanc. However, this issue is not mentioned in the manuscript until the Discussion. The subject is important enough to deserve careful attention when examining the sources and sinks for NPP, it is the difference between new production using the subsurface load of inorganic nutrients or production after remineralization.*
*The offshore waters in the southern subregion are also largely affected by the presence of upward Ekman pumping, i.e. offshore upwelling resulting from positive wind-stress curl. Again this is an important aspect in the dynamics and NPP balance of this subregion, which is again acknowledged very late in the manuscript and only partly discussed.*
*The model could be used to assess these different contributions. Perhaps this was not the objective of the authors, which is fine, but then the potential relevance of the upwelling and transport of inorganic nutrients on the NPP and NCP within the entire region should be properly discussed since early in the manuscript.*

We agree with Dr. Pelegrí on the importance of including a discussion of sources and sinks of nutrients in the region.

- For this reason we have  included in the paper a figure showing the lateral and vertical fluxes of inorganic nutrients (Figure B7 p. 39 of the manuscript, here Figure MC3-1).

- The figure was used in the Results section to better explain the pattern of NCP in the CanUS region, as follows (p.21, ll.5-11):

> "In the euphotic layer the pattern of production changes with latitude transitioning from a sharp offshore NCP gradient in the northern CanUS to an wide offshore extent of high NCP in the southern CanUS. These gradients can be explained by the pattern of the nutrients fluxes (see Appendix B: Supplementary figures, Figure B7). In the northern CanUS, nutrients are in fact mostly provided by coastal upwelling, while the positive signature of the wind-stress curl in the southern CanUS favors Ekman pumping of nutrients also offshore (Figure B7c). Intense production in the surroundings of Cape Blanc is likely due to the convergence of the alongshore nutrient fluxes (Figure B7b), in agreement with Auger et al. (2016) and Pastor et al. (2013)."

- The figure was also recalled in the Discussion in the context of the discussion of the two possible patterns of production and recycling that maintain the long-range offshore transport (p.27, ll.22-23):

> Further, the inorganic nutrients fluxes (see Appendix B: Supplementary figures, Figure B7) are of sufficient magnitude to refuel new growth of organic matter to replace that part that is lost by sinking.

[Figure]

**Figure MC3-1:** Inorganic Nitrogen fluxes in the first 100m. Horizontal fluxes were integrated in the first 100m, while vertical fluxes were sliced at 100m depth.

**Answers to Minor comments**

(4) p 1, l 10: *divergence or convergence?*
The divergence is negative, which means flow is convergent.

(5) p 2, l 11: *replace "and can create" by "can create."*
Thank you, it was a typo. We have corrected it as suggested.

(6) p 2, l 15: *"Arístegui."*
Thank you, it was a typo. We have corrected it as suggested.

(7) p 3, l 5: *other relevant references are Pelegrí et al. (2006) and Pastor et al. (2013).*
Thank you for your suggestion. We have added the references in the text.

(8) p 3, l 24: *also Pastor et al. (2013).*
Thank you, we have added the reference in the text.

(9) section on Methods: *how does the model calculate the vertical velocity?*
The vertical velocity is computed through integration of the mass-conservation equation of an incompressible fluid ( $\vec{\nabla} \cdot \vec{u} = 0$ ) from the ocean floor upwards. For detail about the equations solved by ROMS and its numerical solution technique we refer to Shchepetkin and McWilliams (2005) and the ROMS Wiki (https://www.myroms.org/wiki/Equations_of_Motion).

(10) caption of Figure 1: *Gran Canaria is cited in the caption but not located in the map.*
Thank you for the suggestion, we have marked the island of Gran Canaria in the picture with the letter "G", and we have recalled it in the caption.

(11) p 10, l 10-13: *please clarify.*
We have modified the passage as follows (p.11, ll.5-8):

> "Less well captured is the surface CHL in the productive southern sector of the CanUS, where the model substantially underestimates CHL at the surface. This is also the region where the model is biased too warm and salty, and where the modeled MLD exceeds the expected near-zero value, suggesting that this low surface CHL is primarily consequence of our physical biases in circulation and vertical stratification."

(12) caption of Figure 4: *VGPM is first mentioned here but it is defined nowhere in the manuscript.*
The used SeaWiFS VGPM product is described in detail in Table A3 (Appendix) among the products used for the Model evaluation.

(13) p 12, l 21-24: *asides the Canary Current and the Mauritanian Current you should also probably refer to the Canary Upwelling Current (associated to the coastal upwelling jet) and the Poleward Undercurrent (please see references below).*
Thank you. We have added these currents in our description as follows (p.10 ll.13-16):

> "Between the CC and the African coast, an intense and narrow Canary Upwelling Current (CUC) flows southward along the shelf (Mason et al., 2011). A poleward undercurrent (not shown) flows along the whole North African coast with its core typically centered at 200 m - 300m depth (Pelegrí and Benazzouz, 2015)."

(14) p 12, l 31-32: *". . . NPP is a better measure than chlorophyll for evaluating. . ."*
Thank you, we have corrected the sentence as suggested.

(15) p 13, l 1: *Pastor et al. (2013) is probably a better reference.*
Thank you for your suggestion, we have added this reference.

(16) caption Figure 6: *panel b also includes sediment remineralization?*
Yes, all panels include remineralization. We have adjusted the caption in order to make this clear.

(17) p 18, l 3-5: *are you using two different definitions for excess export?*
No, ΔE is always defined as ΔE= vertical export – NCP.
To avoid confusion we have substituted the symbol Δ (previously used in Figure 9) with ΔE both in the plot and in the caption. We have also adapted the legend of Figure 10 in line with this choice.

(18) p 20, l 1-2: *here and elsewhere it is best to not refer to lines, they should be defined in the figure's caption or legend (otherwise you would have to define them everywhere).*
Thank you for your suggestion. However, we have decided to maintain the references to the lines to ease the reading of the manuscript.

(19) p 20, l 27 and 33: *"north of the Cape Verde front. . ."*
Thank you for your suggestion, we have corrected this error throughout the whole manuscript.

(20) p 22, l 2: *". . .south of Cape Blanc."*
Thank you for your suggestion, we have corrected this error throughout the whole manuscript.

(21) *please revise caption of Figure 12.*
Thank you for highlighting this typo, we have revised the caption.

(22) Figure 13: *I suggest that you also show the meridional fluxes.*
Thank you for your suggestion. However, due to the length of the manuscript and to the fact that the vertical slices of the mean meridional fluxes do not add any substantial information that cannot be

inferred from the 2D plot of the meridional fluxes (Figure 11b), we have decided not to include this plot.

(23) caption Figure 13: *"vertical" rather than "vertcal."*
Thank you. We have corrected the typo as suggested.

(24) p 24, l 14: *"(Figure 13)."*
Thank you. We have corrected the typo as suggested.

(25) p 25, l 6-8: *this is likely an artefact of the SW-NE orientation of the coast.*
Thank you for your suggestion. We have explored this question, also observing some animations of the organic carbon concentration from the model output. However, we are still not sure that the inclination of the coastline can result in an onshore flux in the open waters of the northern CanUS. Instead, we clearly see the organic carbon being transported eastward by the Azores current at the northern boundary of the CanUS domain. As a consequence, we think that this current is responsible for the positive signature of the zonal flux in the northern subregion.

(26) p 25, l 10 and 15: *please include references.*
Thank you, we have added the following references: Arístegui et al. (2009), Gabric et al. (1993), Fischer et al. (2009).

(27) p 27, l 8: *see also Pastor et al. (2013).*
Thank you for your helpful suggestion. We have added this reference in the text.

(28) p 28, l 7: *"these."*
Thank you, we have corrected it as suggested.

(29) p 28, l 10-18: *usage of so many conditionals raises doubts on the reader.*
Thank you, we have revised the paragraph.

(30) p 28, l 27: *remove "from Section 4.1."*
Thank you, we have removed it.

(31) p 28, l 28: *is this the right way to cite a figure within a reference?*
We have modified the reference as follows (p.28, ll.34-35):
"The spatial pattern of modeled near-surface autotrophy (Figure 6b) agrees with the calculated global distribution of NCP (Williams et al., 2013, Figure 1), ..."

(32) p 29, l 3: *here and elsewhere separate numbers from units, i.e. "2000 km" rather than "2000km."*
Thank you, we have corrected it throughout the whole manuscript.

(33) p 29, l 16-17: *please revise writing.*
Thank you. We have revised the Conclusions as a whole.

(34) additional references: *asides those mentioned above, there are other works that would help better describe the circulation patterns in the CanUS, such as Mason et al.(2011), Peña-Izquierdo et al. (2012, 2015), Pelegrí and Peña-Izquierdo (2015), Pelegrí and Benazzouz (2015).*
Thank you for the valuable suggestions, we have included these references in the manuscript.

[revised manuscript text omitted]

---

## Author Response (AR2)

**Author's Response**

Dear Dr. Jack Middelburg,
we are very glad to read your decision, and we thank you for selecting our paper as a highlight on Biogeosciences.

As suggested, we have corrected the listed typos. We have also double checked the whole text and references, made a few other grammatical corrections, and adapted the numbers/unit spacing to the journal standards.

A few more corrections to the text were made in the Discussion and Conclusion sections, with the aim of improving the readability of the manuscript.
Please, find below our changes.

Best regards,
Elisa Lovecchio

[revised manuscript text omitted]